# Stochastic Gradient Variational Inference with Price's Gradient Estimator from Bures–Wasserstein to Parameter Space

**Kyurae Kim** [1]  **Qiang Fu** [2]  **Yi-An Ma** [3]  **Jacob R. Gardner** [1]  **Trevor Campbell** [4]

## Abstract

For approximating a target distribution given only its unnormalized log-density, stochastic gradient-based variational inference (VI) algorithms are a popular approach. For example, Wasserstein VI (WVI) and black-box VI (BBVI) perform gradient descent in measure space (Bures–Wasserstein space) and parameter space, respectively. Previously, for the Gaussian variational family, convergence guarantees for WVI have shown superiority over existing results for black-box VI with the reparametrization gradient, suggesting the measure space approach might provide some unique benefits. In this work, however, we close this gap by obtaining identical state-of-the-art iteration complexity guarantees for both. In particular, we identify that WVI's superiority stems from the specific gradient estimator it uses, which BBVI can also leverage with minor modifications. The estimator in question is usually associated with Price's theorem and utilizes second-order information (Hessians) of the target log-density. We will refer to this as Price's gradient. On the flip side, WVI can be made more widely applicable by using the reparametrization gradient, which requires only gradients of the log-density. We empirically demonstrate that the use of Price's gradient is the major source of performance improvement.

## 1. Introduction

Variational inference (VI; Jordan et al., 1998; Blei et al., 2017; Peterson & Hartman, 1989; Hinton & van Camp,

---
[1]University of Pennsylvania, Philadelphia, U.S. [2]Yale University, New Haven, U.S. [3]University of California San Diego, San Diego, U.S. [4]University of British Columbia, Vancouver, Canada. Correspondence to: Kyurae Kim <kyrkim@seas.upenn.edu>, Qiang Fu <qiang.fu@yale.edu>, Yi-An Ma <yianma@ucsd.edu>, Jacob R. Gardner <jacobrg@seas.upenn.edu>, Trevor Campbell <trevor@stat.ubc.ca>.

*Proceedings of the 43rd International Conference on Machine Learning*, Seoul, South Korea. PMLR 306, 2026. Copyright 2026 by the author(s).

1993) is a collection of algorithms for approximating a target distribution $\pi$ over some family of parametric distributions $\mathcal{Q}$ when only the unnormalized density of $\pi$, denoted by $\widetilde{\pi}$, is available. When $\pi$ is supported on $\mathbb{R}^d$ such that its associated potential function $U = -\log \widetilde{\pi}$ is in $\mathbb{R}^d \to \mathbb{R}$, it is common to leverage stochastic gradient-based algorithms, as they only require local information of $U$ (Graves, 2011; Salimans & Knowles, 2013; Wingate & Weber, 2013; Titsias & Lázaro-Gredilla, 2014; Ranganath et al., 2014; Kingma & Welling, 2014; Rezende et al., 2014). Most VI algorithms are designed to minimize the variational free energy, also known as the negative evidence lower bound (Jordan et al., 1999), defined as $\mathcal{F}(q) \triangleq \mathcal{E}(q) + \mathcal{H}(q)$, where $\mathcal{E}$ is the energy functional associated with $U$, while $\mathcal{H}$ is the Boltzmann entropy. That is, we solve

$$\underset{q \in \mathcal{Q}}{\text{minimize}} \quad \left\{ \mathcal{F}(q) = \mathrm{KL}(q, \pi) + \mathcal{F}(\pi) \right\}$$

through only zeroth-, first-, and second-order information of $U$. Since $\mathcal{F}$ is equal to the exclusive KL divergence (Kullback & Leibler, 1951) between $q$ and $\pi$ up to the constant $\mathcal{F}(\pi)$, this also minimizes $q \mapsto \mathrm{KL}(q, \pi)$.

A common approach for minimizing $\mathcal{F}$ is, informally speaking, to leverage some sort of stochastic gradient descent (SGD; Robbins & Monro, 1951; Bottou, 1999; Bottou et al., 2018; Bach & Moulines, 2011; Nemirovski et al., 2009; Shalev-Shwartz et al., 2011) scheme, where intractable terms (such as the gradient of the energy $\mathcal{E}$) are stochastically estimated. There are two popular ways to realize this conceptual algorithm. The most widely used approach in practice is to represent each $q_\lambda \in \mathcal{Q}$ via a Euclidean vector of *variational parameters* $\lambda \in \Lambda$, and run gradient descent on the Euclidean space of parameters $\Lambda \subseteq \mathbb{R}^p$ (Wingate & Weber, 2013; Kucukelbir et al., 2017; Titsias & Lázaro-Gredilla, 2014; Ranganath et al., 2014; Salimans & Knowles, 2013). This is now referred to as *black-box variational inference* (BBVI). The other approach is to define a tractable notion of measure-valued derivatives, which can directly perform gradient descent in measure space. In particular, recent advances in our understanding of the Wasserstein geometry (Villani, 2009; Chewi et al., 2025) and gradient flows (Ambrosio et al., 2005; Jordan et al., 1998) have con-

*Table 1.* Overview of Main Theoretical Results.

| Algorithm | Space | Gradient Estimator | Maximum Step Size | Iterations Complexity | Reference |
|---|---|---|---|---|---|
| SPGD | $\Lambda$ | Reparam. | $1/(dL\kappa)$ | $d\kappa^2 \mathrm{tr}(\mu\Sigma_*)^1/\epsilon$ | Domke et al. (2023) Kim et al. (2023) |
| SPBWGD | $\mathrm{BW}(\mathbb{R}^d)$ | Bonnet–Price | $1/(L\kappa^2)$ | $d\kappa\frac{1}{\epsilon}\log\frac{1}{\epsilon} + \kappa^3 \log\left(\Delta^2\frac{1}{\epsilon}\right)$ | Diao et al. (2023) |
| SPGD | $\Lambda$ | Bonnet–Price | $1/(L\kappa)$ | $d\kappa\frac{1}{\epsilon} + \sqrt{d}\kappa^{3/2}\log\left(\kappa\Delta^2\right)\frac{1}{\sqrt{\epsilon}} + \kappa^2\log\left(\Delta^2\frac{1}{\epsilon}\right)$ | Theorem 3.3 |
| SPBWGD | $\mathrm{BW}(\mathbb{R}^d)$ | Bonnet–Price | $1/(L\kappa)$ | $d\kappa\frac{1}{\epsilon} + \sqrt{d}\kappa^{3/2}\log\left(\kappa\Delta^2\right)\frac{1}{\sqrt{\epsilon}} + \kappa^2\log\left(\Delta^2\frac{1}{\epsilon}\right)$ | Theorem 3.2 |

*Note:* The complexity statements assume $\mu$-strong convexity and $L$-smoothness of $U$ (Assumption 3.1); Denote the initialization as $q_0$, the last iterate as $q_T$, and the global optimizer of $\mathcal{F}$ as $q_*$; $\epsilon > 0$ is the target accuracy level for ensuring $\mu\mathbb{E}\mathrm{W}_2(q_T, q_*)^2 \leq \epsilon$; $\kappa = L/\mu$ is the condition number of $U$; $\Delta^2 = \mu\mathrm{W}_2(q_0, q_*)^2$ is the distance between the initialization and the optimum.

tributed to the development of (parametric[1]) Wasserstein variational inference (WVI; Lambert et al., 2022; Diao et al., 2023; Huix et al., 2024; Talamon et al., 2025) algorithms that take this route. We also note that natural gradient VI algorithms (Khan & Rue, 2023; Khan & Nielsen, 2018; Lin et al., 2019; Tan, 2025) utilize parameter gradients while measuring distance using the KL pseudo-metric. However, NGVI will not be the focus of this work.

Given that the WVI methods utilize a proper metric (Villani, 2009) between measures—the Wasserstein-2 metric $\mathrm{W}_2$—it is natural to expect them to outperform BBVI. Indeed, current theoretical evidence suggests that this is the case. Consider the Gaussian variational family, also known as the Bures–Wasserstein space (Bures, 1969; Bhatia et al., 2019), set as

$$\mathcal{Q} = \mathrm{BW}(\mathbb{R}^d) \triangleq \{\mathrm{Normal}(m, \Sigma) \mid m \in \mathbb{R}^d, \Sigma \in \mathbb{S}_{\succ 0}^d\}.$$

Here, $(\mathrm{BW}(\mathbb{R}^d), \mathrm{W}_2)$ forms a metric space. Denoting the global minimizer as $q_* = \mathrm{Normal}(m_*, \Sigma_*) = \arg\min_{q \in \mathcal{Q}} \mathcal{F}(q)$, for a $\mu$-strongly convex and $L$-smooth potential $U$, the algorithm by Diao et al. (2023) requires $\mathrm{O}\left(d\kappa\epsilon^{-1}\log\epsilon^{-1} + \kappa^3\log\epsilon^{-1}\right)$ steps to ensure $\mu\mathbb{E}[\mathrm{W}_2(q, q_*)^2] \leq \epsilon$. In contrast, the BBVI equivalent with a certain covariance parametrization, the reparametrization gradient estimator (Ho & Cao, 1983; Rubinstein, 1992), and stochastic proximal gradient descent (SPGD; Nemirovski et al., 2009) requires $\mathrm{O}\left(d\kappa^2\mathrm{tr}(\mu\Sigma_*)\epsilon^{-1}\right)$ steps (Kim et al., 2023; Domke et al., 2023). It might therefore appear that the guarantees for BBVI are weaker in the limit $\epsilon \to 0$.

In this work, we demonstrate that the difference in theoretical guarantees originates from the specific gradient estimator for the scale parameter used in stochastic implementations of WVI rather than the geometry being used. The estimator in question can be derived via Price's theorem (Price, 1958) and leverages second-order information (Hessians) of the target log-density. We will refer to this as Price's gradient esti-

mator. Through Stein (Liu, 1994; Stein, 1981) or Price's theorem, BBVI with SPGD can also make use of essentially the same gradient estimator, resulting in an iteration complexity of $\mathrm{O}\left(d\kappa\epsilon^{-1} + \sqrt{d}\kappa^{3/2}\log\left(\kappa\Delta^2\right)\epsilon^{-1/2} + \kappa^2\log\epsilon^{-1}\right)$. Furthermore, to ensure a fair comparison, we present a refined analysis of the WVI counterpart of SPGD (Diao et al., 2023), resulting in an iteration complexity of $\mathrm{O}\left(d\kappa\epsilon^{-1} + \sqrt{d}\kappa^{3/2}\log\left(\kappa\Delta^2\right)\epsilon^{-1/2} + \kappa^2\log\epsilon^{-1}\right)$. These results suggest that the specific implementation of BBVI studied here and WVI might not be that different after all. While this has been suggested by Yi & Liu (2023); Hoffman & Ma (2020), this work further supports this fact by contributing a non-asymptotic discrete-time analysis. Our theoretical results are organized in Table 1.

In addition, we demonstrate that WVI can also leverage the reparametrization gradient traditionally used in BBVI (Titsias & Lázaro-Gredilla, 2014; Kingma & Welling, 2014; Rezende et al., 2014). Unlike Price's gradient previously used in WVI, the reparametrization gradient only requires first-order information ($\nabla U$). Thus, the resulting WVI algorithm should be more widely applicable in practice. Given this fact, we empirically compare the performance of BBVI and WVI, both with the Hessian-based gradient estimators and the reparametrization gradient. Results demonstrate that a large fraction of the performance difference stems from the use of Hessian-based gradients, supporting the claim that the gradient estimator is the main source of performance.

## 2. Background

**Notation.** For any $x, y \in \mathbb{R}^d$, we denote the Euclidean inner product and norm as $\langle x, y \rangle = x^\top y$ and $\|x\|_2 \triangleq \sqrt{\langle x, x \rangle}$. For any matrix $A, B \in \mathbb{R}^{d \times d}$, $\mathrm{tr}(A) \triangleq \sum_{i=1}^d A_{ii}$, $\langle A, B \rangle_\mathrm{F} \triangleq \mathrm{tr}(A^\top B)$, $\|A\|_\mathrm{F} \triangleq \sqrt{\langle A, A \rangle_\mathrm{F}}$, and the $\ell_2$ operator norm of $A$ is denoted as $\|A\|_2$. The symmetric and positive definite subsets of $\mathbb{R}^{d \times d}$ will be denoted as $\mathbb{S}^d$ and $\mathbb{S}_{\succ 0}^d$, while $\mathbb{L}_{\succ 0}^d$ denotes the set of lower-triangular matrices with strictly positive eigenvalues. We will represent a measure and its density with

---

[1]We focus on WVI on *parametric* families, which excludes particle-based WVI methods.

the same symbol. For the space of square-integrable measures $\mathcal{P}_2(\mathcal{X}) \triangleq \{q \mid \int_{\mathcal{X}} \|x\|^2 \mathrm{d}q(x) < +\infty\}$, for some $q \in \mathcal{P}_2(\mathbb{R}^d)$, the set of integrable functions is denoted as $\mathrm{L}^2(q) \triangleq \{f \mid \int \|f\|_2^2 \mathrm{d}q < +\infty\}$. For any two probability measures $p, q \in \mathcal{P}_2(\mathbb{R}^d)$, we denote the set of couplings between the two as $\Psi(p, q)$. Then the squared Wasserstein-2 distance between $p$ and $q$ is $\mathrm{W}_2(p, q)^2 \triangleq \inf_{\psi \in \Psi(p,q)} \int_{\mathbb{R}^d \times \mathbb{R}^d} \|x - y\|_2^2 \, \mathrm{d}\psi(x, y)$. For some measurable map $M : \mathbb{R}^d \to \mathbb{R}^d$ and measure $q$ supported on $\mathbb{R}^d$, $M_{\#q}$ denotes the corresponding push-forward measure. Unless stated otherwise, the coupling attaining the infimum of $\mathrm{W}_2$, $\psi^* \in \Psi(p, q)$, is referred to as "the optimal coupling," which is guaranteed to exist (Villani, 2009, Theorem 4.1) and is unique by Brenier's Theorem (Brenier, 1991).

### 2.1. Problem Setup

Our focus will be on first-order stochastic optimization algorithms for solving the problem

$$\underset{q \in \mathcal{Q}}{\text{minimize}} \; \left\{ \mathcal{F}(q) \triangleq \mathcal{E}(q) + \mathcal{H}(q) \right\},$$

where
$$\begin{aligned} \mathcal{E}(q) &\triangleq \int_{\mathbb{R}^d} U(z) \, q(\mathrm{d}z) && \text{(Energy)} \\ \mathcal{H}(q) &\triangleq \int_{\mathbb{R}^d} \log q(z) \, q(\mathrm{d}z) \, . && \text{(Boltzmann entropy)} \end{aligned}$$

We consider the "non-conjugate" setup, where $\mathcal{E}$ is intractable due to the expectation over $q$. Suppose we can parametrize each $q_\lambda \in \mathcal{Q}$ with a Euclidean vector of parameters $\lambda \in \Lambda$. Then it is equivalent to minimize $\lambda \mapsto \mathcal{F}(q_\lambda)$ over the Euclidean parameter space $\Lambda \subseteq \mathbb{R}^p$.

Informally speaking, when $U$ is "regular," $\mathcal{E}$ also tends to be regular. For instance, if $U$ is Lipschitz-smooth, then $\mathcal{E}$ also tends to exhibit appropriate notion of Lipschitz-smoothness (Domke, 2020; Diao et al., 2023; Lambert et al., 2022). The entropy term $\mathcal{H}$, however, does not enjoy Lipschitz smoothness in general. For instance, for Gaussians $q = \mathrm{Normal}(m, \Sigma)$, $\mathcal{H}(q)$ blows up as the covariance $\Sigma$ becomes singular. Typically, in optimization, such non-smoothness is remedied by relying on proximal gradient algorithms (Wright & Recht, 2021, §9.3). Indeed, for minimizing $\mathcal{F}$, stochastic proximal gradient algorithms have been proposed for both the Bures–Wasserstein (Diao et al., 2023) and Euclidean parameter spaces (Domke, 2020). In the following sections, we will introduce these algorithms.

### 2.2. Wasserstein Variational Inference via Stochastic Proximal Bures–Wasserstein Gradient Descent

Since the seminal work of Jordan et al. (1998), it is known that $\mathcal{F}$ can be minimized by simulating its Wasserstein gradient flow. The forward-backward discretization of this flow results in the Wasserstein-analog of proximal gradient descent (Wibisono, 2018; Bernton, 2018; Salim et al., 2020) operating on the metric space $(\mathcal{P}_2(\mathbb{R}^d), \mathrm{W}_2)$. This algorithm, however, is not directly implementable. Recently, Lambert et al. (2022); Diao et al. (2023) demonstrated that, by constraining optimization to the *Bures*–Wasserstein manifold $\mathrm{BW}(\mathbb{R}^d) \subset \mathcal{P}_2(\mathbb{R}^d)$ (Bures, 1969; Bhatia et al., 2019), the algorithm becomes implementable. In particular, the proximal Bures–Wasserstein gradient descent scheme iterates, for each $t \geq 0$,

$$\begin{aligned} q_{t+1/2} &= (\mathrm{Id} - \gamma_t \nabla_{\mathrm{BW}} \mathcal{E}(q_t))_{\#q_t} \qquad (1) \\ q_{t+1} &= \mathrm{JKO}_{\gamma_t \mathcal{H}}(q_{t+1/2}) \, , \end{aligned}$$

where, for any $q = \mathrm{Normal}(m, \Sigma) \in \mathrm{BW}(\mathbb{R}^d)$, the Bures–Wasserstein gradient of $\mathcal{E}$ can be derived (Lambert et al., 2022, Appendix C) as

$$\nabla_{\mathrm{BW}} \mathcal{E}(q) \triangleq x \mapsto \nabla_m \mathcal{E}(q) + 2\nabla_\Sigma \mathcal{E}(q)(x - m) \, ,$$

while, for any $\mathcal{G} : \mathrm{BW}(\mathbb{R}^d) \to \mathbb{R} \cup \{+\infty\}$,

$$\mathrm{JKO}_\mathcal{G}(q) \triangleq \underset{p \in \mathrm{BW}(\mathbb{R}^d)}{\arg\min} \left\{ \mathcal{G}(p) + (1/2)\mathrm{W}_2(p, q)^2 \right\} .$$

is the Wasserstein-analog of the proximal operator, commonly referred to as the "JKO operator." For the special case of $\mathcal{G} = \gamma_t \mathcal{H}$ and the Bures–Wasserstein space, the JKO operator admits the tractable closed-form expression (Wibisono, 2018, Example 7)

$$\mathcal{N}(\mu_t, \Sigma_{t+1}) = \mathrm{JKO}_\mathcal{G}(\mathcal{N}(\mu_t, \Sigma_{t+1/2})) \, ,$$

where

$$\Sigma_{t+1} = \frac{1}{2} \left( \Sigma_{t+1/2} + 2\gamma_t \mathrm{I}_d + \left( \Sigma_{t+1/2} \left( \Sigma_{t+1/2} + 4\gamma_t \mathrm{I}_d \right) \right)^{1/2} \right) ,$$

which is key to obtaining an implementable algorithm.

Still, the Bures–Wasserstein gradient $\nabla_{\mathrm{BW}} \mathcal{E}(q)$ involves expectations over $q = \mathrm{Normal}(\mu, \Sigma)$ that are generally not tractable. Therefore, these have to be replaced with stochastic estimates (Lambert et al., 2022) of $\nabla_m \mathcal{E}$ and $\nabla_\Sigma \mathcal{E}$ resulting in the estimator

$$\widehat{\nabla_{\mathrm{BW}}} \mathcal{E}(q; \epsilon) \triangleq x \mapsto \widehat{\nabla_m} \mathcal{E}(q; \epsilon) + 2\widehat{\nabla_\Sigma} \mathcal{E}(q; \epsilon)(x - m) \, ,$$

where $\epsilon \sim \varphi = \mathrm{Normal}(0_d, \mathrm{I}_d)$ is standard Gaussian noise. Replacing $\nabla_{\mathrm{BW}} \mathcal{E}$ in Eq. (1) with $\widehat{\nabla_{\mathrm{BW}}} \mathcal{E}$ results in stochastic proximal Bures–Wasserstein gradient descent (SPBWGD; Diao et al., 2023). Lambert et al.; Diao et al. rely on the estimators

$$\begin{aligned} \widehat{\nabla_m^{\mathrm{bonnet}}} \mathcal{E}(q; \epsilon) &\triangleq \nabla U(Z) \\ \widehat{\nabla_\Sigma^{\mathrm{price}}} \mathcal{E}(q; \epsilon) &\triangleq (1/2)\nabla^2 U(Z) \, , \qquad (2) \end{aligned}$$

where $Z = \mathrm{cholesky}(\Sigma)\epsilon + \mu$. The fact that these estimators are unbiased follows from the theorems by Bonnet (1964) and Price (1958) or Riemannian geometry (Altschuler et al., 2021, Appendix B.3). For each $t \geq 0$, the resulting update

rule for the iterate $q_t = \text{Normal}(m_t, \Sigma_t)$ is

$$m_{t+1} = m_t - \gamma_t \widehat{\nabla_{m_t} \mathcal{E}}(q_t; \epsilon_t)$$

$$M_{t+1} = \mathrm{I}_d - 2\gamma_t \widehat{\nabla_{\Sigma_t} \mathcal{E}}(q_t; \epsilon_t)$$

$$\Sigma_{t+1/2} = M_{t+1} \Sigma_t M_{t+1}^\top$$

$$\Sigma_{t+1} = \frac{1}{2}\Big(\Sigma_{t+1/2} + 2\gamma_t \mathrm{I}_d + \big(\Sigma_{t+1/2}\big(\Sigma_{t+1/2} + 4\gamma_t \mathrm{I}_d\big)\big)^{1/2}\Big)$$

where the standard Gaussian noise sequence $(\epsilon_t)_{t \geq 0}$ is sampled as $\epsilon_t \overset{\text{i.i.d.}}{\sim} \varphi$. Note the update rule for $\Sigma_{t+1/2}$ is different from the one originally presented by Diao et al. (2023); a transpose has been added to $M_{t+1}$. This change will become necessary later in Section 4 when we replace $\widehat{\nabla_\Sigma \mathcal{E}}$ with an estimator that is not almost surely symmetric.

### 2.3. Black-Box Variational Inference via Stochastic Proximal Gradient Descent

An alternative to SBWPGD is to optimize over the Euclidean space of parameters $\Lambda$. Recall, in this case, each $q_\lambda \in \Lambda$ is assumed to be associated with a Euclidean vector $\lambda \in \Lambda$. Then, if we have access to an unbiased estimator of $\nabla_\lambda \mathcal{E}(q_\lambda)$, denoted as $\widehat{\nabla_\lambda \mathcal{E}}(q_\lambda)$, $\mathcal{F}$ can be minimized via SPGD, which, for each $t \geq 0$, updates the *variational parameters* $\lambda_t$ as

$$\lambda_{t+1/2} = \lambda_t - \gamma_t \widehat{\nabla_{\lambda_t} \mathcal{E}}(q_{\lambda_t}; \epsilon_t)$$

$$\lambda_{t+1} = \text{prox}_{\lambda \mapsto \gamma_t \mathcal{H}(q_\lambda)}\big(\lambda_{t+1/2}\big),$$

where $\epsilon_t \overset{\text{i.i.d.}}{\sim} \varphi$ is some randomness source and prox is the canonical Euclidean proximal operator (Parikh & Boyd, 2014) defined as, for any proper lower semi-continuous convex function $g : \mathbb{R}^p \to \mathbb{R} \cup \{+\infty\}$,

$$\text{prox}_g(\lambda) \triangleq \underset{\lambda' \in \Lambda}{\arg\min}\left\{g(\lambda') + \frac{1}{2}\|\lambda' - \lambda\|_2^2\right\}.$$

For some classes of variational families $\mathcal{Q}$ and parametrizations, the proximal operator can be made tractable (Domke, 2020). Before that, however, we must come up with an unbiased estimator of the parameter gradient $\nabla_\lambda \mathcal{E}(q_\lambda)$.

Suppose the variational family $\mathcal{Q}$ and the parametrization $\lambda \mapsto q_\lambda$ satisfy the following:

**Definition 2.1.** *For some $\Lambda \subset \mathbb{R}^p$, the variational family $\mathcal{Q} = \{q_\lambda \mid \lambda \in \Lambda\}$ is referred to as a reparameterizable family if there exists some bijective map $\phi_\lambda : \mathbb{R}^d \to \mathbb{R}^d$ differentiable with respect to $\lambda$ and a base distribution $\varphi \in \mathcal{P}_2(\mathbb{R}^d)$ such that, for all $\lambda \in \Lambda$,*

$$Z \sim q_\lambda \quad \Leftrightarrow \quad Z \overset{\mathrm{d}}{=} \phi_\lambda(\epsilon); \ \epsilon \sim \varphi.$$

Here, $\overset{\mathrm{d}}{=}$ is equivalence in distribution. Then an immediate option for estimating $\nabla_\lambda \mathcal{E}(q_\lambda)$ is to use the reparametrization gradient (Ho & Cao, 1983; Rubinstein, 1992; see also

Mohamed et al., 2020)

$$\widehat{\nabla_\lambda^{\text{rep}} \mathcal{E}}(q_\lambda) \triangleq \nabla_\lambda \phi_\lambda(\epsilon) \nabla U(\phi_\lambda(\epsilon)), \tag{3}$$

which can be derived by combining the law of the unconscious statistician with the Leibniz integral rule. This combination of SGD with the reparametrization gradient—commonly referred to as BBVI—is widely used in practice through probabilistic programming frameworks such as Stan (Carpenter et al., 2017), Turing (Fjelde et al., 2025), Pyro (Bingham et al., 2019), and PyMC (Patil et al., 2010).

The wide adoption of BBVI in practice is partly due to its flexibility: Definition 2.1 applies to a very wide range of families from Gaussians (Titsias & Lázaro-Gredilla, 2014) to normalizing flows (Rezende & Mohamed, 2015). Furthermore, Eq. (3) uses only gradients of $U$, which can be efficiently computed via automatic differentiation (Kucukelbir et al., 2017). In this work, however, we will further restrict our attention to the Gaussian variational family with a *specific* parametrization:

**Assumption 2.2.** *The variational family $\mathcal{Q}$ is the Gaussian variational family, where each member $q_\lambda = \text{Normal}(m, CC^\top) \in \mathcal{Q}$ is parametrized as*

$$\Lambda = \big\{\lambda = (m, \text{vec}(C)) \mid m \in \mathbb{R}^d, C \in \mathbb{L}_{\succ 0}^d\big\} \subset \mathbb{R}^p$$

*while the reparametrization function is set as*

$$\phi_\lambda(\epsilon) = C\epsilon + m \quad and \quad \varphi = \text{Normal}(0_d, \mathrm{I}_d).$$

Under this parametrization, Eq. (3) reduces to

$$\widehat{\nabla_\lambda^{\text{rep}} \mathcal{E}}(q_\lambda; \epsilon) = \begin{bmatrix} \widehat{\nabla_m^{\text{bonnet}} \mathcal{E}}(q_\lambda; \epsilon) \\ \widehat{\nabla_C^{\text{rep}} \mathcal{E}}(q_\lambda; \epsilon) \end{bmatrix} = \begin{bmatrix} \nabla U(\phi_\lambda(\epsilon)) \\ \epsilon \nabla U(\phi_\lambda(\epsilon))^\top \end{bmatrix}. \tag{4}$$

Furthermore, the proximal operator for the entropy has the closed-form solution (Domke, 2020; Domke et al., 2023)

$$(m, C') = \text{prox}_{\lambda \mapsto \gamma_t \mathcal{H}(q_{\lambda_t})}((m, C)), \text{ where}$$

$$[C']_{ij} = \begin{cases} (1/2)\big(C_{ii} + \sqrt{C_{ii} + 4\gamma_t}\big) & \text{if } i = j \\ C_{ij} & \text{if } i \neq j. \end{cases}$$

Compared to alternative ways to parametrize Gaussians, this "linear" parametrization is particularly well-behaved (Kim et al., 2023) and also computationally efficient: each step of BBVI only needs $O(d^2)$ operations except for evaluating $\nabla U$. Furthermore, it has been shown that the Bures–Wasserstein gradient $\nabla_{\text{BW}} \mathcal{F}$ is equal to the parameter gradient $\nabla_\lambda \mathcal{F}(q_\lambda)$ under the parametrization of Assumption 2.2 up to a coordinate transformation (Yi & Liu, 2023).

### 2.4. Price Gradient Estimators

A crucial point here is that, for the Gaussian variational family, the estimators traditionally used in WVI (Eq. (7)) and BBVI (Eq. (3)) both target the same quantities up to constant factor adjustments and are therefore interchangeable.

**Proposition 2.3.** *For any twice-differentiable function $f$, Gaussian $q = \mathrm{Normal}(m, \Sigma)$, and assuming the expectations exist,*

$$\nabla_\Sigma \mathbb{E}_q f = \frac{1}{2}\mathbb{E}_q \nabla^2 f \qquad (5)$$

$$= \frac{1}{2}\Sigma^{-1}\mathbb{E}_{X \sim q}\Big[(X - m)\nabla f(X)^\top\Big] . \qquad (6)$$

*Proof.* Eq. (5) is Price's theorem (Price, 1958), while Eq. (6) is Stein's identity (Stein, 1981; Liu, 1994). □

Denoting $\Sigma = CC^\top$, an immediate corollary is

$$\nabla_C \mathcal{E}(q_\lambda) = C^\top \mathbb{E}_q \nabla^2 U$$

$$= C^{-1}\mathbb{E}_{X \sim q}\big[(X - m)\nabla U(X)^\top\big] , \qquad (7)$$

where, when restricting $C \in \mathbb{L}^d_{\succ 0}$, the gradient only needs to be projected to the lower-triangular subspace (`tril`). Under Assumption 2.2, $C^{-1}(X - m)\nabla U(X)^\top$ exactly corresponds to the reparametrization gradient in Eq. (3). At the same time, Eq. (7) points towards an analog of $\nabla^{\mathrm{price}}_\Sigma$ for the scale parameter $C$:

$$\widehat{\nabla^{\mathrm{price}}_C}\mathcal{E}(q_\lambda; \epsilon) = C^\top \nabla^2 U(X) ,$$
$$\text{where} \quad X = \phi_\lambda(\epsilon) . \qquad (8)$$

Conveniently, this estimator also stays unbiased when $\nabla^2 U$ is replaced with an unbiased estimator of $\nabla^2 U$, enabling doubly stochastic optimization (Titsias & Lázaro-Gredilla, 2014). Similarly, Proposition 2.3 points towards a gradient estimator that could be used in WVI,

$$\widehat{\nabla^{\mathrm{rep}}_\Sigma}\mathcal{E}(q_\lambda; \epsilon) = \Sigma^{-1}(X - m)\nabla U(X)^\top, \qquad (9)$$
$$\text{where} \quad X = \phi_\lambda(\epsilon) .$$

Note that similar remarks have already been made by Rezende et al. (2014); Lin et al. (2025); Graves (2011); Opper & Archambeau (2009). Therefore, the use of these estimators is by no means new. However, these Hessian-based estimators have not been widely adopted in BBVI, nor have they been analyzed in detail.

A natural question here is how much the choice of gradient estimator affects the performance of different algorithms. Past experience in stochastic gradient-based VI has shown that the choice of gradient estimator crucially affects performance both in practice (Kucukelbir et al., 2017; Geffner & Domke, 2020a; 2021; 2018; 2020b; Agrawal et al., 2020; Miller et al., 2017; Wang et al., 2024; Fujisawa & Sato, 2021; Buchholz et al., 2018) and in theory (Kim et al., 2024; Xu et al., 2019; Luu et al., 2025). Indeed, in our theoretical analysis, we will demonstrate that, once we use the same gradient estimator, the state-of-the-art iteration complexities of SPBWGD and SPGD become the same.

## 3. Theoretical Analysis

### 3.1. Theoretical Setup

For our theoretical analysis, we assume the following regularity conditions on $U$.

**Assumption 3.1.** *The potential $U : \mathbb{R}^d \to \mathbb{R}$ is twice differentiable and there exists some $\mu \in (0, +\infty)$ and $L \in [0, +\infty)$ such that, for all $z \in \mathbb{R}^d$, the following holds:*

$$\mu \mathrm{I}_d \quad \preceq \quad \nabla^2 U(z) \quad \preceq \quad L\mathrm{I}_d .$$

This assumption corresponds to assuming that the density of $\pi$ is $\mu$-log-concave and $L$-log-smooth, and has been widely used to establish the iteration complexity of stochastic gradient-based VI (Kim et al., 2023; Domke et al., 2023; Lambert et al., 2022; Diao et al., 2023) and sampling algorithms (Chewi, 2024). Crucially, the energy $\mathcal{E}$ is now well behaved: On the Bures–Wasserstein geometry, it is $\mu$-geodesically convex and $L$-geodesically smooth (Diao et al., 2023). Similarly, under Assumption 2.2, $\lambda \mapsto \mathcal{E}(q_\lambda)$ is $\mu$-strongly convex and $L$-smooth (Domke, 2020).

For stochastic first-order optimization algorithms, the choice of step size *schedule* is crucial for obtaining tight bounds (Bach & Moulines, 2011). In all cases, we will consider a two-stage step size schedule (Gower et al., 2019; Stich, 2019) of the form of, for some base step size $\gamma_0 \in (0, +\infty)$, switching time $t_* \geq 0$, and offset $\tau \geq 0$,

$$\gamma_t = \begin{cases} \gamma_0 & \text{if } t < t_* \\ \frac{1}{\mu}\frac{2(t+\tau)+1}{(t+\tau+1)^2} & \text{if } t \geq t_* . \end{cases} \qquad (10)$$

This two-stage schedule holds the step size constant for a certain period ($t \in \{0, \ldots, t_* - 1\}$) and then starts decreasing the step size at a rate of $\gamma_t \asymp 1/(\mu t)$. The choice of asymptote $1/(\mu t)$ ensures an optimal asymptotic convergence rate of $\mathrm{O}(1/T)$ for strongly convex objectives (Lacoste-Julien et al., 2012; Shamir & Zhang, 2013).

### 3.2. Main Results

We now present the iteration-complexity guarantees for VI with SPBWGD and SPGD using Price's gradient for the scale/covariance component. First, the Bonnet–Price estimator of the Bures–Wasserstein gradient is formally defined in functional form as, for any $q = \mathrm{Normal}(m, \Sigma)$,

$$\widehat{\nabla^{\mathrm{Bonnet–Price}}_{\mathrm{BW}}}\mathcal{E}(q; \epsilon)$$
$$\triangleq x \mapsto \widehat{\nabla^{\mathrm{bonnet}}_m}\mathcal{E}(q; \epsilon) + 2\widehat{\nabla^{\mathrm{price}}_\Sigma}\mathcal{E}(q; \epsilon)(x - m)$$
$$= x \mapsto \nabla U(Z) + \nabla^2 U(Z)(x - m) ,$$

where $Z = \mathrm{cholesky}(\Sigma)\epsilon + \mu$ and $\epsilon \sim \varphi = \mathrm{Normal}(0_d, \mathrm{I}_d)$. Then we obtain the following iteration complexity:

**Theorem 3.2** (SPBWGD). *Suppose Assumption 3.1 holds and the gradient estimator $\widehat{\nabla_{\mathrm{BW}}^{\mathrm{bonnet-price}}}\mathcal{E}$ is used. Then, for any $\epsilon > 0$, there exists some $t_*$ and $\tau$ (shown explicitly in the proof) such that running stochastic proximal Bures–Wasserstein gradient descent with the step size schedule in Eq. (10) with $\gamma_0 = 1/(10L\kappa)$ guarantees*

$$T \gtrsim d\kappa\frac{1}{\epsilon} + \sqrt{d}\,\kappa^{3/2}\log(\kappa\Delta^2)\frac{1}{\sqrt{\epsilon}} + \kappa^2\log\left(\Delta^2\frac{1}{\epsilon}\right)$$
$$\Rightarrow \quad \mu\mathbb{E}[\mathrm{W}_2(q_T, q_*)^2] \le \epsilon \, ,$$

*where $\Delta^2 = \mu\mathrm{W}(q_0, q_*)^2$.*

*Proof.* The full proof is deferred to Section D.1.3. $\square$

This improves over the $\mathrm{O}(d\kappa\epsilon^{-1}\log\epsilon^{-1} + \kappa^3\log(\Delta\epsilon^{-1}))$ complexity obtained by Diao et al. (2023, Thm 5.8). In particular, our result allows for step sizes larger by a factor of $\kappa$. Consequently, the dependence on $\kappa$ in the non-asymptotic term $(\log 1/\epsilon)$ is improved by a factor of $\kappa$. Furthermore, the asymptotic complexity in $\epsilon \to 0$ is improved by a factor of $\log 1/\epsilon$.

Let's now compare this result against the iteration complexity of SPGD. Formally, we define the Bonnet–Price gradient estimator

$$\widehat{\nabla_\lambda^{\mathrm{Bonnet-Price}}}\mathcal{E}(q_\lambda; \epsilon) \triangleq \begin{bmatrix} \widehat{\nabla_m^{\mathrm{bonnet}}}\mathcal{E}(q_\lambda; \epsilon) \\ \widehat{\nabla_C^{\mathrm{price}}}\mathcal{E}(q_\lambda; \epsilon) \end{bmatrix} = \begin{bmatrix} \nabla U(Z) \\ C^\top \nabla^2 U(Z) \end{bmatrix} .$$

Using this estimator in BBVI with PSGD results in the following iteration complexity.

**Theorem 3.3** (SPGD). *Suppose Assumption 3.1 holds and the gradient estimator $\widehat{\nabla_\lambda^{\mathrm{bonnet-price}}}\mathcal{E}$ is used. Then, for any $\epsilon > 0$, there exists some $t_*$ and $\tau$ (stated explicitly in the proof) such that running stochastic proximal gradient descent with the step size schedule in Eq. (10) with $\gamma_0 = 1/(10L\kappa)$ guarantees*

$$T \gtrsim d\kappa\frac{1}{\epsilon} + \sqrt{d}\,\kappa^{3/2}\log(\kappa\Delta^2)\frac{1}{\sqrt{\epsilon}} + \kappa^2\log\left(\Delta^2\frac{1}{\epsilon}\right)$$
$$\Rightarrow \quad \mu\mathbb{E}[\mathrm{W}_2(q_T, q_*)^2] \le \epsilon \, ,$$

*where $\Delta^2 = \mu\|\lambda_0 - \lambda_*\|^2$.*

*Proof.* The full proof is deferred to Section D.2.1. $\square$

Previously, for Gaussian variational families with a dense covariance (the "full-rank" Gaussian family), in the limit of $\epsilon \to 0$, Kim et al. (2023); Domke et al. (2023) reported an iteration complexity of $\mathrm{O}(d\kappa^2\,\mathrm{tr}(\mu\Sigma_*)\epsilon^{-1})$, which used the canonical reparametrization gradient (Eq. (4)). Compared to this, Price's gradient improves the iteration complexity by a factor of $\kappa\,\mathrm{tr}(\mu\Sigma_*)$. (Note that $d/L \le \mathrm{tr}(\Sigma_*) \le d/\mu$.) This is comparable to the complexity of BBVI with the mean-field Gaussian family (diagonal covariance), which is $\mathrm{O}((\log d)\kappa^2\,\mathrm{tr}(\mu\Sigma_*)\epsilon^{-1})$ (Kim et al., 2025). This suggests

that, with Price's gradient, BBVI on a full-rank Gaussian family can be as fast as using a mean-field Gaussian family and the reparametrization gradient.

An immediate implication of Theorems 3.2 and 3.3 is that the gap between the best known iteration complexity bounds between the two algorithms has now been closed. In addition, Section 3.3 that follows will explain that this resemblance is unsurprising, as the convergence analyses of both algorithms rely on nearly the same properties. Though, since we lack matching lower bounds, we cannot yet claim that the two algorithms behave exactly the same. However, our results do provide evidence towards this fact along with the continuous-time results of Yi & Liu (2023); Hoffman & Ma (2020). This again reinforces the intuition that, for stochastic optimization algorithms, the quality of the gradient estimator has the largest impact on the performance.

### 3.3. Proof Sketch

The overall structure of the proofs for both SPBWGD and SPGD is identical. If we had access to exact gradients instead of stochastic estimates, under Assumption 3.1, $\|\lambda_t - \lambda_*\|_2$ or $\mathrm{W}_2(q_t, q_*)$ would contract exponentially in $t$. When dealing with stochastic gradients, however, the noise in the estimates perturbs the iterates. We thus need to show that the variance of the noise is bounded and the contraction is strong enough such that controlling the step size schedule $\gamma_t$ can neutralize the perturbations.

First, under Assumption 3.1, we can define a Bregman divergence associated with $U$,

$$\mathrm{D}_U(x, y) \triangleq U(x) - U(y) - \langle\nabla U(y), x - y\rangle \, .$$

For both SPBWGD and SPGD, we establish gradient variance bounds involving $\mathrm{D}_U$.

**Lemma 3.4.** *Suppose Assumption 3.1 holds and $q_* = \mathrm{Normal}(\mu_*, \Sigma_*) \in \arg\min_{q\in\mathrm{BW}(\mathbb{R}^d)}\mathcal{F}(q)$. Then, for any $q \in \mathrm{BW}(\mathbb{R}^d)$, and any coupling $\psi \in \Psi(q, q_*)$,*

$$\mathbb{E}_{(X,X_*)\sim\psi,\epsilon\sim\varphi}\left[\|\widehat{\nabla_{\mathrm{BW}}^{\mathrm{bonnet-price}}}\mathcal{E}(q;\epsilon)(X) - \nabla\mathcal{E}(q_*)(X_*)\|_2^2\right]$$
$$\le 10L\kappa\,\mathbb{E}_{(X,X_*)\sim\psi}[\mathrm{D}_U(X, X_*)] + 10dL \, .$$

*Proof.* The proof is deferred to Section D.1.4. $\square$

This is a refinement of Lemma 5.6 by Diao et al. (2023). Specifically, instead of upper-bounding the gradient variance with the squared Wasserstein distance $\mathrm{W}_2(q, q_*)^2$, we bound it with the Bregman divergence $\mathrm{D}_U$. In fact, for the optimal coupling $\psi^* \in \Psi(q, q_*)$, we have

$$\mathbb{E}_{(X,X_*)\sim\psi^*}[\mathrm{D}_U(X, X_*)] \le L\mathrm{W}_2(q, q_*)^2$$

(Eq. (14) in Section C.1.) Therefore, the use of the Bregman divergence avoids paying for an extra factor of $\kappa = L/\mu$. The corresponding bound for parameter-space SPGD has the exact same constants.

**Lemma 3.5.** *Suppose Assumptions 3.1 and 2.2 holds, and $\lambda_* \in \arg\min_{\lambda \in \Lambda} \mathcal{F}(q_\lambda)$. Then, for any $\lambda \in \Lambda$ and any coupling $\psi \in \Psi(q_\lambda, q_{\lambda_*})$,*

$$\mathbb{E}_{\epsilon \sim \varphi}\left[\|\widehat{\nabla_\lambda^{\mathrm{bonnet-price}}}\mathcal{E}(q_\lambda; \epsilon) - \nabla\mathcal{E}(q_{\lambda_*})\|_2^2\right]$$
$$\leq 10L\kappa\,\mathbb{E}_{(X,X_*)\sim\psi}[\mathrm{D}_U(X, X_*)] + 10dL \ .$$

*Proof.* The proof is deferred to Section D.2.2. □

The bounds Lemmas 3.4 and 3.5, however, are not immediately usable for a convergence analysis. The "growth" or "multiplicative noise" term $\mathbb{E}[\mathrm{D}_U(X, X_*)]$ needs to be related to the growth of $\mathcal{E}$. Notice that both Lemmas 3.4 and 3.5 hold for any coupling in $\Psi(q, q_*)$. By specifying the coupling $\psi$, we will invoke properties of the geometry associated with $\psi$ induced by each SPGD and SPBWGD, which will allow us to relate $\mathbb{E}[\mathrm{D}_U(X, X_*)]$ with the appropriate notion of growth of $\mathcal{E}$.

Let's start with SPBWGD. For the optimal coupling $\psi^* \in \Psi(q, q_*)$, we will define the Wasserstein analog of the Bregman divergence

$$\mathrm{D}_\mathcal{E}(q, q_*) \triangleq \mathcal{E}(q) - \mathcal{E}(q_*)$$
$$- \mathbb{E}_{(X,X_*)\sim\psi^*}\langle\nabla\mathcal{E}(q_*)(X_*), X - X_*\rangle \geq 0 \ . \tag{11}$$

The non-negativity of $\mathrm{D}_\mathcal{E}$ follows from the fact that $\mathcal{E}$ is geodesically convex under Assumption 3.1 (Lemma D.1). Then the Bures–Wasserstein geometry yields the following:

**Lemma 3.6.** *For any $p, q \in \mathrm{BW}(\mathbb{R}^d)$, denote the coupling optimal for the squared Euclidean cost between $p$ and $q$ as $\psi_* \in \Psi(p, q)$. Then*

$$\mathbb{E}_{(X,Y)\sim\psi_*}[\mathrm{D}_U(X, Y)] = \mathrm{D}_\mathcal{E}(p, q) \ .$$

*Proof.* The proof is deferred to Section D.1.5. □

For SPGD, first consider some $\lambda, \lambda' \in \Lambda$, where $\lambda = (m, \mathrm{vec}\,C)$ and $\lambda' = (m', \mathrm{vec}\,C')$ and the map

$$M_{q_\lambda \mapsto q_{\lambda'}}^{\mathrm{rep}}(z) \triangleq C'C^{-1}(z - m) + m' \ .$$

This is, in fact, a transport map from $q_\lambda$ to $q_{\lambda'}$. Then, under Assumption 2.2, the identities

$$\|\lambda - \lambda'\|_2^2 = \mathbb{E}_{Z\sim q}\|Z - M_{q_\lambda \mapsto q_{\lambda'}}^{\mathrm{rep}}(Z)\|_2^2 \tag{12}$$

and

$$\langle\nabla_\lambda\mathcal{E}(q_{\lambda'}), \lambda - \lambda'\rangle$$
$$= \mathbb{E}_{Z'\sim q_{\lambda'}}\langle\nabla U(Z'), M_{q_{\lambda'} \mapsto q_\lambda}^{\mathrm{rep}}(Z') - Z'\rangle \tag{13}$$

hold. Also, from the fact that the Wasserstein distance is the cost of the optimal coupling, we have the ordering $\|\lambda - \lambda'\|_2 \geq \mathrm{W}_2(q_\lambda, q_{\lambda'})$. That is, the metric associated with parameter space is a coupling-based distance measure. Now, under Assumptions 2.2 and 3.1, $\lambda \mapsto \mathcal{F}(q_\lambda)$ is $\mu$-strongly convex. Then it is well known that the gradient

flow minimizes $\|\lambda_t - \lambda_*\|_2^2$ exponentially in time. The identity Eq. (12) implies that this flow is also minimizing a coupling distance, and in turn the Wasserstein distance. Back to the proof sketch, Eq. (13) implies the following:

**Lemma 3.7.** *Suppose Assumption 2.2 hold. Then, for any $\lambda, \lambda' \in \Lambda$, denote the coupling induced by the transport map $M_{q_\lambda \mapsto q_{\lambda'}}^{\mathrm{rep}}$ as $\psi^{\mathrm{rep}}$. Then*

$$\mathbb{E}_{(X,X')\sim\psi^{\mathrm{rep}}}[\mathrm{D}_U(X, X')] = \mathrm{D}_{\lambda\mapsto\mathcal{E}(q_\lambda)}(\lambda, \lambda') \ .$$

*Proof.* The proof is deferred to Section D.2.3. □

Therefore, the optimal coupling $\psi^*$ and the coupling associated with Assumption 2.2 $\psi^{\mathrm{rep}}$ respectively retrieve a relationship with the growth of $\mathcal{E}$.

Lastly, we need to show that the contraction is able to counteract the "growth" of the gradient variance. Typically, the contraction of first-order methods follows from the coercivity of the gradient operator. For our proof, however, instead of obtaining a full contraction, we establish a slight generalization of coercivity (previously developed by Gorbunov et al. 2020), that allow us to control the Bregman terms $\mathrm{D}_{\lambda\mapsto\mathcal{E}(q_\lambda)}$ and $\mathrm{D}_\mathcal{E}$. For SPBWGD, our result reads:

**Lemma 3.8.** *Suppose Assumption 3.1 holds. Then, for any $q \in \mathrm{BW}(\mathbb{R}^d)$ and $q_* = \arg\min_{q \in \mathrm{BW}(\mathbb{R}^d)} \mathcal{F}(q)$, where we denote their coupling $\psi_* \in \Psi(q, q_*)$ optimal in terms of squared Euclidean distance,*

$$\mathbb{E}_{(X,Y)\sim\psi_*}\langle\nabla_{\mathrm{BW}}\mathcal{E}(p)(X) - \nabla_{\mathrm{BW}}\mathcal{E}(q)(Y), X - Y\rangle$$
$$\geq \frac{\mu}{2}\,\mathrm{W}_2(q_t, q_*)^2 + \mathrm{D}_\mathcal{E}(q_t, q_*) \ .$$

*Proof.* The proof is deferred to Section D.1.6. □

The corresponding result for parameter space SPGD is:

**Lemma 3.9.** *Suppose Assumptions 3.1 and 2.2 hold. Then, for any $\lambda \in \Lambda$ and $\lambda_* = \arg\min_{\lambda \in \Lambda} \mathcal{F}(q_\lambda)$,*

$$\langle\nabla_{\lambda_t}\mathcal{E}(q_{\lambda_t}) - \nabla_{\lambda_*}\mathcal{E}(q_{\lambda_*}), \lambda_t - \lambda_*\rangle$$
$$\geq \frac{\mu}{2}\|\lambda - \lambda_*\|_2^2 + \mathrm{D}_{\lambda\mapsto\mathcal{E}(q_\lambda)}(\lambda, \lambda_*) \ .$$

*Proof.* The proof is deferred to Section D.2.4. □

The extra "Bregman term" on the right-hand sides directly allows control over the Bregman term in the gradient variance bounds.

## 4. Empirical Analysis

For the empirical evaluation, we will compare the performance of VI with SPGD and SPBWGD with the reparametrization and Price estimators.

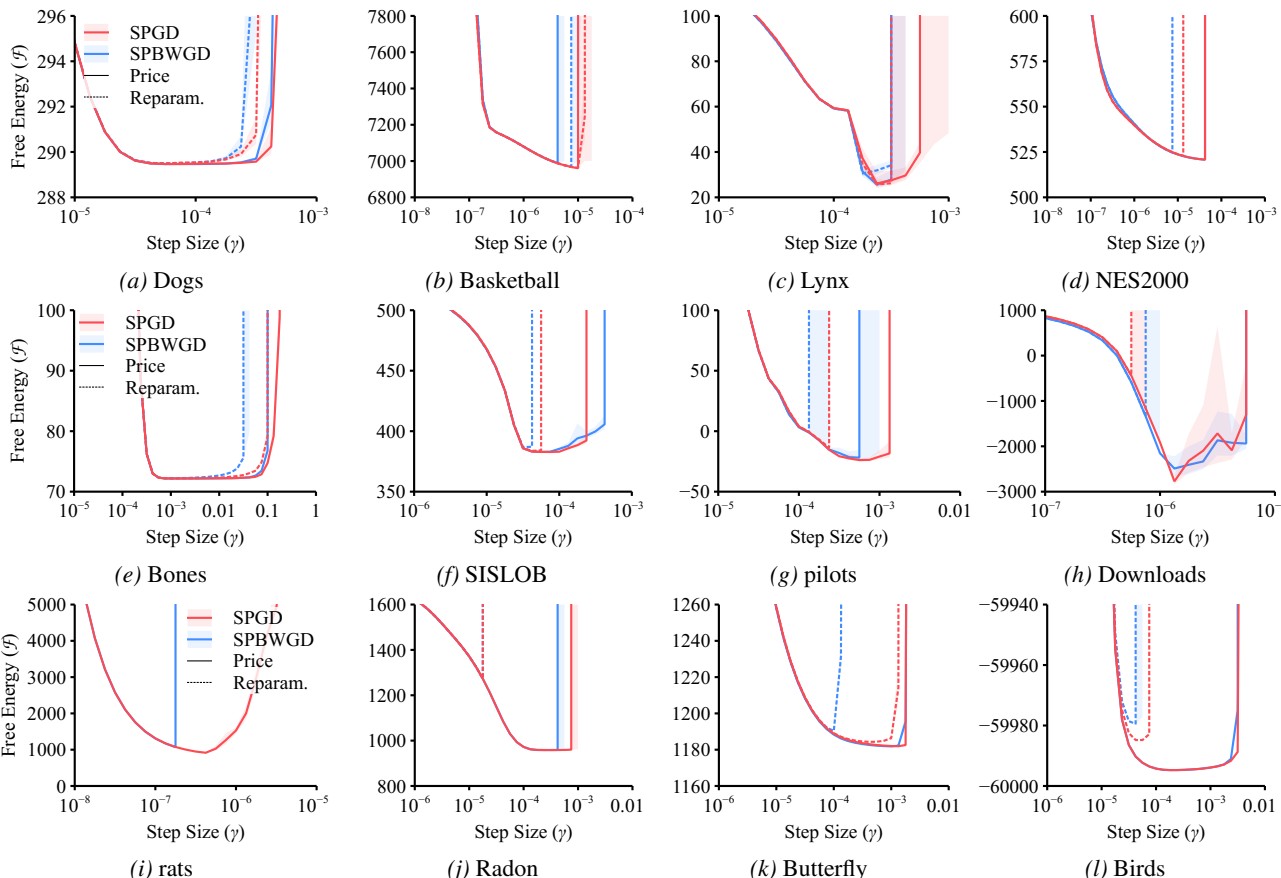

*Figure 1.* **Variational free energy** ($\mathcal{F}$) **at** $T = 4000$ **versus step size** $\gamma$. Additional results for different stopping times $T$ can be found in Section B of the Appendix. Note that the dotted line is missing on Rats because all the corresponding runs diverged. The solid lines are the mean estimated over 32 independent repetitions, while the shaded regions are the 95% bootstrap confidence intervals.

**Setup and Implementation.** For SPGD and SPBWGD, we follow the implementation described in Section 2.3 and Section 2.2, respectively. As mentioned in Section 2.2, the reparametrization gradient $\widehat{\nabla_\Sigma^{\mathrm{rep}}\mathcal{E}}$ is not almost surely symmetric. Therefore, the modified SPBWGD implementation in Section 2.2 is used. All algorithms were implemented in the Julia language (Bezanson et al., 2017) and the AdvancedVI.jl library (v0.6.1), which is part of the Turing probabilistic programming ecosystem (Ge et al., 2018; Fjelde et al., 2025) [2]. The experimental problems were taken from the PosteriorDB (Magnusson et al., 2025) benchmark suite of Stan models (Carpenter et al., 2017), which was made accessible from Julia through the BridgeStan interface (Roualdes et al., 2023). The benchmark problems are described in more detail in Section A. All methods are initialized at $q_0 = \mathrm{Normal}(0_d, 0.34\,\mathrm{I}_d)$. The gradients were estimated using 8 Monte Carlo samples in all cases, while

for evaluation, $\mathcal{F}$ was estimated using $2^{12}$ samples. We run both SPGD and SPBWGD with a fixed step size $\gamma$ and estimate the free energy $\mathcal{F}(q_t)$ of the iterate $q_t$ at each iteration.

**Results.** Part of the results are shown in Fig. 1, while the full set of results can be found in Section B. First, when using Price's gradient (solid line), both SPGD and SPBWGD achieve similar performance, except for Rats, where SPGD remains stable over a wider range of step sizes. On the other hand, when using the reparametrization gradient, both SPGD and SPBWGD perform poorly: they require smaller step sizes. In fact, on Rats, none of the methods using the reparametrization gradient converged for all step sizes between $10^{-8}$ and $10^0$. In addition, on some problems, SPBWGD appears to perform worse than SPGD. For example, on Rats with Price's gradient, SPBWGD requires step sizes orders of magnitude smaller to prevent divergence. A possible explanation is that SPBWGD requires an estimator of $\nabla_\Sigma\mathcal{E}(q)$, whereas SPGD uses an estimator for $\nabla_C\mathcal{E}(q)$. These two are related through the chain rule as $(1/2)C^{-\top}\nabla_\Sigma\mathcal{E}(q) = \nabla_C\mathcal{E}(q)$; the extra scaling of $C^{-\top}$ could make an estimator for $\nabla_\Sigma\mathcal{E}(q)$ noisier than $\nabla_C\mathcal{E}(q)$.

[2]All of the code needed to reproduce the results is available in the following repository: https://github.com/Red-Portal/sgvi_second_order_gradient_estimators.git

## 5. Discussions

In this work, we theoretically analyzed stochastic gradient-based VI algorithms operating in the Euclidean space of parameters (SPGD) and Bures–Wasserstein space (SPBWGD). Our results improve upon the state-of-the-art complexity guarantees for both, closing the gap between them. For SPBWGD, we have technically improved the previous results by Diao et al. (2023). Meanwhile, for SPGD, we have shown that the use of the Price gradient $\widehat{\nabla_C^{\text{price}}}\mathcal{E}$ achieves better theoretical guarantees than those obtained (Domke et al., 2023; Kim et al., 2023) under the reparametrization gradient $\widehat{\nabla_C^{\text{rep}}}\mathcal{E}$. This shows that the previously observed advantage of SPBWGD was due to the choice of gradient estimator rather than the geometry.

However, this doesn't completely rule out the possibility that measure-space algorithms can be more effective. NGVI, which uses the Fisher metric, yields a preconditioned update to the location parameter $m_t$ reminiscent of Newton's method (Khan & Rue, 2023). Possibly due to this, empirical evidence suggests that NGVI methods can converge significantly faster than BBVI (Lin et al., 2019). However, our theoretical understanding of NGVI is still limited, where existing analyses assume conjugacy (Wu & Gardner, 2024) or assumptions much stronger than those considered in this work (Sun et al., 2025; Kumar et al., 2025).

An interesting theoretical aspect of Price's gradient is that its variance becomes zero when $\pi$ is Gaussian. (The Hessian is constant.) But this happens only for the scale $C_t$ or covariance $\Sigma_t$ component of the Bonnet–Price gradient, and the location component $m_t$ is still noisy. Fortunately, this can be addressed by applying the control variate of Luu et al. (2025) with the coefficient $c_k$ set as $c_k = 1$, which enables the "interpolation condition" (Vaswani et al., 2019) (zero variance on the optimum of $\mathcal{F}$). For Gaussian targets, this would imply linear convergence (Schmidt & Roux, 2013; Kim et al., 2024; Domke et al., 2023).

On a practical note, it isn't clear if Price's gradient is always better. For instance, at each iteration, SPGD with the reparametrization gradient requires $\Omega(d^2)$ operations (matrix-vector product for computing $\phi_\lambda$). Moving to second-order increases the cost to $\Omega(d^3)$ operations (matrix-matrix product for computing $C\nabla^2 U$). Meanwhile, SPBWGD requires $\Omega(d^3)$ in both cases. Therefore, when $\nabla U$ can be computed in $O(d^2)$ time, as $d \to \infty$, BBVI with SPGD and the reparametrization gradient could be more efficient ($\Theta(d^2)$ versus $\Theta(d^3)$) depending on the conditioning $\kappa$. In addition, WVI requires numerically sensitive operations, such as matrix square roots and Cholesky decompositions, which makes it less robust.

## Acknowledgements

The authors sincerely thank Kaiwen Wu for bringing the second-order gradient estimators to our attention, Jason Altschuler for suggesting that we look into Bures–Wasserstein variational inference methods, Andre Wibisono for providing valuable comments, and the anonymous reviewers for constructive comments.

Y.-A. Ma was supported by the NSF award CCF-2112665 (TILOS) and partly by the CDC-RFA-FT-23-0069 from the CDC's Center for Forecasting and Outbreak Analytics; T. Campbell was supported by an NSERC Discovery Grant;

## Impact Statement

This work studies the theoretical properties of variational inference, which is a collection of algorithms for approximating distributions. Therefore, our work is not expected to have direct societal consequences other than those of downstream applications of variational inference.

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

# A. Benchmark Problems

*Table 2.* Overview of Benchmark Problems

| Name | Description | $d$ | Reference |
|------|-------------|-----|-----------|
| Dogs | Logistic regression model of the traumatic learning behavior of dogs. The dataset is from the Solomon-Wynne experiment. (model: dogs; dataset: dogs) | 3 | Gelman & Hill (2021) Solomon & Wynne (1953) |
| Basketball | Hidden Markov model of NBA basketball player SportVU tracking data during a drive event. (model: hmm_drive_1; dataset: bball_drive_event_1) | 6 | Ali (2019) |
| Lynx | Lotka-Volterra model of a lynx-hare population. The dataset is the number of pelts collected by the Hudson's Bay Company in the years 1910–1920. (model: lotka_volterra; dataset: hudson_lynx_hare) | 8 | Carpenter (2018) Hewitt (1921) |
| NES2000 | Linear model of political party identification. The data set is from the 2000 National Election Study. (model: nes2000; dataset: nes) | 10 | Gelman & Hill (2021) |
| Bones | Latent trait model for multiple ordered categorical responses for quantifying skeletal maturity from radiograph maturity ratings with missing entries. (model: bones_model; dataset: bones_data) | 13 | Spiegelhalter et al. (1996) |
| SISLOB | Loss model of insurance claims. The model is the single line-of-business, single insurer (SISLOB) variant, where the dataset is the "ppauto" line of business, part of the "Schedule P loss data" provided by the Casualty Actuarial Society. (model: losscurve_sislob; dataset: loss_curves) | 15 | Cooney (2017) |
| Pilots | Linear mixed effects model with varying intercepts for estimating the psychological effect of pilots when performing flight simulations on various airports. (model: pilots; dataset: pilots) | 18 | Gelman & Hill (2021) |
| Downloads | Prophet time series model applied to the download count of rstan over time. The model is an additive combination of (i) a trend model, (ii) a model of seasonality, and (iii) a model for events such as holidays. (model: prophet; dataset: rstan_downloads) | 62 | Taylor & Letham 2018 Bales et al. 2019 |
| Rats | Linear mixed effects model with varying slopes and intercepts for modeling the weight of young rats over five weeks. (model: rats_model; data: rats_data) | 65 | Spiegelhalter et al. (1996) Gelfand et al. (1990) |
| Radon | Multilevel mixed effects model with log-normal likelihood and varying intercepts for modeling the radon level measured in U.S. households. We use the Minnesota state subset. (model: radon_hierarchical_intercept_centered; dataset: radon_mn) | 90 | Magnusson et al. 2025 Gelman et al. 2014 |
| Butterfly | Multispecies occupancy model with correlation between sites. The dataset contains counts of butterflies from twenty grassland sites in south-central Sweden (model: butterfly; dataset: multi_occupancy) | 106 | Dorazio et al. (2006) |
| Birds | Mixed effects model with a Poisson likelihood and varying intercepts for modeling the occupancy of the Coal tit (*Parus ater*) bird species during the breeding season in Switzerland. (model: GLMM1_model; dataset: GLMM_data) | 237 | Kéry & Schaub (2012) |

The benchmark problems used in the experiments of Section 4 and Section B are organized in Table 2.

# B. Additional Experimental Results

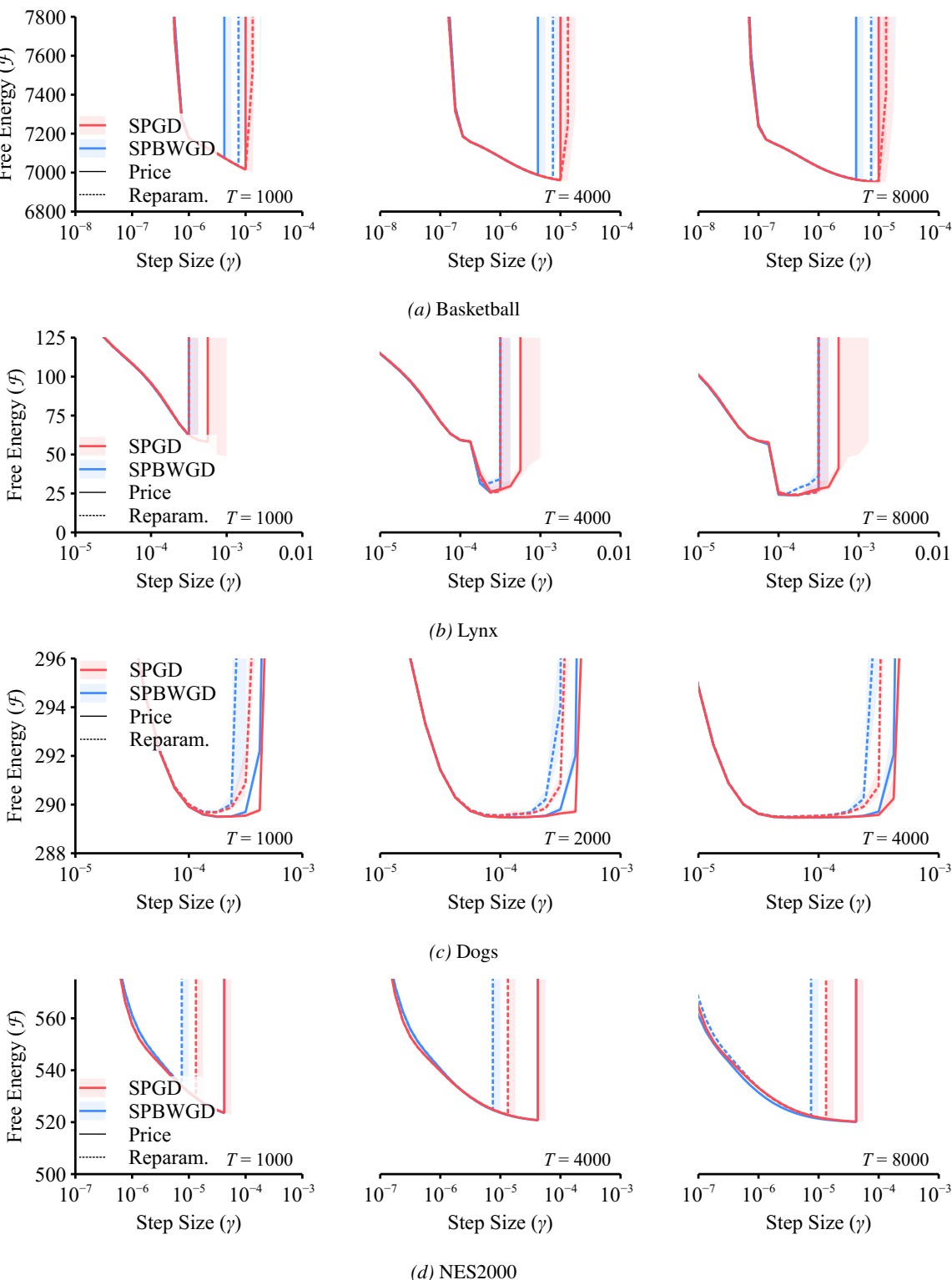

*Figure 2.* **Variational free energy ($\mathcal{F}$) of the last iterate $q_T$ versus step size $\gamma$.** The solid lines are the mean estimated over 32 independent repetitions, while the shaded regions are the corresponding 95% bootstrap confidence intervals.

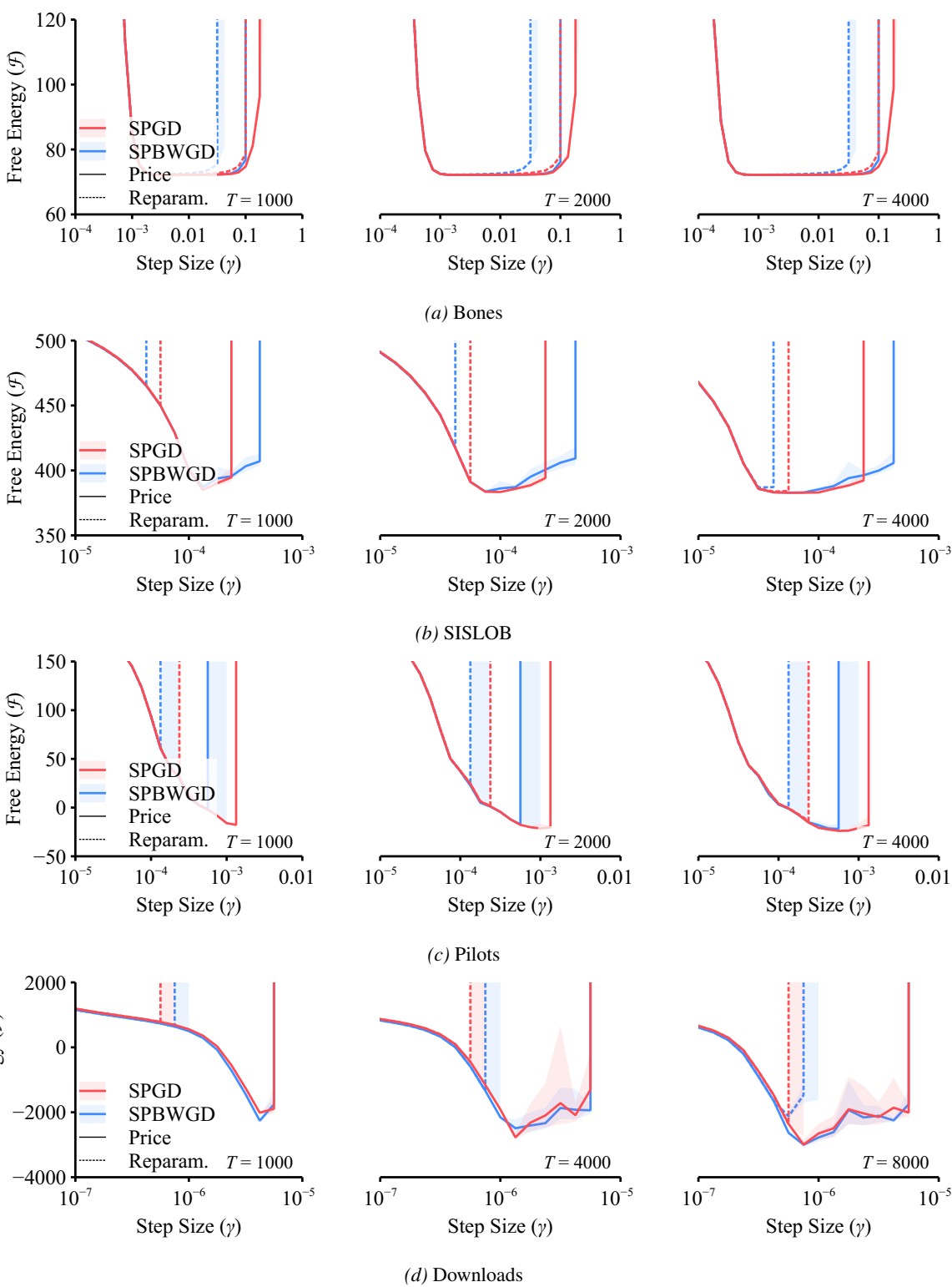

*(a)* Bones

*(b)* SISLOB

*(c)* Pilots

*(d)* Downloads

*Figure 3.* **Variational free energy ($\mathcal{F}$) of the last iterate $q_T$ versus step size $\gamma$.** The solid lines are the mean estimated over 32 independent repetitions, while the shaded regions are the corresponding 95% bootstrap confidence intervals.

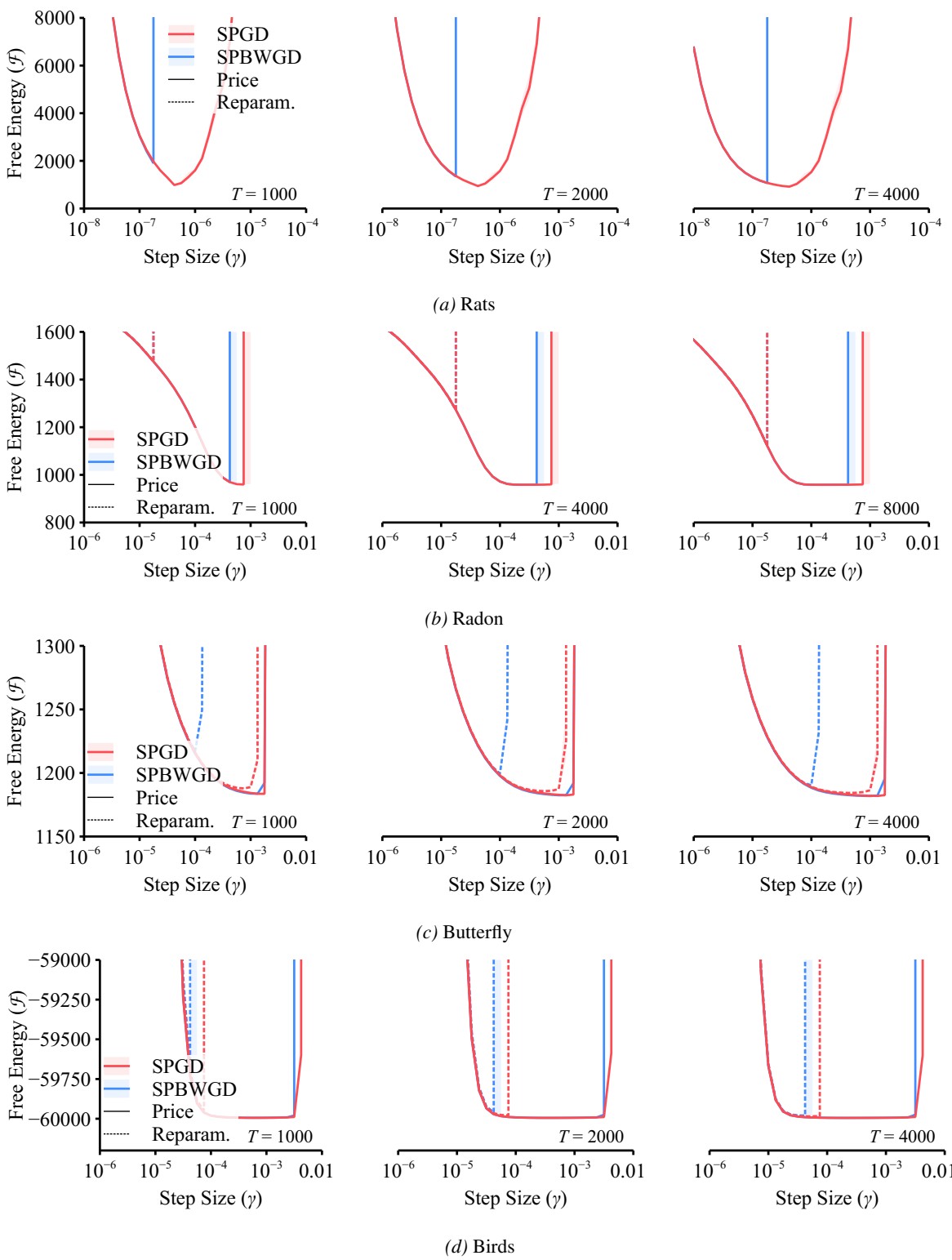

*Figure 4.* **(continued) Variational free energy ($\mathcal{F}$) of the last iterate** $q_T$ **versus step size** $\gamma$**.** In the case of the Rats problem, methods using first-order estimators didn't converge for any step size between $10^{-8}$ and $10^{0}$, which is why the dotted lines are not visible. The solid lines are the mean estimated over 32 independent repetitions, while the shaded regions are the corresponding 95% bootstrap confidence intervals.

# C. Auxiliary Results

## C.1. Properties of the Bregman Divergence

Under Assumption 3.1, it is well known (Garrigos & Gower, 2023, Lemma 2.14 & 2.25) that the Bregman divergence is related to the Euclidean distance via the inequality

$$\frac{\mu}{2}\|x - y\|_2^2 \quad \leq \quad \mathrm{D}_U(x, y) \quad \leq \quad \frac{L}{2}\|x - y\|_2^2 \ . \quad (14)$$

When $x$ and $y$ are replaced with two coupled random variables $X \sim p$ and $Y \sim q$, where $(X, Y) \sim \psi^*$ and $\psi^* \in \Psi(p, q)$ is the optimal coupling between $p$ and $q$, we can also notice that $\mathbb{E}_{(X,Y)\sim\psi^*}[\mathrm{D}_U(X, Y)]$ is related to the Wasserstein distance as

$$\frac{\mu}{2}\mathrm{W}_2(p, q)^2 \leq \mathbb{E}_{(X,Y)\sim\psi^*}[\mathrm{D}_U(X, Y)] \leq \frac{L}{2}\mathrm{W}_2(p, q)^2 \ . \quad (15)$$

In addition, under Assumption 3.1, it is well known (Garrigos & Gower, 2023, Lemma 2.29) that, for any $x, y \in \mathbb{R}^d$,

$$\|\nabla U(x) - \nabla U(y)\|_2^2 \quad \leq \quad 2L \, \mathrm{D}_U(x, y) \ . \quad (16)$$

## C.2. Miscellaneous Results

The following is the multivariate version of Stein's identity (Stein, 1981).

**Proposition C.1** (Stein's Identity). *Suppose $X \sim \mathrm{Normal}(\mu, \Sigma)$. Then, for any differentiable function $g : \mathbb{R}^d \to \mathbb{R}^d$, where, for all $i = 1, \ldots, d$, $\partial g / \partial x_i$ is continuous almost everywhere and $\mathbb{E}|\partial g / \partial x_i| < +\infty$, we have*

$$\mathbb{E}\big[(X - \mu)g(X)^\top\big] = \Sigma \, \mathbb{E}\nabla g(X) \ .$$

*Proof.* This is a direct consequence of integration by parts. See the full proof by Liu (1994, Lemma 1). □

The next proposition is the multivariate analog of the Poincaré inequality specialized to Gaussians.

**Proposition C.2** (Gaussian Poincaré Inequality). *Suppose $X \sim \mathrm{Normal}(\mu, \Sigma)$ and $g : \mathbb{R}^d \to \mathbb{R}^d$ is some continuously differentiable function satisfying $\mathbb{E}\|\nabla g(X)\|_{\mathrm{F}}^2 < +\infty$, where $\nabla g(x) \in \mathbb{R}^{d\times d}$ is its Jacobian. Then*

$$\mathbb{E} \|g(X) - \mathbb{E}[g(X)]\|^2 \leq \mathbb{E} \, \mathrm{tr} \, \nabla g(X)\Sigma\nabla g(X)^\top \ .$$

Formally, we say a probability measure $\nu$ satisfies a $C_{\mathrm{PI}}$-Poincaré inequality if there exists some $C_{\mathrm{PI}} \in (0, +\infty)$ such that, for any Lipschitz smooth function $f : \mathbb{R}^d \to \mathbb{R}$, the following inequality holds:

$$\mathrm{Var}_\nu(f) \leq \frac{1}{C_{\mathrm{PI}}}\mathbb{E}_\nu\|\nabla f\|^2 \ . \quad (17)$$

Then the result is a basic consequence of Eq. (17).

*Proof of Proposition C.2.* Denote $g$ by $(g_1, ..., g_d)^\top$ and the coloring transform $M(\epsilon) \triangleq \Sigma^{1/2}\epsilon + m$ such that $X = M(\epsilon)$ where $\epsilon \sim \mathrm{Normal}(0_d, \mathrm{I}_d)$. Through $M$, we can convert the expectation to be over a standard Gaussian, which satisfies Eq. (17) with $C_{\mathrm{PI}} = 1$ (Beckner, 1989).

$$\begin{aligned}
&\mathbb{E} \|g(X) - \mathbb{E}[g(X)]\|^2 \\
&= \textstyle\sum_{i=1}^d \mathbb{E}(g_i(X) - \mathbb{E}[g_i(X)])^2 \\
&= \textstyle\sum_{i=1}^d \mathbb{E}(g_i(M(\epsilon)) - \mathbb{E}[g_i(M(\epsilon))])^2 \\
&\leq \textstyle\sum_{i=1}^d \mathbb{E}\|\nabla g_i(M(\epsilon))\|^2 \qquad \text{(Eq. (17))}
\end{aligned}$$

By the chain rule,

$$\nabla_\epsilon g(M(\epsilon)) = \nabla g(M(\epsilon))\nabla M(\epsilon) = \nabla g(X)\Sigma^{1/2} \ .$$

Then

$$\begin{aligned}
\mathbb{E} \|g(X) - \mathbb{E}[g(X)]\|^2 &\leq \mathbb{E} \, \mathrm{tr} \, \nabla_\epsilon g(M(\epsilon))\nabla_\epsilon g(M(\epsilon))^\top \\
&= \mathbb{E} \, \mathrm{tr} \, \nabla g(X)\Sigma\nabla g(X)^\top \ .
\end{aligned}$$

□

Diao et al. (2023) also demonstrate that Proposition C.2 can be seen as a consequence of the Brescamp-Lieb inequality (Brascamp & Lieb, 2002), which generalizes Proposition C.2 to all strictly log-concave $\mathrm{Law}(X)$.

In addition, we prove an upper bound on the trace of a product of positive definite matrices.

**Proposition C.3.** *Suppose $A, B \in \mathbb{S}^d_{\succ 0}$ with $\|A\|_2 < +\infty$. Then*

$$\operatorname{tr} ABA \quad \leq \quad \|A\|_2 \operatorname{tr} AB .$$

*Proof.* Since $A$ is positive definite, there exists a collection of eigenvalues $(\lambda_i)_{i=1}^d$ and eigenvectors $(v_i)_{i=1}^d$ such that

$$A = \sum_{i=1}^d \lambda_i v_i v_i^\top . \tag{18}$$

Given this representation,

$$\operatorname{tr} ABA = \operatorname{tr} A^2 B \qquad \text{(Cyclic property of } \operatorname{tr})$$

$$= \operatorname{tr} \sum_{i=1}^d \lambda_i^2 v_i v_i^\top B \quad \text{(Eq. (18))}$$

$$= \sum_{i=1}^d \lambda_i^2 \operatorname{tr} v_i v_i^\top B \quad \text{(Linearity of } \operatorname{tr})$$

$$= \sum_{i=1}^d \lambda_i^2 \operatorname{tr} v_i^\top B v_i \quad \text{(Cyclic property of } \operatorname{tr})$$

Now, from the fact that $A$ and $B$ are positive definite and that $\|A\|_2 < +\infty$, for all $i = 1, \ldots, d$, we have

$$\lambda_i^2 \leq \big( \max_{j=1,\ldots,d} \lambda_j \big) \lambda_i \quad \text{and} \quad v_i^\top B v_i > 0 .$$

Therefore,

$$\operatorname{tr} ABA \leq \big( \max_{i=1,\ldots,d} \lambda_i \big) \sum_{i=1}^d \lambda_i \operatorname{tr} v_i^\top B v_i$$

$$= \big( \max_{i=1,\ldots,d} \lambda_i \big) \sum_{i=1}^d \operatorname{tr} \lambda_i v_i v_i^\top B$$

$$= \big( \max_{i=1,\ldots,d} \lambda_i \big) \sum_{i=1}^d \operatorname{tr} AB \qquad \text{(Eq. (18))}$$

$$= \|A\|_2 \operatorname{tr} AB . \qquad (A \succ 0)$$

$\square$

## C.3. Stationarity Condition

The following proposition characterizes the properties of the minimizer of $\mathcal{F}$, which also corresponds to the stationary point of the algorithms considered in this work.

**Proposition C.4** (Stationary condition)**.** *Suppose $q_* = \operatorname{Normal}(\mu_*, \Sigma_*) \in \arg\min_{q \in \mathrm{BW}(\mathbb{R}^d)} \mathcal{F}(q)$. Then*

$$\mathbb{E}_{q_*} \nabla U = 0, \quad \mathbb{E}_{q_*} \nabla^2 U = \Sigma_*^{-1}.$$

*Proof.* The Bures-Wasserstein gradient of $\mathcal{F}$ can be derived (Lambert et al., 2022, Appendix C.1) as, for each $q = \operatorname{Normal}(m, \Sigma) \in \mathrm{BW}(\mathbb{R}^d)$,

$$\nabla_{\mathrm{BW}} \mathcal{F}(q) = x \mapsto \mathbb{E}_q \nabla U + \big( \mathbb{E}_q \nabla^2 U - \Sigma^{-1} \big)(x - m)$$

Solving for the first-order stationarity condition immediately yields the result. $\square$

## C.4. Bound on the Covariance-Weighted Hessian Norm

A crucial step in our analysis of the variance of the stochastic gradients is to bound the quantity $\mathbb{E}_{Z\sim q}\mathrm{tr}\,\nabla^2 U(Z)\,\Sigma\,\nabla^2 U(Z)$. Specifically, we need to relate it to some notion of "growth" of $\mathcal{E}$.

The main technical contributions of our new results come from the fact that we relate $\mathbb{E}_{Z\sim q}\mathrm{tr}\,\nabla^2 U(Z)\,\Sigma\,\nabla^2 U(Z)$ with the Bregman divergence of $U$, $\mathrm{D}_U$.

**Lemma C.5.** *Suppose Assumption 3.1 holds and denote $q_* = \arg\min_{q\in\mathrm{BW}(\mathbb{R}^d)}\mathcal{F}(q)$. Then, for any $q = \mathrm{Normal}(m,\Sigma)$ and any coupling $\psi\in\Psi(q,q_*)$,*

$$\mathbb{E}_{Z\sim q}\mathrm{tr}\,\nabla^2 U(Z)\,\Sigma\,\nabla^2 U(Z)$$
$$\leq L\big(2\sqrt{\kappa}+\kappa\big)\mathbb{E}_{(X,X_*)\sim\psi}[\mathrm{D}_U(X,X_*)]+3dL\,.$$

This contrasts with the previous analysis by Diao et al. (2023, Lemma 5.6), who bounded $\mathbb{E}_{Z\sim q}\mathrm{tr}\,\nabla^2 U(Z)\,\Sigma\,\nabla^2 U(Z)$ by the Wasserstein distance $\mathrm{W}_2(q_t,q_*)^2$. From Eq. (15), it is apparent that the Bregman divergence results in a tighter bound; it avoids paying for an additional factor of $L$, which is the source of the $\kappa$-factor improvement in Theorem 3.2.

The first steps in the proof closely mirror the steps of Diao et al. (2023) up to the error decomposition:

$$\mathbb{E}_{Z\sim q}\mathrm{tr}\,\nabla^2 U(Z)\,\Sigma\,\nabla^2 U(Z)\,.$$
Applying Proposition C.3 with $\|\nabla^2 U\|_2\leq L$,
$$\leq L\,\mathrm{tr}\,\mathbb{E}_q\big[\nabla^2 U\big]\Sigma$$
and by Stein's identity (Proposition C.1),
$$= L\,\mathbb{E}_{X\sim q}\langle\nabla U(X),X-m\rangle$$
$$= L\,\underbrace{\mathbb{E}\langle\nabla U(X_*),X_*-m_*\rangle}_{\triangleq E_1}$$
$$+ L\,\underbrace{\mathbb{E}\langle\nabla U(X)-\nabla U(X_*),(X-m)-(X_*-m_*)\rangle}_{\triangleq E_2}$$
$$+ L\,\underbrace{\mathbb{E}\langle\nabla U(X_*),(X-m)-(X_*-m_*)\rangle}_{\triangleq E_3}$$
$$+ L\,\underbrace{\mathbb{E}\langle\nabla U(X)-\nabla U(X_*),X_*-m_*\rangle}_{\triangleq E_4}\,. \qquad (19)$$

Each error term $E_1, E_2, E_3, E_4$, however, will be bounded by the Bregman divergence instead of the squared Euclidean. For this, we will repeatedly use the following result:

**Lemma C.6.** *Suppose Assumption 3.1 holds. Then, for any two random variables $X, X_*$ on $\mathbb{R}^d$ satisfying $\mathbb{E}X = m$ and $\mathbb{E}X_* = m_*$, where $\|m\|_2 < +\infty$ and $\|m_*\|_2 < +\infty$, we have*

$$\mathbb{E}\|(X-m)-(X_*-m_*)\|_2^2\leq\frac{2}{\mu}\mathbb{E}[\mathrm{D}_U(Z,Z_*)]\,.$$

*Proof.*

$$\mathbb{E}\|(X-m)-(X_*-m_*)\|_2^2$$
$$\leq\mathbb{E}\|X-X_*\|_2^2 \qquad (\mathrm{tr}\,\mathrm{Var}(X)\leq\mathbb{E}\|X\|_2^2)$$
$$\leq\frac{2}{\mu}\mathbb{E}[\mathrm{D}_U(Z,Z_*)]\,. \qquad (\text{Eq. (14)})$$

$\square$

Let's proceed to the proof of Lemma C.5.

*Proof of Lemma C.5.* From Eq. (19), we have

$$\mathbb{E}_{Z\sim q}\mathrm{tr}\,\nabla^2 U(Z)\,\Sigma\,\nabla^2 U(Z)\leq L(E_1+E_2+E_3+E_4)\,.$$

The error terms can be bounded as follows.

For $E_1$, Stein's identity (Proposition C.1) states that

$$E_1 = \mathbb{E}_{X_*\sim q_*}\langle\nabla U(X_*),X_*-m_*\rangle$$
$$= \mathrm{tr}\big(\mathbb{E}_{q_*}\big[\nabla^2 U\big]\Sigma_*\big)$$

and the stationary condition (Proposition C.4) yields

$$= \mathrm{tr}\big(\Sigma_*{}^{-1}\Sigma_*\big)$$
$$= d\,.$$

For $E_2$, Young's inequality yields

$$E_2 = \mathbb{E}_{(X,X_*)\sim\psi}\langle\nabla U(X)-\nabla U(X_*),$$
$$(X-m)-(X_*-m_*)\rangle$$
$$\leq\frac{1}{2\sqrt{L\mu}}\mathbb{E}_{(X,X_*)\sim\psi}\|\nabla U(X)-\nabla U(X_*)\|_2^2$$
$$+\frac{\sqrt{L\mu}}{2}\mathbb{E}_{(X,X_*)\sim\psi}\|(X-m)-(X_*-m_*)\|_2^2\,.$$

Then apply Eq. (16)

$$\leq\sqrt{\kappa}\,\mathbb{E}_{(X,X_*)\sim\psi}[\mathrm{D}_U(X,X_*)]$$
$$+\frac{\sqrt{L\mu}}{2}\mathbb{E}_{(X,X_*)\sim\psi}\|(X-m)-(X_*-m_*)\|_2^2$$

and Lemma C.6 such that

$$\leq\sqrt{\kappa}\,\mathbb{E}_{(X,X_*)\sim\psi}[\mathrm{D}_U(X,X_*)]$$
$$+\sqrt{\kappa}\,\mathbb{E}_{(X,X_*)\sim\psi}[\mathrm{D}_U(X,X_*)]$$
$$\leq 2\sqrt{\kappa}\,\mathbb{E}_{(X,X_*)\sim\psi}[\mathrm{D}_U(X,X_*)]\,.$$

For $E_3$, we begin by applying Young's inequality as

$$E_3 = \mathbb{E}_{(X,X_*)\sim\psi}\langle\nabla U(X_*),(X-m)-(X_*-m_*)\rangle$$

$$\leq \frac{1}{L}\,\mathbb{E}_{(X,X_*)\sim\psi}\|\nabla U(X_*)\|_2^2$$
$$+\frac{L}{4}\,\mathbb{E}_{(X,X_*)\sim\psi}\|(X-m)-(X_*-m_*)\|_2^2\,.$$

Applying Lemma C.6 yields

$$\leq \frac{1}{L}\,\mathbb{E}_{(X,X_*)\sim\psi}\|\nabla U(X_*)\|_2^2+\frac{\kappa}{2}\,\mathbb{E}[\mathrm{D}_U(X,X_*)]\,.$$

The stationary condition $\mathbb{E}_{q_*}[\nabla U]=0$ yields

$$=\frac{1}{L}\,\mathbb{E}_{(X,X_*)\sim\psi}\|\nabla U(X_*)-\mathbb{E}_{q_*}[\nabla U]\|_2^2$$
$$+\frac{\kappa}{2}\,\mathbb{E}[\mathrm{D}_U(X,X_*)]\,.$$

Then, by the Gaussian Poincaré inequality (Proposition C.2),

$$\leq \frac{1}{L}\,\mathbb{E}_{q_*}\mathrm{tr}\left((\nabla^2 U)^2\Sigma_*\right)+\frac{\kappa}{2}\,\mathbb{E}[\mathrm{D}_U(X,X_*)]\,.$$

Since $U$ is $L$-smooth under Assumption 3.1,

$$\leq \mathrm{tr}\left(\mathbb{E}_{q_*}[\nabla^2 U]\Sigma_*\right)+\frac{\kappa}{2}\,\mathbb{E}_{(X,X_*)\sim\psi}[\mathrm{D}_U(X,X_*)]$$

and applying the stationary condition (Proposition C.4),

$$=\mathrm{tr}\left(\Sigma_*^{-1}\Sigma_*\right)+\frac{\kappa}{2}\,\mathbb{E}_{(X,X_*)\sim\psi}[\mathrm{D}_U(X,X_*)]$$
$$=d+\frac{\kappa}{2}\,\mathbb{E}_{(X,X_*)\sim\psi}[\mathrm{D}_U(X,_*)]\,.$$

For $E_4$, we again begin with Young's inequality.

$$E_4=\mathbb{E}_{(X,X_*)\sim\psi}\langle\nabla U(X)-\nabla U(X_*),X_*-m_*\rangle$$
$$\leq \frac{1}{4\mu}\,\mathbb{E}_{(X,X_*)\sim\psi}\|\nabla U(X)-\nabla U(X_*)\|_2^2$$
$$+\mu\,\mathbb{E}_{X_*\sim q_*}\|X_*-m_*\|_2^2\,.$$

From Eq. (16),

$$\leq \frac{\kappa}{2}\,\mathbb{E}_{(X,X_*)\sim\psi}[\mathrm{D}_U(X,X_*)]$$
$$+\mu\,\mathbb{E}\|X_*-m_*\|_2^2$$
$$=\frac{\kappa}{2}\,\mathbb{E}_{(X,X_*)\sim\psi}[\mathrm{D}_U(X,X_*)]+\mu\,\mathrm{tr}(\Sigma_*)$$

and the stationary condition (Proposition C.4),

$$=\frac{\kappa}{2}\,\mathbb{E}_{(X,X_*)\sim\psi}[\mathrm{D}_U(X,X_*)]+\mu\,\mathrm{tr}\left((\mathbb{E}_{q_*}\nabla^2 U)^{-1}\right)\,.$$

Finally, from the fact that $(\mathbb{E}_{q_*}\nabla^2 U)^{-1}\preceq(1/\mu)\mathrm{I}_d$,

$$\leq \frac{\kappa}{2}\,\mathbb{E}_{(X,X_*)\sim\psi}[\mathrm{D}_U(X,X_*)]+d\,.$$

Combining all the results,

$$\mathbb{E}_{Z\sim q}\mathrm{tr}\,\nabla^2 U(Z)\,\Sigma\,\nabla^2 U(Z)$$
$$\leq L\{E_1+E_2+E_3+E_4\}$$
$$\leq L\mathbb{E}_{(X,X_*)\sim\psi}\Bigg\{d+2\sqrt{\kappa}\,\mathrm{D}_U(X,X_*)$$
$$+\left(d+\frac{\kappa}{2}\,\mathrm{D}_U(X,X_*)\right)+\left(\frac{\kappa}{2}\,\mathrm{D}_U(X,X_*)+d\right)\Bigg\}$$
$$=L\left(2\sqrt{\kappa}+\kappa\right)\mathbb{E}_{(X,X_*)\sim\psi}[\mathrm{D}_U(X,X_*)]+3dL\,.$$

$\square$

## C.5. Variance Bound for Bonnet's Gradient Estimator

### C.5.1. LEMMA C.7

The following result will be used to bound the variance of Bonnet's gradient estimator (Bonnet, 1964) for the location component $m$ of any $q=\mathrm{Normal}(m,\Sigma)\in\mathrm{BW}(\mathbb{R}^d)$

$$\widehat{\nabla_m^{\mathrm{bonnet}}\mathcal{E}}(q;\epsilon)\triangleq\nabla U(Z),\quad\text{where}\quad Z\sim q\,,$$

where $\epsilon$ is the randomness needed to sample $Z$ from $q$.

**Lemma C.7.** *Suppose Assumption 3.1 holds and $q_*=\mathrm{Normal}(\mu_*,\Sigma_*)\in\arg\min_{q\in\mathrm{BW}(\mathbb{R}^d)}\mathcal{F}(q)$. Then, for any $q\in\mathrm{BW}(\mathbb{R}^d)$ and any coupling $\psi\in\Psi(q,q_*)$,*

$$\mathbb{E}_{Z\sim q}\|\nabla U(Z)-\mathbb{E}_{q_*}\nabla U\|_2^2$$
$$\leq 4L\,\mathbb{E}_{(X,X_*)\sim\psi}[\mathrm{D}_U(X,X_*)]+2dL\,.$$

This serves a similar purpose as the usual "variance transfer" lemma used to analyze SGD on expected risk minimization-type problems (Garrigos & Gower, 2023, Lemma 8.21). Notice that the result does not specify the coupling $\psi$ and generalizes to all couplings in $\Psi(q,q_*)$.

*Proof of Lemma C.7.* First, decompose the gradient variance using Young's inequality.

$$\mathbb{E}_{Z\sim q}\|\nabla U(Z)-\mathbb{E}_{q_*}\nabla U\|_2^2$$
$$=\mathbb{E}_{(Z,Z_*)\sim\psi}\|\nabla U(Z)-\nabla U(Z_*)+\nabla U(Z_*)-\mathbb{E}_{q_*}\nabla U\|_2^2$$
$$\leq 2\underbrace{\mathbb{E}_{(Z,Z_*)\sim\psi}\|\nabla U(Z)-\nabla U(Z_*)\|}_{\triangleq V_{\mathrm{mult}}}$$
$$+2\underbrace{\mathbb{E}_{Z_*\sim q_*}\|\nabla U(Z_*)-\mathbb{E}_{q_*}\nabla U\|_2^2}_{\triangleq V_{\mathrm{add}}}$$

The multiplicative noise follows from the $L$-smoothness of $U$. By applying Eq. (14),

$$V_{\mathrm{mult}}=\mathbb{E}_{(Z,Z_*)\sim\psi}\|\nabla U(Z)-\nabla U(Z_*)\|_2^2$$
$$\leq 2L\,\mathbb{E}_{(Z,Z_*)\sim\psi}\big(U(Z)-U(Z_*)$$
$$-\langle\nabla U(Z_*),Z_t-Z_*\rangle\big)$$
$$=2L\,\mathbb{E}_{(Z,Z_*)\sim\psi}\mathrm{D}_U(Z,Z_*)\,. \tag{20}$$

The additive noise, on the other hand, follows from the Gaussian Poincaré inequality and the $L$-smoothness of $U$.

$$V_{\mathrm{add}}=\mathbb{E}_{Z_*\sim q_*}\|\nabla U(Z_*)-\mathbb{E}_{q_*}\nabla U\|_2^2$$

Due to the Gaussian Poincaré inequality (Proposition C.2),

$$\leq \mathbb{E}_{Z_*\sim q_*}\mathrm{tr}\left(\nabla^2 U(Z_*)\right)\Sigma_*(\nabla^2 U(Z_*))$$

and applying Proposition C.3 with $\|\nabla^2 U\|_2\leq L$,

$$\leq L\,\mathrm{tr}(\Sigma_*\mathbb{E}_{q_*}[\nabla^2 U])\,.$$

The stationary condition (Proposition C.4) yields

$$=L\,\mathrm{tr}(\Sigma_*\Sigma_*^{-1})$$
$$=dL\,. \tag{21}$$

Combining Eqs. (20) and (21) yields the statement. $\square$

## C.6. Lyapunov Convergence Lemma

In this section, we will provide an auxiliary result that will be used throughout this work for analyzing the convergence of stochastic first-order algorithms on strongly convex objectives.

### C.6.1. PROPOSITION C.8

Historically, convergence guarantees for stochastic first-order methods on strongly convex objectives come in two flavors: results based on a fixed step size or a decreasing step size schedule. Consider some distance metric $d(\cdot, \cdot)$, denote the iterates generated by SGD as $(x_t)_{t=0}^{T}$ and the unique global optimum as $x_*$. For any target accuracy level $\epsilon > 0$, to obtain an $\epsilon$-accurate solution such that $\mathbb{E}\, d(x_T, x_*)^2 \leq \epsilon$, SGD with a *fixed step size* results in an iteration complexity of $O(1/\epsilon \log(\Delta_0/\epsilon))$ (Garrigos & Gower, 2023, Corollary 5.9). Notice that the dependence on $\epsilon$ is $O(1/\epsilon \log 1/\epsilon)$, while the dependence on the initial distance $d(x_0, x_*)$ is logarithmic. On the other hand, a *decreasing step size schedule* (Shamir & Zhang, 2013; Lacoste-Julien et al., 2012; Gower et al., 2019) is able to improve the dependence on $\epsilon$ to $O(1/\epsilon)$. For strongly convex and smooth objectives, Gower et al. (2019) showed that the two-stage schedule in Eq. (10) can achieve an iteration complexity of $O(1/\epsilon + d(x_0, x_*)^2/\sqrt{\epsilon})$. Unfortunately, however, the dependence on the distance $d(x_0, x_*)$ is now polynomial instead of logarithmic.

An improvement was presented by Stich (2019) by relying on a step size schedule that optimizes the dependence on both $\epsilon$ and $d(x_0, x_*)$ simultaneously. However, the step size schedule of Stich requires knowing the maximum number of iterations $T$, which means $T$ must be fixed before executing the optimization run. As such, the schedule does not provide an *any-time* convergence guarantee. Since then, Kim et al. (2025, Proposition 2.9) presented a refined schedule that does not rely on $T$, and therefore provides any-time convergence guarantees. We present this result in a more general form following the Lyapunov style of convergence analysis (Wilson, 2018; Dieuleveut et al., 2023; Bansal & Gupta, 2019).

Consider some dynamical system generating the state variable sequence $(x_t)_{t \geq 0}$, where, for each $t \geq 0$, $x_t \in \mathcal{X}$ controlled by some sequence of step sizes $(\gamma_t)_{t \geq 0}$. In our case, $(x_t)_{t \geq 0}$ will be an iterate sequence generated by some corresponding optimization algorithm. Suppose there exists some Lyapunov function $V : \mathcal{X} \to \mathbb{R}_{\geq 0}$ quantifying the energy of the dynamical system. Denoting $V_t \triangleq V(x_t)$, our interest is the sufficient number of iterations $T$ and the conditions on the step size sequence $(\gamma_t)_{t \geq 0}$ for the system $(x_t)_{t \geq 0}$ to achieve $\epsilon$-Lyapunov stability: $V_T = V(x_T) < \epsilon$. This is given by the following proposition.

**Proposition C.8.** *Consider a sequence of Lyapunov function values $(V_t)_{t=0}^{T}$ associated with some dynamical system controlled by some bounded step size sequence $(\gamma_t)_{t=0}^{T}$, where $\gamma_t \leq \gamma_{\max}$ for some $\gamma_{\max} \in (0, +\infty)$. Suppose there exist some constants $\mu \in (0, +\infty)$ and $b \in [0, +\infty)$ such that, for all $t \geq 0$, the sequence satisfies the Lyapunov condition*

$$V_{t+1} - V_t \quad \leq \quad -\mu\gamma_t V_t + b\gamma_t^2 \,. \tag{22}$$

*Then, if the step size schedule in Eq. (10) is used with some $\gamma_0 \leq \gamma_{\max}$ and the remaining parameters set as*

$$\tau = t_* + \frac{2}{\gamma_0 \mu}$$

$$t_* = \left\lceil \frac{1}{\log 1/\rho} \log\left( \frac{\mu}{\gamma_0 b} V_0 \right) \right\rceil,$$

*for any $\epsilon > 0$, we have*

$$T \geq \max\{B_{\mathrm{var}}, B_{\mathrm{bias}}\} \quad \Rightarrow \quad V_T \leq \epsilon \,,$$

*where $\rho \triangleq 1 - \mu\gamma_0$,*

$$B_{\mathrm{var}} = \frac{4b}{\mu}\frac{1}{\mu\epsilon} + \frac{4\sqrt{b}}{\sqrt{\mu}}\frac{1}{\sqrt{\mu\gamma_0}}$$

$$\times \left\{ \log\left( \frac{\mu V_0}{b\gamma_0} \right) + \mu\gamma_0 + \sqrt{2} \right\} \frac{1}{\sqrt{\mu\epsilon}}$$

$$B_{\mathrm{bias}} = \frac{1}{\mu\gamma_0} \log\left( 2V_0 \frac{1}{\epsilon} \right) \,.$$

### C.6.2. PROOF OF PROPOSITION C.8

Under the two-stage step size schedule in Eq. (10), the sequence exhibits two different regimes of convergence behavior. In the first stage, where $\gamma_t = \gamma_0$, we have the following:

**Lemma C.9.** *Suppose there exists some $\mu \in (0, +\infty)$ and $b \in [0, +\infty)$ such that, for some $t_* \geq 0$ and for all $t \in \{0, \ldots, t_*\}$, Eq. (22) holds and the step size schedule is constant such that $\gamma_t = \gamma_0$. Then*

$$V_{t_*} \quad \leq \quad \rho^{t_*} V_0 + \frac{b}{\mu} \gamma_0 \ .$$

*Proof.* Under the stated assumptions, Eq. (22) implies

$$V_{t+1} \quad \leq \quad \rho V_t + b\gamma_0^2 \ .$$

Unrolling this over $t = 0, \ldots, t_* - 1$,

$$V_T \leq \rho^T V_0 + b\gamma_0^2 \sum_{t=0}^{t_*-1} (1 - \mu\gamma_0)^t$$

$$\leq \rho^{t_*} V_0 + \frac{b}{\mu} \gamma_0 \ ,$$

where the last step follows from the geometric series sum formula. $\qquad\square$

For the second stage, we have a decreasing step size schedule. It is well known in the stochastic optimization literature that the step size schedule in Eq. (10) yields a rate that is equivalent to $O(1/T)$ asymptotically in $T$ for both non-smooth (Lacoste-Julien et al., 2012; Shamir & Zhang, 2013; Stich, 2019) and smooth objectives (Gower et al., 2019).

**Lemma C.10.** *Suppose, for all $t \in \{t_*, \ldots, T-1\}$, there exists some $\mu \in (0, +\infty)$ and $b \in [0, +\infty)$ such that Eq. (22) holds and the step size schedule satisfies, for all $t \in \{t_*, \ldots, T-1\}$, some $\tau > 0$,*

$$\gamma_t = \frac{1}{\mu} \frac{2(t+\tau) + 1}{(t+\tau+1)^2} \ .$$

*Then*

$$V_T \leq \frac{(t_*+\tau)^2}{(T+\tau)^2} V_{t_*} + \frac{4b}{\mu^2} \frac{T - t_*}{(T+\tau)^2} \ .$$

*Proof.* Under the assumptions, Eq. (22) becomes

$$V_{t+1} - V_t \leq -\frac{2(t+\tau) + 1}{(t+\tau+1)^2} V_t + \frac{b}{\mu^2} \frac{(2(t+\tau) + 1)^2}{(t+\tau+1)^4}$$

Multiplying $(t+\tau+1)^2$ to both hand sides,

$$(t+\tau+1)^2 V_{t+1} - (t+\tau+1)^2 V_t$$

$$\leq -(2(t+\tau) + 1)V_t + \frac{b}{\mu^2} \frac{(2(t+\tau) + 1)^2}{(t+\tau+1)^2} \ ,$$

and moving the $V_t$ term on the right-hand side to the left,

$$(t+\tau+1)^2 V_{t+1} - (t+\tau)^2 V_t \leq \frac{b}{\mu^2} \frac{(2(t+\tau) + 1)^2}{(t+\tau+1)^2} \ .$$

Summing up the inequality over $t \in \{t_*, \ldots, T-1\}$ yields

$$(T+\tau)^2 V_T - (t_*+\tau)^2 V_0 \leq \sum_{t=t_*}^{T-1} \frac{b}{\mu^2} \frac{(2(t+\tau) + 1)^2}{(t+\tau+1)^2}$$

$$\leq \sum_{t=t_*}^{T-1} \frac{4b}{\mu^2}$$

$$= \frac{4b}{\mu^2}(T - t_*) \ .$$

Re-organizing yields the inequality in the statement. $\qquad\square$

When $t_* < T$, where both stages of the steps size kick in, we have to combine both Lemma C.9 and Lemma C.10. For Eq. (10), applying Lemma C.9 for $t \in \{0, \ldots, t_* - 1\}$ yields

$$V_{t_*} \leq \rho^{t_*} V_0 + \frac{b}{\mu} \gamma_0 \ .$$

Then, for $t \in \{t_*, \ldots, T-1\}$, since $\tau = 2/(\gamma_0 \mu)$, the schedule is bounded by $\gamma_0$ as

$$\gamma_t = \frac{1}{\mu} \frac{2(t+\tau) + 1}{(t+\tau+1)^2}$$

$$\leq \frac{2}{\mu} \frac{1}{t+\tau+1}$$

$$\leq \frac{2}{\mu} \frac{1}{\tau}$$

$$= \frac{2}{\mu} \frac{\gamma_0 \mu}{2}$$

$$= \gamma_0 \qquad\qquad\qquad (23)$$

$$\leq \gamma_{\max} \ .$$

Therefore, we can apply Lemma C.10, which yields

$$V_T \leq V_{t_*} \frac{(t_*+\tau)^2}{(T+\tau)^2} + \frac{4b}{\mu^2} \frac{T - t_*}{(T+\tau)^2} \ .$$

Combining the two bounds,

$$V_T \leq \left\{ \rho^{t_*} V_0 + \frac{b}{\mu} \gamma_0 \right\} \frac{(t_*+\tau)^2}{(T+\tau)^2} + \frac{4b}{\mu^2} \frac{T - t_*}{(T+\tau)^2} \ . \quad (24)$$

For any fixed $t_*$, this already results in an asymptotic rate of $O(1/T)$. Optimizing the bound with respect to the switching time $t_*$, however, requires a bit of work.

*Proof of Proposition C.8.* For the total number of iterations $T$, we have to separately consider the case where $T \leq t_*$ and $T > t_*$. That is, the cases where the second stage doesn't kick in at all, and when it does.

First, in the case of $t_* \geq T$, we have the implications

$$
\begin{aligned}
T \leq t_* \quad &\Leftrightarrow \quad T \leq \left\lceil \frac{1}{\log 1/\rho} \log\left(\frac{\mu}{\gamma_0 b} V_0\right) \right\rceil \\
&\Rightarrow \quad T - 1 \leq \frac{1}{\log 1/\rho} \log\left(\frac{\mu}{\gamma_0 b} V_0\right) \\
&\Leftrightarrow \quad V_0 \rho^{T-1} \geq \frac{b}{\mu} \gamma_0 .
\end{aligned}
\tag{25}
$$

Furthermore, the second stage never kicks in. Therefore, we can invoke Lemma C.9, which yields

$$
V_T \quad \leq \quad \rho^T V_0 + \frac{b}{\mu} \gamma_0 \quad \leq \quad 2\rho^{T-1} V_0 . \quad \text{(Eq. (25))}
$$

Let's identify the condition on $T$ to ensure $V_T \leq \epsilon$. This follows from

$$
\begin{aligned}
V_T \leq \epsilon \quad &\Leftrightarrow \quad 2\rho^T V_0 \leq \epsilon \\
&\Leftrightarrow \quad T \geq \frac{1}{\log(1/\rho)} \log\left(2V_0 \frac{1}{\epsilon}\right) \\
&\Leftarrow \quad T \geq \frac{1}{1-\rho} \log\left(2V_0 \frac{1}{\epsilon}\right) ,
\end{aligned}
$$

where the last step used the inequality $\log(1/\rho) \geq 1 - \rho$. Therefore, in this regime,

$$
T \geq B_{\text{bias}} \quad \Rightarrow \quad V_T \leq \epsilon .
\tag{26}
$$

Let's turn to $t_* < T$. Notice that

$$
\begin{aligned}
t_* &= \left\lceil \frac{1}{\log(1/\rho)} \log\left(\frac{\mu V_0}{b \gamma_0}\right) \right\rceil \\
&> \frac{1}{\log(1/\rho)} \log\left(\frac{\mu V_0}{b \gamma_0}\right) .
\end{aligned}
$$

Therefore,

$$
V_0 \rho^{t_*} > \frac{b}{\mu} \gamma_0 .
$$

Then Eq. (24) can be developed as

$$
\begin{aligned}
V_T &\leq \left\{\frac{2b}{\mu} \gamma_0\right\} \frac{(t_* + \tau)^2}{(T + \tau)^2} + \frac{4b}{\mu^2} \frac{T - t_*}{(T + \tau)^2} . \\
&\leq \frac{2b}{\mu} \gamma_0 \frac{(t_* + \tau)^2}{T^2} + \frac{4b}{\mu^2} \frac{1}{T} .
\end{aligned}
$$

Substituting our choice of $\tau = 2/(\gamma_0 \mu)$,

$$
= \frac{2b}{\mu} \gamma_0 \left(t_* + \frac{2}{\gamma_0 \mu}\right)^2 \frac{1}{T^2} + \frac{4b}{\mu^2} \frac{1}{T} ,
$$

and applying Young's inequality,

$$
\begin{aligned}
&\leq 4\left(\frac{b\gamma_0}{\mu} t_*^2 + 2\frac{b}{\gamma_0 \mu^3}\right) \frac{1}{T^2} + \frac{4b}{\mu^2} \frac{1}{T} \\
&= \alpha \frac{1}{T^2} + \beta \frac{1}{T} ,
\end{aligned}
$$

where the upper bound is now a quadratic function in $1/T$ with the coefficients

$$
\alpha = \frac{4b\gamma_0}{\mu} t_*^2 + \frac{8b}{\gamma_0 \mu^3} \quad \text{and} \quad \beta = \frac{4b}{\mu^2} .
$$

To ensure the condition $V_T \leq \epsilon$ we must now identify the smallest $T > 0$ that satisfies the inequality

$$
\alpha \frac{1}{T^2} + \beta \frac{1}{T} \quad \leq \quad \epsilon .
$$

By the basic properties of quadratics, the quadratic formula yields the condition

$$
\begin{aligned}
\frac{1}{T} \quad &\leq \quad \frac{-\beta + \sqrt{\beta^2 + 4\alpha\epsilon}}{2\alpha} \\
&\leq \quad \frac{4\alpha\epsilon}{4\alpha\sqrt{\beta^2 + 4\alpha\epsilon}} \quad \text{(Symbol-1, 2022)} \\
&\leq \quad \frac{\epsilon}{\sqrt{\beta^2 + 4\alpha\epsilon}} \\
\Leftarrow \quad T \quad &\geq \quad \beta \frac{1}{\epsilon} + 2\sqrt{\alpha} \frac{1}{\sqrt{\epsilon}} ,
\end{aligned}
$$

where the last step used the inequality $\sqrt{\alpha + \beta} \leq \sqrt{\alpha} + \sqrt{\beta}$. That is, in this regime,

$$
T \geq \frac{4b}{\mu^2} \frac{1}{\epsilon} + 2\left\{\sqrt{\frac{4b\gamma_0}{\mu}} t_* + \sqrt{\frac{8b}{\gamma_0 \mu^3}}\right\} \frac{1}{\sqrt{\epsilon}} \quad \Rightarrow \quad V_T \leq \epsilon
$$

Since

$$
\begin{aligned}
t_* &< \frac{1}{\log(1/\rho)} \log\left(\frac{\mu V_0}{b \gamma_0}\right) + 1 \\
&< \frac{1}{\mu \gamma_0} \log\left(\frac{\mu V_0}{b \gamma_0}\right) + 1 , \quad (\log 1/\rho \geq 1 - \rho)
\end{aligned}
$$

it is sufficient to ensure

$$
\begin{aligned}
T &\geq \frac{4b}{\mu^2} \frac{1}{\epsilon} + \frac{4\sqrt{b}}{\mu} \frac{1}{\sqrt{\mu\gamma_0}} \left\{\log\left(\frac{\mu V_0}{b \gamma_0}\right) + \mu\gamma_0 + \sqrt{2}\right\} \frac{1}{\sqrt{\epsilon}} \\
&= B_{\text{var}} .
\end{aligned}
\tag{27}
$$

The last step is to combine the results from case $t_* \geq T$ and $t_* < T$. That is, to ensure that, for any $\epsilon > 0$, $V_T \leq \epsilon$ holds in all cases, it suffices to ensure the conditions in Eqs. (26) and (27) hold simultaneously. Taking $T$ to be the maximum of both $B_{\text{bias}}$ and $B_{\text{var}}$ is sufficient. □

## C.7. General Convergence Analysis of Proximal Bures–Wasserstein Gradient Descent

We first provide a general convergence result for SPBWGD. In particular, instead of assuming a specific gradient estimator, we will assume only that the estimator is unbiased and satisfies a specific variance bound.

**Assumption C.11.** *Denote the global optimum as $q_* \in \arg\min_{q\in\mathrm{BW}(\mathbb{R}^d)} \mathcal{F}(q)$. There exist some constants $L_\epsilon \in (0, +\infty)$ and $\sigma^2 \in [0, +\infty)$ such that, for all $q \in \mathrm{BW}(\mathbb{R}^d)$, the stochastic estimator of the Bures–Wasserstein gradient is unbiased and satisfies the inequality*

$$\mathbb{E}_{(X,X_*)\sim\psi^*,\epsilon\sim\varphi}\|\widehat{\nabla_{\mathrm{BW}}\mathcal{E}}(q;\epsilon)(X) - \nabla_{\mathrm{BW}}\mathcal{E}(q_*)(X_*)\|_2^2$$
$$\leq 4L_\epsilon \mathrm{D}_\mathcal{E}(q, q_*) + 2\sigma^2 \,,$$

*where $\psi^* \in \Psi(q, q_*)$ is the optimal coupling between $q$ and $q_*$.*

Here, the term $L_\epsilon \mathrm{D}_\mathcal{E}(q, q_*)$ represents the "multiplicative noise," while $\sigma^2$ represents the "additive noise." Typically, the $L_\epsilon$ factor restricts the largest step size we can use, essentially playing the same role as the Lipschitz smoothness constant $L$. In fact, $L_\epsilon \geq L$ always holds by Jensen's inequality. On the other hand, $\sigma^2$ is the factor dominating the asymptotic complexity of the algorithm.

Assumption C.11 is closely related to the "convex expected smoothness" condition used for the analysis of SPGD (Gorbunov et al., 2020, Assumption 4.1). (See also Assumption 3.2 of Khaled et al. 2023 and Eq. (65) of Garrigos & Gower 2023.) Furthermore, the special case of $\sigma^2 = 0$ is historically referred to as the "interpolation condition" in the optimization literature (Vaswani et al., 2019). When $\sigma^2 = 0$, it is generally expected that the iteration complexity of the stochastic algorithm improves dramatically (Schmidt & Roux, 2013; Vaswani et al., 2019). We will see that our analysis covers this special case.

### C.7.1. PROPOSITION C.12

Given a gradient estimator satisfying Assumption C.11 and a potential satisfying Assumption 3.1, we have the following iteration complexity guarantee.

**Proposition C.12.** *Suppose Assumption 3.1 holds and the chosen stochastic estimator of the Bures–Wasserstein gradient satisfies Assumption C.11. Then, for any $q_0 \in \mathrm{BW}(\mathbb{R}^d)$, running stochastic Bures–Wasserstein proximal gradient descent with the step size schedule in Eq. (10) with*

$$\gamma_0 = \frac{1}{4L_\epsilon}$$
$$\tau = \frac{2}{\mu\gamma_0}$$
$$t_* = \left\lceil \frac{1}{\log(1/(1-\mu\gamma_0))} \log\left(\frac{1}{2\gamma_0\sigma^2}\Delta_0^2\right) \right\rceil \,,$$

*where we denote $\Delta^2 = \mu\mathrm{W}_2(q_0, q_*)^2$, guarantees that*

$$T \geq \max\{B_{\mathrm{var}}, B_{\mathrm{bias}}\} \quad \Rightarrow \quad \mu\mathbb{E}[\mathrm{W}_2(q_T, q_*)^2] \leq \epsilon \,,$$

*where*

$$B_{\mathrm{var}} = \frac{8\sigma^2}{\mu}\frac{1}{\epsilon} + 2\frac{\sqrt{2\sigma^2}}{\sqrt{\mu}}\frac{\sqrt{L_\epsilon}}{\sqrt{\mu}}$$
$$\times \left\{\log\left(\frac{2L_\epsilon}{\sigma^2}\Delta^2\right) + \frac{\mu}{4L_\epsilon} + \sqrt{2}\right\}\frac{1}{\sqrt{\epsilon}}$$
$$B_{\mathrm{bias}} = \frac{4L_\epsilon}{\mu}\log\left(2\Delta^2\frac{1}{\epsilon}\right) \,.$$

This result is general in the sense that, to analyze the behavior of any gradient estimator, one only needs to establish Assumption C.11 and substitute the corresponding constants into Proposition C.12. Indeed, we even retrieve "linear convergence" ($\log 1/\epsilon$ complexity) under the interpolation condition ($\sigma^2 = 0$).

**Corollary C.13** (Linear Convergence under Interpolation). *Suppose the assumptions of Proposition C.12 hold with $\sigma^2 = 0$. Then the same result holds with*

$$T \geq \frac{4L_\epsilon}{\mu}\log\left(2\mu\mathrm{W}_2(q_0, q_*)^2\frac{1}{\epsilon}\right) \,.$$

## C.7.2. PROOF OF PROPOSITION C.12

The proof will establish a Wasserstein contraction. That is, we will establish an inequality of the form of

$$\mathbb{E}W_2(q_{t+1}, q_*)^2 \leq \rho_t \mathbb{E}W_2(q_t, q_*)^2 + \gamma_t^2 b$$

for some corresponding constants $\rho_t \in (0, 1)$ and $b_t < +\infty$. Informally, this amounts to obtaining a contraction between the two sequences generated as, for each $t \geq 0$,

$$q_{t+1} = \mathrm{JKO}_{\gamma_t \mathcal{H}}\left(\left(\mathrm{Id} - \gamma_t \widehat{\nabla_{\mathrm{BW}}\mathcal{E}}(q_t; \epsilon_t)\right)_{\#q_t}\right)$$
$$q_* = \mathrm{JKO}_{\gamma_t \mathcal{H}}\left(\left(\mathrm{Id} - \gamma_t \nabla_{\mathrm{BW}}\mathcal{E}(q_*)\right)_{\#q_*}\right), \quad (28)$$

which is reminiscent of the synchronous coupling approach of analyzing sampling algorithms (Durmus & Moulines, 2019; Dalalyan, 2017). (See also §4.1 of Chewi 2024.) More importantly, this differs from the strategy of Diao et al. (2023), who analyzed changes in the functionals $\mathcal{E}$ and $\mathcal{H}$ rather than the distance between iterates. We argue that our approach is more natural for strongly convex potentials and better aligned with contemporary analysis strategies of PSGD (Garrigos & Gower, 2023, §12.2).

To proceed with the Wasserstein contraction strategy, we need to establish that the sequence simulated by Eq. (28) is stationary and therefore represents the minimizer of $\mathcal{F}$.

**Lemma C.14.** *Suppose* $q_* \in \arg\min_{q \in \mathrm{BW}(\mathbb{R}^d)}\{\mathcal{F} = \mathcal{E} + \mathcal{H}\}$. *Then* $q_*$ *is a fixed point of the composition of the Bures–Wasserstein gradient descent step and the JKO operator such that*

$$q_* = \mathrm{JKO}_{\gamma_t \mathcal{H}}((\mathrm{Id} - \gamma_t \nabla_{\mathrm{BW}}\mathcal{E}(q_*))_{\#q_*}).$$

*Proof.* The proof is deferred to Section D.1.1.  □

Secondly, we need to ensure that the proximal step does not push the iterates farther apart. In the analysis of the canonical Euclidean proximal operator, this is represented by the non-expansiveness (1-Lipschitzness) of proximal operators. For the JKO operator, however, directly establishing and relying on non-expansiveness as follows appears to be new.

**Lemma C.15.** *Suppose the functional* $\mathcal{G} : \mathcal{P}_2(\mathbb{R}^d) \to (-\infty, +\infty]$ *satisfies the following:*

*(a) $\mathcal{G}$ admits a Bures–Wasserstein gradient for all $p \in \mathrm{BW}(\mathbb{R}^d)$,*

*(b) The output of $\mathrm{JKO}_{\mathcal{G}}(p)$ is unique for all $p \in \mathrm{BW}(\mathbb{R}^d)$, and*

*(c) $\mathcal{G}$ is convex along generalized geodesics such that, for any $p, q \in \mathrm{BW}(\mathbb{R}^d)$ and $\nu \in \mathrm{BW}(\mathbb{R}^d)$,*

$$\mathcal{G}(p) - \mathcal{G}(q) \geq \\ \mathbb{E}_\nu \langle \nabla_{\mathrm{BW}}\mathcal{G}(q) \circ M^*_{\nu \mapsto q}, M^*_{\nu \mapsto p} - M^*_{\nu \mapsto q} \rangle,$$

*where $M^*_{\nu \mapsto p}$ and $M^*_{\nu \mapsto q}$ are the optimal transport maps from $\nu$ to $p$ and $q$, respectively.*

*Then, for any $p, q \in \mathrm{BW}(\mathbb{R}^d)$, the Bures–Wasserstein JKO operator associated with $\mathcal{G}$ satisfies*

$$\mathrm{W}_2(\mathrm{JKO}_{\mathcal{G}}(p), \mathrm{JKO}_{\mathcal{G}}(q)) \quad \leq \quad \mathrm{W}_2(p, q).$$

*Proof.* The proof is deferred to Section D.1.2.  □

One might think that Lemma C.15 imposes assumptions that are stronger than typically expected for analyzing a proximal operator. In particular, we assumed the existence of the Bures–Wasserstein gradient and therefore ruled out non-Bures–Wasserstein-differentiable functionals. This is because, in this work, we only consider $\mathcal{G} = \gamma_t \mathcal{H}$, which is Bures–Wasserstein-differentiable. To rule out technicalities associated with non-differentiability beyond the scope of this paper, we opted for slightly stronger assumptions. However, we conjecture that Lemma C.15 should be generalizable to functionals with only Fréchet subdifferentials. (Refer to the work of Salim et al. 2020 for further discussions on the JKO operator.)

For our use case of $\mathcal{G} = \gamma_t \mathcal{H}$, (a) was established by Lambert et al. (2022), (b) is immediate by inspecting the closed form solution of $\mathrm{JKO}_{\gamma_t \mathcal{H}}$ established by Wibisono (2018), and (c) was shown by Diao et al. (2023, Lemma 3.2).

Since we now have Lemmas C.15 and C.14, we can show that the Bures–Wasserstein gradient descent step always makes the iterates go closer to each other up to some noise-induced perturbation.

**Lemma C.16.** *Suppose Assumption 3.1 holds and the chosen stochastic gradient estimator of the Bures–Wasserstein gradient satisfies Assumption C.11. Then, for any $t \geq 0$ and any $\gamma_t \leq 1/(2L_\epsilon)$, the iterates generated by SPBWGD satisfy*

$$\mathbb{E}[W_2(q_{t+1}, q_*)^2] \leq (1 - \mu\gamma_t)\mathbb{E}[W_2(q_t, q_*)^2] + 2\gamma_t^2\sigma^2 .$$

This follows from the fact that the Bures–Wasserstein gradient is coercive under Assumption 3.1, which is established in Lemma 3.8.

*Proof.* Denote

$$q_{t+1/2}^* = (\mathrm{Id} - \gamma_t \nabla_{\mathrm{BW}}\mathcal{E}(q_*))_{\#q_*} .$$

Given Lemmas C.15 and C.14, we have that

$$W_2(q_{t+1}, q_*)$$
$$= W_2\big(\mathrm{JKO}_{\gamma_t \mathcal{H}}(q_{t+1/2}), \mathrm{JKO}_{\gamma_t \mathcal{H}}(q_{t+1/2}^*)\big)$$
$$\leq W_2\big(q_{t+1/2}, q_{t+1/2}^*\big)$$
$$= W_2\big((\mathrm{Id} - \gamma_t \widehat{\nabla\mathcal{E}}(q_t; \epsilon_t))_{\#q_t}, (\mathrm{Id} - \gamma_t \nabla\mathcal{E}(q_*))_{\#q_*}\big) .$$

Denote the optimal coupling $\psi_t^* \in \Psi(q_t, q_*)$ between $q_t$ and $q_*$. Then, for the random variables $(X_t, X_*) \sim \psi_t^*$ and by the definition of the Wasserstein distance,

$$W_2\big((\mathrm{Id} - \gamma_t \widehat{\nabla_{\mathrm{BW}}\mathcal{E}}(q_t; \epsilon_t))_{\#q_t}, (\mathrm{Id} - \gamma_t \nabla_{\mathrm{BW}}\mathcal{E}(q_*))_{\#q_*}\big)^2$$
$$= \mathbb{E}\|X_t - \gamma_t \widehat{\nabla_{\mathrm{BW}}\mathcal{E}}(q_t; \epsilon_t)(X_t)$$
$$\qquad - X_* + \gamma_t \nabla_{\mathrm{BW}}\mathcal{E}(q_*)(X_*)\|_2^2 .$$

Expanding the square,

$$W_2(q_{t+1}, q_*)^2$$
$$\leq \mathbb{E}\Big[\|X_t - X_*\|_2^2$$
$$- 2\gamma_t \langle \widehat{\nabla_{\mathrm{BW}}\mathcal{E}}(q_t)(X_t; \epsilon_t) - \nabla_{\mathrm{BW}}\mathcal{E}(q_*)(X_*), X_t - X_* \rangle$$
$$+ \gamma_t^2 \|\widehat{\nabla_{\mathrm{BW}}\mathcal{E}}(q_t)(X_t; \epsilon_t) - \nabla_{\mathrm{BW}}\mathcal{E}(q_*)(X_*)\|_2^2\Big]$$
$$= W_2(q_t, q_*)^2$$
$$- 2\gamma_t \mathbb{E}\langle \widehat{\nabla_{\mathrm{BW}}\mathcal{E}}(q_t)(X_t; \epsilon_t) - \nabla_{\mathrm{BW}}\mathcal{E}(q_*)(X_*), X_t - X_* \rangle$$
$$+ \gamma_t^2 \mathbb{E}\|\widehat{\nabla_{\mathrm{BW}}\mathcal{E}}(q_t)(X_t; \epsilon_t) - \nabla_{\mathrm{BW}}\mathcal{E}(q_*)(X_*)\|_2^2 .$$

Taking expectation conditional on the filtration $\mathscr{F}_t$ of the iterates generated up to iteration $t$, the unbiasedness of the gradient estimator yields

$$\mathbb{E}[W_2(q_{t+1}, q_*) \mid \mathscr{F}_t]$$

$$\leq W_2(q_t, q_*)^2$$
$$- 2\gamma_t \mathbb{E}\langle \nabla_{\mathrm{BW}}\mathcal{E}(q_t)(X_t) - \nabla_{\mathrm{BW}}\mathcal{E}(q_*)(X_*), X_t - X_* \rangle$$
$$+ \gamma_t^2 \mathbb{E}\Big[\|\widehat{\nabla_{\mathrm{BW}}\mathcal{E}}(q_t)(X_t; \epsilon_t) - \nabla_{\mathrm{BW}}\mathcal{E}(q_*)(X_*)\|_2^2\Big] .$$

Under Assumption C.11, the gradient variance is bounded as

$$\leq W_2(q_t, q_*)^2$$
$$- 2\gamma_t \mathbb{E}\langle \nabla_{\mathrm{BW}}\mathcal{E}(q_t)(X_t) - \nabla_{\mathrm{BW}}\mathcal{E}(q_*)(X_*), X_t - X_* \rangle$$
$$+ \gamma_t^2 \left(4L_\epsilon \mathrm{D}_\mathcal{E}(q, q_*) + 2\sigma^2\right) ,$$

while the Bures–Wasserstein gradient is coercive in the sense of Lemma 3.8 such that

$$\leq W_2(q_t, q_*)^2 - 2\gamma_t \Big(\frac{\mu}{2}W_2(q_t, q_*)^2 + \mathrm{D}_\mathcal{E}(q_t, q_*)\Big)$$
$$+ \gamma_t^2 \left(4L_\epsilon \mathrm{D}_\mathcal{E}(q, q_*) + 2\sigma^2\right) .$$
$$= (1 - \mu\gamma_t)W_2(q_t, q_*)^2$$
$$- 2\gamma_t(1 - 2L_\epsilon\gamma_t)\mathrm{D}_\mathcal{E}(q, q_*) + 2\gamma_t^2\sigma^2$$
$$\leq (1 - \mu\gamma_t)W_2(q_t, q_*)^2 + 2\gamma_t^2\sigma^2 .$$

The last step follows from the non-negativity of $\mathrm{D}_\mathcal{E}$ and the step size limit $\gamma_t \leq 1/(2L_\epsilon)$. Taking full expectation yields the statement. $\qquad\square$

Notice that Lemma C.16 implies Eq. (22) under a specific choice of Lyapunov function. Therefore, Proposition C.12 follows by invoking Proposition C.8.

*Proof of Proposition C.12.* Lemma C.16 implies the Lyapunov decrease condition in Eq. (22) holds for all $t \geq 0$ with $\gamma_{\max} = \gamma_0 = 1/(2L_\epsilon)$, $V_t = \mathbb{E}[W_2(q_t, q_*)^2]$, and $b = 2\sigma^2$. Therefore, we can invoke Proposition C.8 with $\gamma_0 = 1/(2L_\epsilon)$, $\tau = 2/(\gamma_0\mu)$, and

$$t_* = \left\lceil \frac{1}{\log(1/(1 - \mu\gamma_0))} \log\left(\frac{\mu}{2\gamma_0\sigma^2}V_0\right) \right\rceil .$$

Then we have

$$T \geq \max\{B_{\mathrm{var}}, B_{\mathrm{bias}}\} \quad \Rightarrow \quad \mathbb{E}[W_2(q_T, q_*)^2] \leq \epsilon$$

with the constants

$$B_{\mathrm{var}} = \frac{8\sigma^2}{\mu}\frac{1}{\mu\epsilon} + 2\frac{\sqrt{2\sigma^2}}{\sqrt{\mu}}\frac{\sqrt{L_\epsilon}}{\sqrt{\mu}}$$
$$\times \left\{\log\left(\frac{2L_\epsilon}{\sigma^2}\mu V_0\right) + \frac{\mu}{4L_\epsilon} + \sqrt{2}\right\}\frac{1}{\sqrt{\mu\epsilon}}$$
$$B_{\mathrm{bias}} = 4\frac{L_\epsilon}{\mu}\log\left(2\mu V_0\frac{1}{\mu\epsilon}\right) .$$

where Adjusting for the dimensionless condition $\mu\mathbb{E}[W_2(q_T, q_*)^2] \leq \epsilon$ yields the statement. $\qquad\square$

## C.8. General Convergence Analysis of Proximal Gradient Descent

### C.8.1. PROPOSITION C.18

We present the Euclidean-space analog of Proposition C.12. The result is essentially a typical non-asymptotic convergence analysis of vanilla PSGD (Nemirovski et al., 2009). For the gradient estimator, we assume the following general condition:

**Assumption C.17.** *Denote the global optimum as $\lambda_* \in \arg\min_{\lambda \in \Lambda} \mathcal{F}(q_\lambda)$. There exists some constants $L_\epsilon, \sigma^2 \in (0, +\infty)$ such that, for all $\lambda \in \Lambda$, the stochastic estimator of the gradient of the energy $\widehat{\nabla_\lambda \mathcal{E}}$ satisfies the inequality*

$$\mathbb{E}_{\epsilon \sim \varphi} \|\widehat{\nabla_\lambda \mathcal{E}}(q_\lambda; \epsilon) - \nabla_{\lambda_*} \mathcal{E}(q_{\lambda_*})\|_2^2$$
$$\leq 4L_\epsilon \mathrm{D}_{\lambda \mapsto \mathcal{E}(q_\lambda)}(\lambda, \lambda_*) + 2\sigma^2 .$$

This assumption is a special case of Assumption 4.1 of Gorbunov et al. (2020). (Khaled et al. 2023 analyze SPGD on non-strongly-convex objectives under the same assumption.) Furthermore, it is a basic consequence of the "expected smoothness" assumption (Gower et al., 2021). Under Assumption C.17 and a fixed step size schedule (for all $t \geq 0$, $\gamma_t = \gamma$ for some $\gamma > 0$), it is known that (Garrigos & Gower, 2023, Theorem 12.10; Gorbunov et al., 2020, Corollary A.1) the iterates of SPGD satisfy the bound

$$\mathbb{E}\|\lambda_T - \lambda_*\|_2^2 \leq (1 - \mu\gamma)\|\lambda_0 - \lambda_*\|_2^2 + \frac{2\sigma^2}{\mu}\gamma .$$

where $\lambda_* = \arg\min_{\lambda \in \Lambda} \mathcal{F}(q_\lambda)$ is the global minimizer. This implies an iteration complexity of $\mathrm{O}(\epsilon^{-1} \log(\Delta\epsilon^{-1}))$ for ensuring $\forall \epsilon > 0$, $\mathbb{E}\|\lambda_T - \lambda_*\|_2^2 \leq \epsilon$ (Garrigos & Gower, 2023, Corollary 12.10).

Later on, Domke et al. (2023); Kim et al. (2023) removed the factor of $\log \epsilon^{-1}$ by employing the two-step step size schedule in Eq. (10) by Gower et al. (2019), originally developed for vanilla SGD. Their choice of switching time $t_*$, however, results in a worse dependence on $\Delta = \|\lambda_0 - \lambda_*\|_2$ in the iteration complexity $\mathrm{O}(\epsilon^{-1} + \Delta^2 \epsilon^{-1/2})$. By employing the switching time stated in Section C.6, we can simultaenously ensure a $\mathrm{O}(\epsilon^{-1})$ dependence on $\epsilon$ and a $\mathrm{O}(\log \Delta)$ dependence on $\Delta$ in an any-time fashion.

**Proposition C.18.** *Suppose Assumption 3.1 holds, the variational family is parametrized as in Assumption 2.2, and the chosen stochastic estimator of the parameter gradient satisfies Assumption C.17. Then, for any $\lambda_0 \in \Lambda$, running stochastic proximal gradient descent with the step size schedule in Eq. (10) with*

$$\gamma_0 = \frac{1}{4L_\epsilon}$$
$$\tau = \frac{2}{\mu\gamma_0}$$
$$t_* = \left\lceil \frac{1}{\log(1/(1-\mu\gamma_0))} \log\left(\frac{1}{2\gamma_0\sigma^2}\Delta^2\right) \right\rceil ,$$

*where we denote $\Delta^2 = \mu\|\lambda_0 - \lambda_*\|^2$, guarantees that*

$$T \geq \max\{B_{\mathrm{var}}, B_{\mathrm{bias}}\} \quad \Rightarrow \quad \mu\mathbb{E}[\mathrm{W}_2(q_{\lambda_T}, q_{\lambda_*})^2] \leq \epsilon,$$

*where*

$$B_{\mathrm{var}} = \frac{8\sigma^2}{\mu}\frac{1}{\epsilon} + 2\frac{\sqrt{2\sigma^2}}{\sqrt{\mu}}\frac{\sqrt{L_\epsilon}}{\sqrt{\mu}}$$
$$\times \left\{\log\left(\frac{2L_\epsilon}{\sigma^2}\Delta^2\right) + \frac{\mu}{4L_\epsilon} + \sqrt{2}\right\}\frac{1}{\sqrt{\epsilon}}$$
$$B_{\mathrm{bias}} = \frac{4L_\epsilon}{\mu}\log\left(2\Delta^2\frac{1}{\epsilon}\right) .$$

## C.8.2. PROOF OF PROPOSITION C.18

On a high level, the proof establishes a contraction of the Euclidean distance in parameter space $\mathbb{E}\|\lambda - \lambda_*\|_2^2$. This follows from the non-expansiveness of the proximal operator and the fact that the gradient descent step on the energy results in a contraction due to coercivity. The properties of the proximal operator are summarized as follows:

**Lemma C.19.** *Denote $\lambda_* \in \arg\min_{\lambda \in \Lambda} \mathcal{F}(q_\lambda)$, where $q_\lambda$ is parametrized as in Assumption 2.2. Then the proximal operator of the Boltzmann entropy $\lambda \mapsto \mathcal{H}(q_\lambda)$ is non-expansive such that, for any $\gamma > 0$, and any $\lambda, \lambda' \in \Lambda$,*

$$\|\mathrm{prox}_{\lambda \mapsto \gamma \mathcal{H}(q_\lambda)} - \mathrm{prox}_{\lambda \mapsto \gamma \mathcal{H}(q_{\lambda'})}\|_2^2 \leq \|\lambda - \lambda'\|_2^2 ,$$

*while $\lambda_*$ is a fixed point of the composition with a gradient descent step on the energy $\mathcal{E}$ such that*

$$\lambda_* = \mathrm{prox}_{\lambda \mapsto \gamma \mathcal{H}(q_\lambda)}(\lambda_* - \gamma \nabla_{\lambda_*} \mathcal{E}(q_{\lambda_*})) .$$

*Proof.* Under Assumption 2.2 $\lambda \mapsto \mathcal{H}(q_\lambda)$ is closed and convex on $\Lambda$ (Domke et al., 2023, Lemma 19). Therefore, the proximal operator of $\lambda \mapsto \gamma_t \mathcal{H}(q_\lambda)$ is well-defined in the sense that its variational problem $\mathrm{minimize}_{\lambda \in \Lambda} \mathcal{H}(q_\lambda) + (1/\gamma_t)\|\lambda - \lambda'\|_2^2$ has a unique solution due to strong convexity. Both non-expansiveness and the fact that $\lambda_*$ is a fixed point of the composition with a gradient descent step on $\mathcal{E}$ are proven by Garrigos & Gower (2023) in Lemma 8.17 and 8.18, respectively. □

Using this result, we obtain the one-step contraction.

**Lemma C.20.** *Suppose Assumption 3.1 holds, the variational family is parametrized as Assumption 2.2, and the chosen stochastic estimator of the parameter gradient satisfies Assumption C.17. Then, for each $t \geq 0$ and any $\gamma_t \leq 1/(2L_\epsilon)$,*

$$\mathbb{E}\|\lambda_{t+1} - \lambda_*\|_2^2 \leq (1 - \mu\gamma_t)\mathbb{E}\|\lambda_t - \lambda_*\|_2^2 + 2\gamma_t^2 \sigma^2 .$$

*Proof.* First, denote

$$\lambda_{t+1/2}^* \triangleq \lambda_* - \gamma_t \nabla_{\lambda_*} \mathcal{E}(q_{\lambda_*}) .$$

From Lemma C.19,

$$\|\lambda_{t+1} - \lambda_*\|_2^2$$
$$= \|\mathrm{prox}_{\lambda \mapsto \gamma_t \mathcal{H}(q_\lambda)}(\lambda_{t+1/2}) - \mathrm{prox}_{\lambda \mapsto \gamma_t \mathcal{H}(q_\lambda)}(\lambda_{t+1/2}^*)\|_2^2$$
$$\leq \|\lambda_{t+1/2} - \lambda_{t+1/2}^*\|_2^2$$
$$= \|\lambda_t - \gamma_t \widehat{\nabla_{\lambda_t} \mathcal{E}}(q_{\lambda_t}; \epsilon_t) - \lambda_* - \gamma_t \nabla_{\lambda_*} \mathcal{E}(q_{\lambda_*})\|_2^2 .$$

Expanding the square and taking expectation conditional on the filtration $\mathscr{F}_t$ generated by the iterates generated up to iteration $t$,

$$\mathbb{E}\big[\|\lambda_{t+1} - \lambda_*\|_2^2 \mid \mathscr{F}_t\big]$$

$$\leq \|\lambda_t - \lambda_*\|_2^2$$
$$\quad - 2\gamma_t \Big\langle \mathbb{E}\big[\widehat{\nabla_{\lambda_t} \mathcal{E}}(q_{\lambda_t}; \epsilon_t) \mid \mathscr{F}_t\big] - \mathcal{E}(q_{\lambda_*}), \lambda_t - \lambda_* \Big\rangle$$
$$\quad + \gamma_t^2 \mathbb{E}\Big[\|\widehat{\nabla_{\lambda_t} \mathcal{E}}(q_{\lambda_t}; \epsilon_t) - \nabla_{\lambda_*} \mathcal{E}(q_{\lambda_*})\|_2^2 \mid \mathscr{F}_t\Big] .$$

Since the gradient estimator is unbiased,

$$= \|\lambda_t - \lambda_*\|_2^2 - 2\gamma_t \langle \nabla_{\lambda_t} \mathcal{E}(q_{\lambda_t}) - \mathcal{E}(q_{\lambda_*}), \lambda_t - \lambda_* \rangle$$
$$\quad + \gamma_t^2 \mathbb{E}\Big[\|\widehat{\nabla_{\lambda_t} \mathcal{E}}(q_{\lambda_t}) - \nabla_{\lambda_*} \mathcal{E}(q_{\lambda_*})\|_2^2 \mid \mathscr{F}_t\Big] .$$

Under Assumption C.17, the gradient variance can be bounded as

$$\leq \|\lambda_t - \lambda_*\|_2^2 - 2\gamma_t \langle \nabla_{\lambda_t} \mathcal{E}(q_{\lambda_t}) - \mathcal{E}(q_{\lambda_*}), \lambda_t - \lambda_* \rangle$$
$$\quad + \gamma_t^2 \big(4L_\epsilon \mathrm{D}_{\lambda \mapsto \mathcal{E}(q_\lambda)}(\lambda_t, \lambda_*) + 2\sigma^2\big)$$

and from the fact that the gradient is coercive due to Lemma 3.9,

$$\leq \|\lambda_t - \lambda_*\|_2^2 - 2\gamma_t \Big(\frac{\mu}{2}\|\lambda_t - \lambda_*\|_2^2 + \mathrm{D}_{\lambda \mathcal{E}(q_\lambda)}(\lambda_t, \lambda_*)\Big)$$
$$\quad + \gamma_t^2 \big(4L_\epsilon \mathrm{D}_{\lambda \mapsto \mathcal{E}(q_\lambda)}(\lambda_t, \lambda_*) + 2\sigma^2\big)$$
$$= (1 - \mu\gamma_t)\|\lambda_t - \lambda_*\|_2^2$$
$$\quad - 2\gamma_t(1 - 2L_\epsilon \gamma_t)\mathrm{D}_{\lambda \mapsto \mathcal{E}(q_\lambda)}(\lambda_t, \lambda_*) + 2\gamma_t \sigma^2$$
$$\leq (1 - \mu\gamma_t)\|\lambda_t - \lambda_*\|_2^2 + 2\gamma_t \sigma^2 .$$

The last step follows from the non-negativity of the Bregman divergence and the step size limit $\gamma_t \leq 1/(2L_\epsilon)$. Taking full expectation yields the result. □

Lemma C.20 satisfies Eq. (22) for a specific Lyapunov function. Therefore, Proposition C.18 follows by invoking Proposition C.8.

*Proof of Proposition C.18.* Lemma C.20 implies the Lyapunov condition Eq. (22) for all $t \geq 0$ with $\gamma_{\max} = 1/(4L_\epsilon)$, $V_t = \mathbb{E}\|\lambda_t - \lambda_*\|_2^2$, and $b = 2\sigma^2$. Invoking Proposition C.8 with $\gamma_0 = 1/(4L_\epsilon)$, $\tau = 2/(\mu\gamma_0)$, and

$$t_* = \left\lceil \frac{1}{\log(1/(1 - \mu\gamma_0))} \log\left(\frac{\mu}{2\gamma_0 \sigma^2}V_0\right)\right\rceil .$$

we have

$$T \geq \max\{B_{\mathrm{var}}, B_{\mathrm{bias}}\} \quad \Rightarrow \quad \mathbb{E}\|\lambda_T - \lambda_*\|^2 \leq \epsilon$$

with the constants

$$B_{\mathrm{var}} = \frac{8\sigma^2}{\mu}\frac{1}{\mu\epsilon} + 2\frac{\sqrt{2\sigma^2}}{\sqrt{\mu}}\frac{\sqrt{L_\epsilon}}{\sqrt{\mu}}$$
$$\times \left\{\log\left(\frac{2L_\epsilon}{\sigma^2}\mu V_0\right) + \frac{\mu}{4L_\epsilon} + \sqrt{2}\right\}\frac{1}{\sqrt{\mu\epsilon}}$$

$$B_{\mathrm{bias}} = 4\frac{L_\epsilon}{\mu}\log\left(2\mu V_0 \frac{1}{\mu\epsilon}\right) .$$

Now, denote the coupling $\psi^{\mathrm{rep}} \in \Psi(q_{\lambda_T}, q_{\lambda_*})$ associated with the transport map $M^{\mathrm{rep}}_{q_{\lambda_T} \mapsto q_{\lambda_*}}$ and the squared Euclidean distance-optimal coupling $\psi^* \in \Psi(q_T, q_*)$. Then, from the identity Eq. (12), we have the ordering

$$\begin{aligned}
\|\lambda_T - \lambda_*\|^2 &= \mathbb{E}_{(Z_T, Z_*) \sim \psi^{\mathrm{rep}}} \|Z_T - Z_*\|_2^2 \\
&\geq \mathbb{E}_{(Z_T, Z_*) \sim \psi^*} \|Z_T - Z_*\|_2^2 \\
&= \mathrm{W}_2(q_{\lambda_T}, q_{\lambda_*})^2 \,.
\end{aligned}$$

Therefore,

$$\mathbb{E}\|\lambda_T - \lambda_*\|^2 \leq \epsilon \quad \Rightarrow \quad \mathbb{E}\mathrm{W}_2(q_{\lambda_T}, q_{\lambda_*})^2 \leq \epsilon \,.$$

Adjusting for the condition $\mu\mathbb{E}[\mathrm{W}_2(q_T, q_*)^2] \leq \epsilon$ yields the statement. $\qquad\square$

# D. Deferred Proofs

## D.1. Stochastic Proximal Bures–Wasserstein Gradient Descent

### D.1.1. STATIONARY POINT OF SPBWGD (PROOF OF LEMMA C.14)

**Lemma C.14** (Restated). *Suppose* $q_* \in \arg\min_{q\in\mathrm{BW}(\mathbb{R}^d)}\{\mathcal{F} = \mathcal{E} + \mathcal{H}\}$. *Then* $q_*$ *is a fixed point of the composition of the Bures–Wasserstein gradient descent step and the JKO operator such that*

$$q_* = \mathrm{JKO}_{\gamma_t\mathcal{H}}((\mathrm{Id} - \gamma_t\nabla_{\mathrm{BW}}\mathcal{E}(q_*))_{\#q_*}) .$$

*Proof.* Recall that, from Proposition C.4,

$$\mathbb{E}_{q^*}[\nabla U] = 0, \quad \mathbb{E}_{q^*}[\nabla^2 U] = \Sigma_*^{-1} .$$

and denote

$$q_{t+1/2}^* = \mathcal{N}(m_{t+1/2}^*, \Sigma_{t+1/2}^*) = (\mathrm{Id} - \gamma_t\nabla_{\mathrm{BW}}\mathcal{E}(q_*))_{\#q_*} ,$$
$$q_* = \mathcal{N}(m_*, \Sigma_*) .$$

The Bures–Wasserstein gradient descent step satisfies

$$m_{t+1/2}^* = m_* - \gamma_t\mathbb{E}_{q^*}[\nabla U] = m_*$$
$$M_{t+1/2}^* = \mathrm{I}_d - \gamma_t\mathbb{E}_{q^*}[\nabla^2 U] = \mathrm{I}_d - \gamma_t\,\Sigma_*^{-1}$$
$$\begin{aligned}\Sigma_{t+1/2}^* &= M_{t+1/2}^*\Sigma_* M_{t+1/2}^* \\ &= (\mathrm{I}_d - \gamma_t\Sigma_*^{-1})\Sigma_*(\mathrm{I}_d - \gamma_t\,\Sigma_*^{-1}) \\ &= (\mathrm{I}_d - \gamma_t\Sigma_*^{-1})^2\Sigma_* ,\end{aligned}$$

where the last step follows from the fact that the factors in the product all share the same eigenvectors. In addition, $q_{t+1}^* = \mathcal{N}(m_{t+1}^*, \Sigma_{t+1}^*) \triangleq \mathrm{JKO}_{\gamma_t\mathcal{H}}(q_{t+1/2}^*)$. Since the proximal step preserves the mean, *i.e.*, $m_{t+1}^* = m_*$, it suffices to prove that $\Sigma_{t+1}^* = \Sigma_*$. From the update of $\Sigma_{t+1/2}^*$, we have

$$\begin{aligned}\Sigma_{t+1/2}^* + 4\gamma_t\mathrm{I}_d &= (\mathrm{I}_d - \gamma_t\Sigma_*^{-1})\Sigma_*(\mathrm{I}_d - \gamma_t\,\Sigma_*^{-1}) + 4\gamma_t\mathrm{I}_d \\ &= \Sigma_* + 2\gamma_t\mathrm{I}_d + \gamma_t^2\Sigma_*^{-1} \\ &= \Sigma_*(\mathrm{I}_d + \gamma_t\Sigma_*^{-1})^2.\end{aligned}$$

Then we obtain

$$\begin{aligned}\Sigma_{t+1}^* &= \frac{1}{2}\Bigg\{\Sigma_{t+1/2}^* + 2\gamma_t\mathrm{I}_d \\ &\qquad + \left[\Sigma_{t+1/2}^*\left(\Sigma_{t+1/2}^* + 4\gamma_t\mathrm{I}_d\right)\right]^{1/2}\Bigg\} \\ &= \frac{1}{2}\Bigg\{\Sigma_* + \gamma_t^2\Sigma_*^{-1} \\ &\qquad + \left[(\mathrm{I}_d - \gamma_t\Sigma_*^{-1})^2\Sigma_*^2(\mathrm{I}_d + \gamma_t\Sigma_*^{-1})^2\right]^{1/2}\Bigg\} \\ &= \frac{1}{2}\Big\{\Sigma_* + \gamma_t^2\Sigma_*^{-1}\end{aligned}$$

$$+ (\mathrm{I}_d - \gamma_t\Sigma_*^{-1})\Sigma_*(\mathrm{I}_d + \gamma_t\Sigma_*^{-1})\Big\}$$
$$= \frac{1}{2}\left\{\Sigma_* + \gamma_t^2\Sigma_*^{-1} + \Sigma_* - \gamma_t^2\Sigma_*^{-1}\right\}$$
$$= \Sigma_*.$$

Thus $q_{t+1}^* = q_*$ and we conclude that $q_*$ is a fixed point. $\square$

### D.1.2. NON-EXPANSIVENESS OF THE JKO OPERATOR (PROOF OF LEMMA C.15)

**Lemma C.15** (Restated). *Suppose the functional* $\mathcal{G}$ : $\mathcal{P}_2(\mathbb{R}^d) \to (-\infty, +\infty]$ *satisfies the following:*

*(a)* $\mathcal{G}$ *admits a Bures–Wasserstein gradient for all* $p \in \mathrm{BW}(\mathbb{R}^d)$,

*(b) The output of* $\mathrm{JKO}_{\mathcal{G}}(p)$ *is unique for all* $p \in \mathrm{BW}(\mathbb{R}^d)$, *and*

*(c)* $\mathcal{G}$ *is convex along generalized geodesics such that, for any* $p, q \in \mathrm{BW}(\mathbb{R}^d)$ *and* $\nu \in \mathrm{BW}(\mathbb{R}^d)$,

$$\mathcal{G}(p) - \mathcal{G}(q) \geq \\ \mathbb{E}_\nu \langle \nabla_{\mathrm{BW}} \mathcal{G}(q) \circ M^*_{\nu \mapsto q}, M^*_{\nu \mapsto p} - M^*_{\nu \mapsto q} \rangle,$$

*where* $M^*_{\nu \mapsto p}$ *and* $M^*_{\nu \mapsto q}$ *are the optimal transport maps from* $\nu$ *to* $p$ *and* $q$, *respectively.*

*Then, for any* $p, q \in \mathrm{BW}(\mathbb{R}^d)$, *the Bures–Wasserstein JKO operator associated with* $\mathcal{G}$ *satisfies*

$$\mathrm{W}_2(\mathrm{JKO}_{\mathcal{G}}(p), \mathrm{JKO}_{\mathcal{G}}(q)) \quad \leq \quad \mathrm{W}_2(p, q).$$

The proof utilizes the regularity properties of the JKO operator established by Salim et al. (2020); Ambrosio et al. (2005). However, to avoid technicalities related to non-differentiability, we restrict our interest to Bures–Wasserstein-differentiable functionals $\mathcal{G}$.

*Proof.* Denote

$$p \triangleq \mathrm{Normal}(m_p, \Sigma_p) \qquad q \triangleq \mathrm{Normal}(m_q, \Sigma_q)$$
$$p' \triangleq \mathrm{JKO}_{\mathcal{G}}(p) \qquad q' \triangleq \mathrm{JKO}_{\mathcal{G}}(q).$$

Under the assumptions on $\mathcal{G}$, the JKO updates can be expressed as

$$X' = X - \nabla_{\mathrm{BW}} \mathcal{G}(p)(X'); \qquad X' \sim p'$$
$$Y' = Y - \nabla_{\mathrm{BW}} \mathcal{G}(q)(Y'); \qquad Y' \sim q',$$

where $(X, Y)$ are optimally coupled. Now, consider some $Z \sim \nu = \mathrm{Normal}(0, \mathrm{I}_d)$. We know that the optimal transport maps $M^*_{\nu \mapsto p'}$ and $M^*_{\nu \mapsto q'}$ exist (Villani, 2009, Theorem 4.1) uniquely (Brenier, 1991). From the assumption on $\mathcal{G}$, we have

$$\mathcal{G}(p') - \mathcal{G}(q') \\ \geq \mathbb{E}_\nu \langle \nabla_{\mathrm{BW}} \mathcal{G}(q') \circ M^*_{\nu \mapsto q'}, M^*_{\nu \mapsto p'} - M^*_{\nu \mapsto q'} \rangle \\ = \mathbb{E} \langle \nabla_{\mathrm{BW}} \mathcal{G}(q')(Y'), X' - Y' \rangle,$$
$$\mathcal{G}(q') - \mathcal{G}(p') \\ \geq \mathbb{E}_\nu \langle \nabla_{\mathrm{BW}} \mathcal{G}(p') \circ M^*_{\nu \mapsto p'}, M^*_{\nu \mapsto q'} - M^*_{\nu \mapsto p'} \rangle$$

$$= \mathbb{E} \langle \nabla_{\mathrm{BW}} \mathcal{G}(p')(X'), Y' - X' \rangle.$$

Summing up the two inequalities above yields

$$\mathbb{E} \langle \nabla_{\mathrm{BW}} \mathcal{G}(p')(X') - \nabla_{\mathrm{BW}} \mathcal{G}(q')(Y'), X' - Y' \rangle \geq 0. \tag{29}$$

Rearranging the JKO updates,

$$X' - Y' + \nabla_{\mathrm{BW}} \mathcal{G}(p')(X') - \nabla_{\mathrm{BW}} \mathcal{G}(q')(Y') = X - Y.$$

Taking the inner product with $X' - Y'$ for both sides and taking the expectation, we have

$$\mathbb{E} \langle \nabla_{\mathrm{BW}} \mathcal{G}(p')(X') - \nabla_{\mathrm{BW}} \mathcal{G}(q')(Y'), X' - Y' \rangle \\ = \mathbb{E} \left[ \langle X - Y, X' - Y' \rangle - \|X' - Y'\|_2^2 \right] \geq 0, \tag{30}$$

where the inequality follows from Eq. (29). Now, from Cauchy-Schwarz and Young's inequality, we know that

$$\mathbb{E}\|X' - Y'\|_2^2 \leq \mathbb{E} \langle X - Y, X' - Y' \rangle \\ \leq \frac{1}{2} \mathbb{E}\|X - Y\|_2^2 + \frac{1}{2} \mathbb{E}\|X' - Y'\|_2^2. \tag{31}$$

Combining Eqs. (31) and (30),

$$\mathbb{E} \left[ \|X' - Y'\|_2^2 \right] \leq \mathbb{E} \left[ \|X - Y\|_2^2 \right].$$

Thus

$$\mathrm{W}_2(\mathrm{JKO}_{\mathcal{G}}(p), \mathrm{JKO}_{\mathcal{G}}(q))^2 \leq \mathbb{E} \left[ \|X' - Y'\|_2^2 \right] \\ \leq \mathbb{E} \left[ \|X - Y\|_2^2 \right] \\ = \mathrm{W}_2(p, q)^2.$$

$\square$

D.1.3. ITERATION COMPLEXITY (PROOF OF
      THEOREM 3.2)

**Theorem 3.2** (Restated). *Suppose Assumption 3.1 holds and the gradient estimator $\widehat{\nabla_{\mathrm{BW}}^{\mathrm{bonnet\text{-}price}}}\mathcal{E}$ is used. Then, for any $\epsilon > 0$, there exists some $t_*$ and $\tau$ (shown explicitly in the proof) such that running stochastic proximal Bures–Wasserstein gradient descent with the step size schedule in Eq. (10) with $\gamma_0 = 1/(10L\kappa)$ guarantees*

$$T \gtrsim d\kappa\frac{1}{\epsilon} + \sqrt{d}\,\kappa^{3/2}\log\big(\kappa\Delta^2\big)\frac{1}{\sqrt{\epsilon}} + \kappa^2\log\Big(\Delta^2\frac{1}{\epsilon}\Big)$$

$$\Rightarrow \quad \mu\mathbb{E}[W_2(q_T, q_*)^2] \le \epsilon \,,$$

*where $\Delta^2 = \mu W(q_0, q_*)^2$.*

This is a corollary of Proposition C.12, where the sufficient conditions are established in Lemmas 3.4 and 3.6.

*Proof.* For any $q \in \mathrm{BW}(\mathbb{R}^d)$, denote $\psi^* \in \Psi(q, q_*)$, the coupling between $q$ and $q_*$ optimal in terms of squared Euclidean distance. Under the stated conditions, we can invoke Lemma 3.4, where the generic coupling $\psi$ in the statement of Lemma 3.4 can be set as $\psi = \psi^*$. That is,

$$\mathbb{E}\|\widehat{\nabla_{\mathrm{BW}}^{\mathrm{bonnet\text{-}price}}}\mathcal{E}(q;\epsilon)(X_t) - \nabla_{\mathrm{BW}}\mathcal{E}(q_*)(X_*)\|_2^2$$
$$\le 10L\kappa\,\mathbb{E}_{(X,X_*)\sim\psi^*}[\mathrm{D}_U(X, X_*)] + 10dL$$
$$= 10L\kappa\,\mathrm{D}_{\mathcal{E}}(q, q_*) + 10dL \,.$$

This establishes

$$\text{Lemma 3.4 \& Lemma 3.6} \quad \Rightarrow \quad \text{Assumption C.11}$$

with the constants $L_\epsilon = 5/2L\kappa$ and $\sigma^2 = 5dL$. Then

$$\text{Assumption 3.1 \& Assumption C.11} \quad \Rightarrow \quad \text{Proposition C.12} \,.$$

Substituting for the constants $L_\epsilon$ and $\sigma^2$, the parameters of the step size schedule in Eq. (10) become

$$\gamma_0 = \frac{1}{10L\kappa}$$
$$\tau = 8\kappa$$
$$t_* = \left\lceil \frac{1}{\log(1/(1 - 1/10\kappa^2))} \log\Big(\frac{\kappa}{d}\Delta^2\Big) \right\rceil \,,$$

which guarantee

$$T \ge \max\{B_{\mathrm{var}}, B_{\mathrm{bias}}\} \quad \Rightarrow \quad \mu\mathbb{E}[W_2(q_T, q_*)^2] \le \epsilon$$

with the constants

$$B_{\mathrm{var}} = 40d\kappa\frac{1}{\epsilon} + 10\sqrt{d}\,\kappa^{3/2}$$
$$\times \left\{ \log\Big(\frac{\kappa}{d}\Delta^2\Big) + \frac{1}{10\kappa^2} + \sqrt{2} \right\}\frac{1}{\sqrt{\epsilon}}$$
$$B_{\mathrm{bias}} = 10\kappa^2\log\Big(2\Delta^2\frac{1}{\epsilon}\Big) \,.$$

$\square$

### D.1.4. VARIANCE BOUND ON THE BURES-WASSERSTEIN GRADIENT ESTIMATOR (PROOF OF LEMMA 3.4)

**Lemma 3.4** (Restated). *Suppose Assumption 3.1 holds and $q_* = \mathrm{Normal}(\mu_*, \Sigma_*) \in \arg\min_{q \in \mathrm{BW}(\mathbb{R}^d)} \mathcal{F}(q)$. Then, for any $q \in \mathrm{BW}(\mathbb{R}^d)$, and any coupling $\psi \in \Psi(q, q_*)$,*

$$\mathbb{E}_{(X,X_*)\sim\psi,\epsilon\sim\varphi}\big[\|\widehat{\nabla^{\mathrm{bonnet-price}}_{\mathrm{BW}}}\mathcal{E}(q;\epsilon)(X) - \nabla\mathcal{E}(q_*)(X_*)\|_2^2\big]$$
$$\leq 10L\kappa\,\mathbb{E}_{(X,X_*)\sim\psi}[\mathrm{D}_U(X,X_*)] + 10dL\,.$$

Recall the definition of the gradient estimator

$$\widehat{\nabla^{\mathrm{bonnet-price}}_{\mathrm{BW}}}\mathcal{E}(q;\epsilon)(x)$$
$$= x \mapsto \widehat{\nabla^{\mathrm{bonnet}}_m}\mathcal{E}(q;\epsilon) + 2\widehat{\nabla^{\mathrm{price}}_\Sigma}\mathcal{E}(q;\epsilon)(x-m)$$
$$= x \mapsto \nabla U(Z) + \nabla^2 U(Z)(x-m)\,,$$

where $Z$ is sampled given $\epsilon$ as, for example, $Z = \phi_\lambda(\epsilon)$. First, we can decompose the gradient variance into two terms, each corresponding to the contribution of the gradient with respect to the two parameters of $q_*$, $m_t$ and $\Sigma_t$.

*Proof.* Since the gradients corresponding to $m_t$ and $\Sigma_t$ are orthogonal in $\mathrm{L}^2(\psi^*)$, taking expectation over $\epsilon \sim \varphi$ and $(X, X_*) \sim \psi^*$, we have

$$\mathbb{E}\left[\|\widehat{\nabla^{\mathrm{bonnet-price}}_{\mathrm{BW}}}\mathcal{E}(q;\epsilon)(X_t) - \nabla_{\mathrm{BW}}\mathcal{E}(q_*)(X_*)\|_2^2\right]$$
$$= \mathbb{E}\Big\|\big(\widehat{\nabla^{\mathrm{bonnet}}_m}\mathcal{E}(q;\epsilon) + 2\widehat{\nabla^{\mathrm{price}}_\Sigma}\mathcal{E}(q;\epsilon)(X-m)\big)$$
$$\quad - \big(\nabla_m\mathcal{E}(q_*) + 2\nabla_\Sigma\mathcal{E}(q_*)(X_*-m_*)\big)\Big\|_2^2$$
$$= \mathbb{E}\Big\|\widehat{\nabla^{\mathrm{bonnet}}_m}\mathcal{E}(q;\epsilon) - \nabla_m\mathcal{E}(q_*)\Big\|_2^2$$
$$\quad + \mathbb{E}\Big\|2\widehat{\nabla^{\mathrm{price}}_\Sigma}\mathcal{E}(q;\epsilon)(X-m) - 2\nabla_\Sigma\mathcal{E}(q_*)(X_*-m_*)\Big\|_2^2$$
$$= \underbrace{\mathbb{E}_{\epsilon\sim\varphi}\Big\|\widehat{\nabla^{\mathrm{bonnet}}_m}\mathcal{E}(q;\epsilon) - \nabla_m\mathcal{E}(q_*)\Big\|_2^2}_{\text{variance of gradient w.r.t. } m_t}$$
$$+ \underbrace{\mathbb{E}\Big\|2\widehat{\nabla^{\mathrm{price}}_\Sigma}\mathcal{E}(q;\epsilon)(X-m) - 2\nabla_\Sigma\mathcal{E}(q_*)(X_*-m_*)\Big\|_2^2}_{\text{variance of gradient w.r.t. }\Sigma}\,.$$
$$(32)$$

We will analyze each term separately.

First, the gradient variance contributed by $m$ is bounded as

$$\mathbb{E}_{\epsilon\sim\varphi}\Big\|\widehat{\nabla^{\mathrm{bonnet}}_m}\mathcal{E}(q;\epsilon) - \nabla_m\mathcal{E}(q_*)\Big\|_2^2$$
$$= \mathbb{E}_{Z\sim q}\|\nabla U(Z) - \mathbb{E}_{q_*}\nabla U\|_2^2$$
$$\leq 4L\,\mathbb{E}_{(X,X_*)\sim\psi}[\mathrm{D}_U(X,X_*)] + 2dL\,, \qquad (33)$$

where the last step follows from Lemma C.7.

Second, the gradient variance contributed by $\Sigma_t$ can be decomposed as

$$\mathbb{E}\Big\|2\widehat{\nabla^{\mathrm{price}}_\Sigma}\mathcal{E}(q;\epsilon)(X-m) - 2\nabla_\Sigma\mathcal{E}(q_*)(X_*-m_*)\Big\|_2^2$$
$$= \mathbb{E}\big\|\nabla^2 U(Z)(X-m) - \mathbb{E}_{q_*}\big[\nabla^2 U\big](X_*-m_*)\big\|_2^2$$
$$\leq 2\underbrace{\mathbb{E}_{Z\sim q,X\sim q}\big\|\nabla^2 U(Z)(X-m)\big\|_2^2}_{\triangleq V_{\mathrm{mul}}}$$
$$+ 2\underbrace{\mathbb{E}_{X_*\sim q_*}\big\|\mathbb{E}_{q_*}\big[\nabla^2 U\big](X_*-m_*)\big\|_2^2}_{\triangleq V_{\mathrm{add}}} \qquad (34)$$

by Young's inequality.

The gradient variance $\widehat{\nabla^{\mathrm{price}}_\Sigma}\mathcal{E}$, and in turn $\widehat{\nabla^{\mathrm{bonnet-price}}_{\mathrm{BW}}}\mathcal{E}$, is dominated by the multiplicative noise variance $V_{\mathrm{mul}}$. Therefore, bounding $V_{\mathrm{mul}}$ is the most crucial step, which follows from Lemma C.5. That is,

$$V_{\mathrm{mul}} = \mathbb{E}_{X\sim q,Z\sim q}\big\|\nabla^2 U(Z)(X-m)\big\|_2^2$$
$$= \mathbb{E}_{Z\sim q}\,\mathrm{tr}\Big(\big(\nabla^2 U(Z)\big)^2\mathbb{E}_{X\sim q}(X-m)(X-m)^\top\Big)$$
$$= \mathbb{E}_{Z\sim q}\,\mathrm{tr}\big(\nabla^2 U(Z)\Sigma\nabla^2 U(Z)\big)$$
$$\leq L\big(2\sqrt{\kappa}+\kappa\big)\mathbb{E}_{(X,X_*)\sim\psi}[\mathrm{D}_U(X,X_*)] + 3dL\,,$$

where the last step follows from Lemma C.5. On the other hand, $V_{\mathrm{add}}$ immediately follows from the stationarity condition for minimizing the free energy $\mathcal{F}$.

$$V_{\mathrm{add}} = \mathbb{E}_{X_*\sim q_*}\big\|\mathbb{E}_{q_*}\nabla^2 U(X_*-m_*)\big\|_2^2$$
$$= \mathrm{tr}\Big(\big(\mathbb{E}_{q_*}\nabla^2 U\big)^2\mathbb{E}_{X_*\sim q_*}(X_*-m_*)(X_*-m_*)^\top\Big)$$
$$= \mathrm{tr}\Big(\big(\mathbb{E}_{q_*}\nabla^2 U\big)^2\Sigma_*\Big)\,.$$

The stationary condition (Proposition C.4) yields

$$= \mathrm{tr}\big(\Sigma_*^{-2}\Sigma_*\big)$$
$$= \mathrm{tr}\big(\Sigma_*^{-1}\big)\,.$$

Therefore, resuming from Eq. (34),

$$\mathbb{E}\Big\|2\widehat{\nabla^{\mathrm{price}}_\Sigma}\mathcal{E}(q;\epsilon)(X-m) - 2\nabla_\Sigma\mathcal{E}(q_*)(X_*-m_*)\Big\|_2^2$$
$$\leq 2V_{\mathrm{add}} + 2V_{\mathrm{mul}}$$
$$\leq L\big(4\sqrt{\kappa}+2\kappa\big)\mathbb{E}_{(X,X_*)\sim\psi}[\mathrm{D}_U(X,X_*)]$$
$$\quad + 6dL + 2\,\mathrm{tr}\big(\Sigma_*^{-1}\big)$$
$$\leq L\big(4\sqrt{\kappa}+2\kappa\big)\mathbb{E}_{(X,X_*)\sim\psi}[\mathrm{D}_U(X,X_*)] + 8dL\,, \quad (35)$$

where we used the fact that $\Sigma_*^{-1} = \mathbb{E}_{q_*}\nabla^2 U \preceq L\mathrm{I}_d$.

Combining Eqs. (33) and (35) into Eq. (32),

$$\mathbb{E}\Big[\|\widehat{\nabla^{\mathrm{bonnet-price}}}\mathcal{E}(q;\epsilon)(X_t) - \nabla\mathcal{E}(q_*)(X_*)\|_2^2\Big]$$

$$\leq \left\{ 4L \, \mathbb{E}_{(X,X_*)\sim\psi}[\mathrm{D}_U(X, X_*)] + 2dL \right\}$$

$$+ \left\{ L\big(4\sqrt{\kappa} + 2\kappa\big) \mathbb{E}_{(X,X_*)\sim\psi}[\mathrm{D}_U(X, X_*)] + 8dL \right\}$$

$$\leq 10L\kappa \, \mathbb{E}_{(X,X_*)\sim\psi}[\mathrm{D}_U(X, X_*)] + 10dL \, ,$$

where we used the fact that $\kappa \geq 1$. $\qquad\qquad\square$

### D.1.5. WASSERSTEIN BREGMAN DIVERGENCE IDENTITY (PROOF OF LEMMA 3.6)

**Lemma 3.6** (Restated). *For any $p, q \in \mathrm{BW}(\mathbb{R}^d)$, denote the coupling optimal for the squared Euclidean cost between $p$ and $q$ as $\psi_* \in \Psi(p, q)$. Then*

$$\mathbb{E}_{(X,Y)\sim\psi_*}[\mathrm{D}_U(X, Y)] = \mathrm{D}_{\mathcal{E}}(p, q) \, .$$

*Proof.* By definition of the Bregman divergence,

$$\mathbb{E}_{(X,Y)\sim\psi^*}[\mathrm{D}_U(X, Y)]$$
$$= \mathbb{E}_{(X,Y)\sim\psi^*}[U(X) - U(Y) - \langle \nabla U(Y), X - Y \rangle]$$
$$= \mathcal{E}(p) - \mathcal{E}(q) - \mathbb{E}_{(X,Y)\sim\psi^*}[\langle \nabla U(Y), X - Y \rangle]$$
$$= \mathrm{D}_{\mathcal{E}}(p, q) + \mathbb{E}_{(X,Y)\sim\psi^*}[\langle \nabla\mathcal{E}(q)(Y), X - Y \rangle]$$
$$\qquad\qquad - \mathbb{E}_{(X,Y)\sim\psi^*}[\langle \nabla U(Y), X - Y \rangle] \, .$$
$$(36)$$

It remains to show that the two inner product terms cancel.

For any $r = \mathrm{Normal}(m, \Sigma) \in \mathrm{BW}(\mathbb{R}^d)$, the Wasserstein gradient $\nabla_{\mathrm{W}}$ of the energy $\mathcal{E}$ is $\nabla_{\mathrm{W}}\mathcal{E} = \nabla U$. Furthermore, the Bures-Wasserstein gradient of $\mathcal{E}$ is the orthogonal $\mathrm{L}^2(r)$ projection of the Wasserstein gradient onto the tangent space of $\mathrm{BW}(\mathbb{R}^d)$ (Lambert et al., 2022, Appendix C.1)

$$\mathcal{T}_r\mathrm{BW}(\mathbb{R}^d) = \left\{ x \mapsto a + S(x - m) \mid a \in \mathbb{R}^d, S \in \mathbb{S}^d \right\} \, .$$

That is,

$$\nabla_{\mathrm{BW}}\mathcal{E}(r) = \mathrm{proj}_{\mathcal{T}_r\mathrm{BW}(\mathbb{R}^d)}(\nabla_{\mathrm{W}}\mathcal{E})$$
$$= \mathrm{proj}_{\mathcal{T}_r\mathrm{BW}(\mathbb{R}^d)}(\nabla U)$$
$$= \underset{w\in\mathcal{T}_r\mathrm{BW}(\mathbb{R}^d)}{\arg\min} \, \mathbb{E}_r\|\nabla U - w\|_2^2 \, ,$$

which is the unique element of $\mathcal{T}_r\mathrm{BW}(\mathbb{R}^d)$ satisfying, for all $v \in \mathcal{T}_r\mathrm{BW}(\mathbb{R}^d)$,

$$\mathbb{E}_r\langle \nabla_{\mathrm{BW}}\mathcal{E}(r), v \rangle = \mathbb{E}_r\langle \mathrm{proj}_{\mathcal{T}_r\mathrm{BW}(\mathbb{R}^d)}(\nabla U), v \rangle$$
$$= \mathbb{E}_p\langle \nabla U, v \rangle \, . \qquad (37)$$

Now, $\mathcal{T}_r\mathrm{BW}(\mathbb{R}^d)$ is equivalent to the set of all affine maps with a symmetric matrix. Therefore, for any such map, the identity Eq. (37) holds.

Under the optimal coupling $\psi^*$, denote the optimal transport map from $p = \mathrm{Normal}(m_p, \Sigma_p)$ to $q = \mathrm{Normal}(m_q, \Sigma_q)$ as $M_{p\mapsto q}^*$, which is an affine function of the form of

$$M_{p\mapsto q}^* = S_{p\mapsto q}(x - m_p) + m_q \, ,$$

where

$$S_{p\mapsto q} = \Sigma_p^{-\frac{1}{2}} \left( \Sigma_p^{\frac{1}{2}} \Sigma_q \Sigma_p^{\frac{1}{2}} \right)^{\frac{1}{2}} \Sigma_p^{-\frac{1}{2}}$$

is a symmetric matrix. From inspection, it is clear that $M^*_{p \mapsto q} \in \mathcal{T}_p \mathrm{BW}(\mathbb{R}^d)$. Consequently, we can also conclude that $M^*_{p \mapsto q} - \mathrm{Id} \in \mathcal{T}_p \mathrm{BW}(\mathbb{R}^d)$. Therefore,

$$
\begin{aligned}
\mathbb{E}_{(X,Y)\sim\psi^*} & \langle \nabla_{\mathrm{BW}} \mathcal{E}(q)(Y), X - Y \rangle \\
&= \mathbb{E}_q \langle \nabla_{\mathrm{BW}} \mathcal{E}(q), M^*_{p \mapsto q} - \mathrm{Id} \rangle \\
&= \mathbb{E}_q \langle \nabla U, M^*_{p \mapsto q} - \mathrm{Id} \rangle \qquad \text{(Eq. (37))} \\
&= \mathbb{E}_{(X,Y)\sim\psi^*} \langle \nabla U(Y), X - Y \rangle \,,
\end{aligned}
$$

which means the outstanding inner products in Eq. (36) cancel out completely. $\qquad\square$

## D.1.6. COERCIVITY OF BURES-WASSERSTEIN GRADIENT (PROOF OF LEMMA 3.8)

**Lemma 3.8** (Restated). *Suppose Assumption 3.1 holds. Then, for any $q \in \mathrm{BW}(\mathbb{R}^d)$ and $q_* = \arg\min_{q \in \mathrm{BW}(\mathbb{R}^d)} \mathcal{F}(q)$, where we denote their coupling $\psi_* \in \Psi(q, q_*)$ optimal in terms of squared Euclidean distance,*

$$
\begin{aligned}
\mathbb{E}_{(X,Y)\sim\psi_*} & \langle \nabla_{\mathrm{BW}} \mathcal{E}(p)(X) - \nabla_{\mathrm{BW}} \mathcal{E}(q)(Y), X - Y \rangle \\
& \geq \frac{\mu}{2} \mathrm{W}_2(q_t, q_*)^2 + \mathrm{D}_{\mathcal{E}}(q_t, q_*) \,.
\end{aligned}
$$

To prove Lemma 3.8, we need the energy $\mathcal{E}$ to be $\mu$-strongly convex under a certain notion of convexity in Bures-Wasserstein space. This is given by the following supporting result:

**Lemma D.1** (Lemma B.1; Diao et al., 2023). *Suppose Assumption 3.1 holds. Then, for any $q \in \mathrm{BW}(\mathbb{R}^d)$ and any affine map $M : \mathbb{R}^d \to \mathbb{R}^d$, the expected energy $\mathcal{E}$ is $\mu$-strongly geodesically convex such that*

$$
\begin{aligned}
\mathcal{E}((\mathrm{Id} + M)_{\#q}) & - \mathcal{E}(q) \\
& \geq \mathbb{E}_q \langle \nabla_{\mathrm{BW}} \mathcal{E}(q), M \rangle + \frac{\mu}{2} \mathbb{E}_q \|M\|_2^2 \,.
\end{aligned}
$$

Since the optimal transport map between $q_t$ and $q_*$ is an affine map, the difference $X - Y$ is also an affine map. Therefore, Lemma D.1 can be used to establish coercivity.

*Proof of Lemma 3.8.* Under the optimal coupling $\psi_* \in \Psi(p, q)$, the difference $X - Y$ is an affine map over the Law of $X$ or $Y$. Therefore,

$$
\begin{aligned}
\mathbb{E}_{(X,Y)\sim\psi_*} & \langle \nabla_{\mathrm{BW}} \mathcal{E}(p)(X) - \nabla \mathcal{E}(q)(Y), X - Y \rangle \\
&= -\mathbb{E}_{(X,Y)\sim\psi_*} \langle \nabla_{\mathrm{BW}} \mathcal{E}(p)(X), Y - X \rangle \\
& \quad - \mathbb{E}_{(X,Y)\sim\psi_*} \langle \nabla_{\mathrm{BW}} \mathcal{E}(q)(Y), X - Y \rangle \\
& \geq -\left( \mathcal{E}(q) - \mathcal{E}(p) - \frac{\mu}{2} \mathrm{W}_2(p,q)^2 \right) .
\end{aligned}
$$

The $\mu$-strong geodesic convexity (Lemma D.1) of $\mathcal{E}$ implies

$$
\begin{aligned}
& - \mathbb{E}_{(X,Y)\sim\psi_*} \langle \nabla_{\mathrm{BW}} \mathcal{E}(q)(Y), X - Y \rangle \\
&= \frac{\mu}{2} \mathrm{W}_2(p,q)^2 + \big( \mathcal{E}(p) - \mathcal{E}(q) \\
& \quad - \mathbb{E}_{(X,Y)\sim\psi_*} \langle \nabla_{\mathrm{BW}} \mathcal{E}(q)(Y), X - Y \rangle \big) \\
&= \frac{\mu}{2} \mathrm{W}_2(p,q)^2 + \mathrm{D}_{\mathcal{E}}(p,q) \,,
\end{aligned}
$$

where we have applied Eq. (11). $\qquad\square$

## D.2. Parameter Space Proximal Gradient Descent

### D.2.1. ITERATION COMPLEXITY (PROOF OF THEOREM 3.3)

**Theorem 3.3** (Restated). *Suppose Assumption 3.1 holds and the gradient estimator $\widehat{\nabla_\lambda^{\text{bonnet–price}}} \mathcal{E}$ is used. Then, for any $\epsilon > 0$, there exists some $t_*$ and $\tau$ (stated explicitly in the proof) such that running stochastic proximal gradient descent with the step size schedule in Eq. (10) with $\gamma_0 = 1/(10L\kappa)$ guarantees*

$$T \gtrsim d\kappa \frac{1}{\epsilon} + \sqrt{d}\,\kappa^{3/2} \log(\kappa\Delta^2)\frac{1}{\sqrt{\epsilon}} + \kappa^2 \log\left(\Delta^2 \frac{1}{\epsilon}\right)$$

$$\Rightarrow \quad \mu\mathbb{E}[W_2(q_T, q_*)^2] \leq \epsilon\,,$$

*where $\Delta^2 = \mu\|\lambda_0 - \lambda_*\|^2$.*

This is a corollary of Proposition C.18, where the sufficient conditions are established in Lemmas 3.5 and 3.7.

*Proof.* Under the stated conditions, we can invoke Lemma 3.5, where, for any $\lambda \in \Lambda$ the generic coupling $\psi$ in the statement of Lemma 3.5 is set as $\psi = \psi^{\text{rep}} \in \Psi(q_\lambda, q_{\lambda_*})$, the coupling associated with the transport map $M_{q_\lambda \mapsto q_{\lambda_*}}^{\text{rep}}$ induced by the parametrization in Assumption 2.2. Then, for any $\lambda \in \Lambda$,

$$\mathbb{E}_{\epsilon \sim \varphi}\left[\|\widehat{\nabla_\lambda^{\text{bonnet–price}}}\mathcal{E}(q_\lambda; \epsilon) - \nabla\mathcal{E}(q_{\lambda_*})\|_2^2\right]$$

$$\leq 4L\kappa\,\mathbb{E}_{(X, X_*) \sim \psi^{\text{rep}}}[D_U(X, X_*)] + 10dL \quad \text{(Lemma 3.5)}$$
$$= 4L\kappa\,D_{\lambda \mapsto \mathcal{E}(q_\lambda)}(\lambda, \lambda_*) + 10dL\,. \quad \text{(Lemma 3.7)}$$

This establishes

$$\text{Lemma 3.5 \& Lemma 3.7} \quad \Rightarrow \quad \text{Assumption C.17}$$

with the constants $L_\epsilon = 5/2L\kappa$ and $\sigma^2 = 5dL$. Then

Assumption 3.1 & Assumption C.17 $\Rightarrow$ Proposition C.18 .

Substituting for the constants $L_\epsilon$ and $\sigma^2$, the parameters of the step size schedule in Eq. (10) become

$$\gamma_0 = \frac{1}{10L\kappa}$$
$$\tau = 8\kappa$$
$$t_* = \min\left\{\left\lceil\frac{1}{\log(1/(1 - 1/10\kappa^2))}\log\left(\frac{\kappa}{d}\Delta^2\right)\right\rceil, T\right\},$$

which guarantee

$$T \geq \max\{B_{\text{var}}, B_{\text{bias}}\} \quad \Rightarrow \quad \mu\mathbb{E}[W_2(q_T, q_*)^2] \leq \epsilon$$

with the constants

$$B_{\text{var}} = 40d\kappa\frac{1}{\epsilon} + 10\sqrt{d}\,\kappa^{3/2}$$

$$\times \left\{\log\left(\frac{\kappa}{d}\mu V_0\right) + \frac{1}{10\kappa^2} + \sqrt{2}\right\}\frac{1}{\sqrt{\epsilon}}$$

$$B_{\text{bias}} = 10\kappa^2 \log\left(2\Delta^2 \frac{1}{\epsilon}\right).$$

$\square$

### D.2.2. VARIANCE BOUND ON THE PARAMETER GRADIENT ESTIMATOR (PROOF OF LEMMA 3.5)

**Lemma 3.5** (Restated). *Suppose Assumptions 3.1 and 2.2 holds, and $\lambda_* \in \arg\min_{\lambda \in \Lambda} \mathcal{F}(q_\lambda)$. Then, for any $\lambda \in \Lambda$ and any coupling $\psi \in \Psi(q_\lambda, q_{\lambda_*})$,*

$$
\mathbb{E}_{\epsilon \sim \varphi}\big[\|\widehat{\nabla_\lambda^{\text{bonnet–price}}}\mathcal{E}(q_\lambda; \epsilon) - \nabla\mathcal{E}(q_{\lambda_*})\|_2^2\big]
$$
$$
\leq 10L\kappa\, \mathbb{E}_{(X,X_*)\sim\psi}[\mathrm{D}_U(X, X_*)] + 10dL .
$$

Recall the definition of the second-order parameter gradient

$$
\widehat{\nabla_\lambda^{\text{bonnet–price}}}\mathcal{E}(q_\lambda; \epsilon) = \begin{bmatrix} \widehat{\nabla_m^{\text{bonnet}}}\mathcal{E}(q_\lambda; \epsilon) \\ \widehat{\nabla_C^{\text{price}}}\mathcal{E}(q_\lambda; \epsilon) \end{bmatrix} = \begin{bmatrix} \nabla U(Z) \\ C^\top \nabla^2 U(Z) \end{bmatrix} ,
$$

where $Z = \phi_\lambda(\epsilon) = C\epsilon + m$. We will decompose the gradient variance into the variance of the location component $\widehat{\nabla_m^{\text{bonnet}}}\mathcal{E}(q_\lambda; \epsilon)$ and the scale component $\widehat{\nabla_C^{\text{price}}}\mathcal{E}(q_\lambda; \epsilon)$, and then bound each separately.

*Proof.* Since the location and scale compoments of the gradient estimator are orthogonal,

$$
\mathbb{E}_{\epsilon \sim \varphi}\Big[\|\widehat{\nabla_\lambda}\mathcal{E}(q_\lambda; \epsilon) - \nabla\mathcal{E}(q_{\lambda_*})\|_2^2\Big]
$$
$$
= \underbrace{\mathbb{E}_{\epsilon \sim \varphi}\Big\|\widehat{\nabla_m^{\text{bonnet}}}\mathcal{E}(q_\lambda; \epsilon) - \nabla_m\mathcal{E}(q_\lambda)\Big\|_2^2}_{\text{variance of gradient w.r.t. } m_t}
$$
$$
+ \underbrace{\mathbb{E}_{\epsilon \sim \varphi}\Big\|\widehat{\nabla_C^{\text{price}}}\mathcal{E}(q_\lambda; \epsilon) - \nabla_C\mathcal{E}(q_\lambda)\Big\|_F^2}_{\text{variance of gradient w.r.t. } \Sigma} . \quad (38)
$$

The variance of the location component can immediately be bounded by Lemma C.7.

$$
\mathbb{E}_{Z\sim q_\lambda}\Big\|\widehat{\nabla_m^{\text{bonnet}}}\mathcal{E}(q_\lambda; \epsilon) - \nabla_m\mathcal{E}(q_\lambda)\Big\|_2^2
$$
$$
\leq \mathbb{E}_{Z\sim q_\lambda}\|\nabla U(Z) - \mathbb{E}_q \nabla U\|_2^2
$$
$$
\leq 4L\, \mathbb{E}_{(X,X_*)\sim\psi}[\mathrm{D}_U(X, X_*)] + 2dL . \quad (39)
$$

On the other hand, for the scale component, we first apply Young's inequality such that

$$
\mathbb{E}_{\epsilon\sim\varphi}\|\widehat{\nabla_C^{\text{price}}}\mathcal{E}(q_\lambda; \epsilon) - \nabla_C\mathcal{E}(q_\lambda)\|_F^2
$$
$$
= \mathbb{E}_{Z\sim q_\lambda}\Big\|C^\top \nabla^2 U(Z) - C_*^\top \mathbb{E}_{q_*}[\nabla^2 U]\Big\|_F^2
$$
$$
\leq 2\underbrace{\mathbb{E}_{Z\sim q_\lambda}\|C^\top \nabla^2 U(Z)\|_F^2}_{V_{\text{mul}}} + 2\underbrace{\|C_*^\top \mathbb{E}_{q_*}[\nabla^2 U]\|_F^2}_{V_{\text{add}}} .
$$
$$
(40)
$$

Again, similarly to the Bures-Wasserstein case, the variance is dominated by the multiplicative noise of the scale $V_{\text{mul}}$, which can be bounded by Lemma C.5.

$$
V_{\text{mul}} = \mathbb{E}_{Z\sim q_\lambda}\|C^\top \nabla^2 U(Z)\|_2^2
$$

$$
= \mathbb{E}_{Z\sim q_\lambda} \mathrm{tr}\big(\nabla^2 U(Z) CC^\top \nabla^2 U(Z)\big)
$$
$$
= \mathbb{E}_{Z\sim q_\lambda} \mathrm{tr}\big(\nabla^2 U(Z) \Sigma \nabla^2 U(Z)\big)
$$
$$
\leq L\big(2\sqrt{\kappa} + \kappa\big)\mathbb{E}_{(X,X_*)\sim\psi}[\mathrm{D}_U(X, X_*)] + 3dL .
$$
$$
(41)
$$

On the other hand, $V_{\text{add}}$ immediately follows from the properties of the stationary point (Proposition C.4) as

$$
V_{\text{add}} = \|C_*^\top \Sigma_*^{-1}\|_F^2
$$
$$
= \mathrm{tr}\big(\Sigma_*^{-1} C_* C_*^\top \Sigma_*^{-1}\big)
$$
$$
= \mathrm{tr}\big(\Sigma_*^{-1} \Sigma_* \Sigma_*^{-1}\big)
$$
$$
= \mathrm{tr}\big(\Sigma_*^{-1}\big) . \quad (42)
$$

Therefore,

$$
\mathbb{E}_{\epsilon\sim\varphi}\|\widehat{\nabla_C^{\text{bonnet–price}}}\mathcal{E}(q_\lambda; \epsilon) - \nabla_C\mathcal{E}(q_\lambda)\|_F^2
$$
$$
\leq 2V_{\text{mul}} + 2V_{\text{add}}
$$
$$
\leq \Big\{L\big(4\sqrt{\kappa} + 2\kappa\big)\mathbb{E}_{(X,X_*)\sim\psi}[\mathrm{D}_U(X, X_*)] + 6dL\Big\}
$$
$$
+ \mathrm{tr}\big(\Sigma_*^{-1}\big)
$$
$$
\leq L\big(4\sqrt{\kappa} + 2\kappa\big)\mathbb{E}_{(X,X_*)\sim\psi}[\mathrm{D}_U(X, X_*)] + 8dL . \quad (43)
$$

where we used the fact that, from Proposition C.4,

$$
\Sigma_*^{-1} = \mathbb{E}_{q_*} \nabla^2 U \preceq L\mathrm{I}_d .
$$

Combining Eqs. (43), (39) and (42) into Eq. (38),

$$
\mathbb{E}_{\epsilon\sim\varphi}\|\widehat{\nabla_\lambda^{\text{bonnet–price}}}\mathcal{E}(q_\lambda; \epsilon) - \nabla\mathcal{E}(q_{\lambda_*})\|_2^2
$$
$$
\leq \Big\{4L\, \mathbb{E}_{(X,X_*)\sim\psi}[\mathrm{D}_U(X, X_*)] + 2dL\Big\}
$$
$$
+ \Big\{L\big(4\sqrt{\kappa} + 2\kappa\big)\mathbb{E}_{(X,X_*)\sim\psi}[\mathrm{D}_U(X, X_*)] + 8dL\Big\}
$$
$$
\leq 10L\kappa\, \mathbb{E}_{(X,X_*)\sim\psi}[\mathrm{D}_U(X, X_*)] + 10dL ,
$$

where we used the fact that $\kappa \geq 1$. $\qquad \square$

### D.2.3. BREGMAN DIVERGENCE IDENTITY (PROOF OF LEMMA 3.7)

**Lemma 3.7** (Restated). *Suppose Assumption 2.2 hold. Then, for any $\lambda, \lambda' \in \Lambda$, denote the coupling induced by the transport map $M^{\mathrm{rep}}_{q_\lambda \mapsto q_{\lambda'}}$ as $\psi^{\mathrm{rep}}$. Then*

$$\mathbb{E}_{(X,X') \sim \psi^{\mathrm{rep}}}[\mathrm{D}_U(X, X')] = \mathrm{D}_{\lambda \mapsto \mathcal{E}(q_\lambda)}(\lambda, \lambda') .$$

*Proof.* By definition,

$$\mathbb{E}_{(X,X') \sim \psi^{\mathrm{rep}}} \mathrm{D}_U(\lambda, \lambda')$$
$$= \mathbb{E}_{(X,X') \sim \psi^{\mathrm{rep}}}[U(X) - U(X') - \langle \nabla U(X'), X - X' \rangle]$$
$$= \mathcal{E}(q_\lambda) - \mathcal{E}(q_{\lambda'}) - \mathbb{E}_{(X,X') \sim \psi^{\mathrm{rep}}}[\langle \nabla U(X'), X - X' \rangle] .$$

Therefore, we only need to show that

$$\mathbb{E}_{(X,X') \sim \psi^{\mathrm{rep}}}[\langle \nabla U(X'), X - X' \rangle] = \langle \nabla_\lambda \mathcal{E}(q_\lambda), \lambda - \lambda' \rangle ,$$

which is essentially Eq. (13). Denoting $\lambda = (m, \mathrm{vec}\, C)$ and $\lambda' = (m', \mathrm{vec}\, C')$, this follows from

$$\langle \nabla_\lambda \mathcal{E}(q_\lambda), \lambda - \lambda' \rangle$$
$$= \left\langle \begin{bmatrix} \mathbb{E}_{\epsilon \sim \varphi} \nabla U(\phi_\lambda(\epsilon)) \\ \mathbb{E}_{\epsilon \sim \varphi} \nabla U(\phi_\lambda(\epsilon)) \epsilon^\top \end{bmatrix}, \begin{bmatrix} m - m' \\ C - C' \end{bmatrix} \right\rangle$$
$$= \langle \mathbb{E}_{\epsilon \sim \varphi} \nabla U(\phi_\lambda(\epsilon)), m - m' \rangle$$
$$\qquad + \langle \mathbb{E}_{\epsilon \sim \varphi} \nabla U(\phi_\lambda(\epsilon)) \epsilon^\top, C - C' \rangle_{\mathrm{F}}$$
$$= \mathbb{E}_{\epsilon \sim \varphi} \Big[ \langle \nabla U(\phi_\lambda(\epsilon)), m - m' \rangle$$
$$\qquad + \langle \nabla U(\phi_\lambda(\epsilon)), (C - C') \epsilon \rangle_{\mathrm{F}} \Big]$$
$$= \mathbb{E}_{\epsilon \sim \varphi} \langle \nabla U(\phi_\lambda(\epsilon)), (C\epsilon + m) - (C'\epsilon + m') \rangle$$
$$= \mathbb{E}_\varphi \langle \nabla U(\phi_\lambda), \phi_\lambda - \phi_{\lambda'} \rangle$$
$$= \mathbb{E}_{(Z,Z') \sim \psi^{\mathrm{rep}}} \langle \nabla U(Z'), Z - Z' \rangle .$$

$\square$

### D.2.4. COERCIVITY OF THE PARAMETER GRADIENT (PROOF OF LEMMA 3.9)

**Lemma 3.9** (Restated). *Suppose Assumptions 3.1 and 2.2 hold. Then, for any $\lambda \in \Lambda$ and $\lambda_* = \arg\min_{\lambda \in \Lambda} \mathcal{F}(q_\lambda)$,*

$$\langle \nabla_{\lambda_t} \mathcal{E}(q_{\lambda_t}) - \nabla_{\lambda_*} \mathcal{E}(q_{\lambda_*}), \lambda_t - \lambda_* \rangle$$
$$\geq \frac{\mu}{2} \|\lambda - \lambda_*\|_2^2 + \mathrm{D}_{\lambda \mapsto \mathcal{E}(q_\lambda)}(\lambda, \lambda_*) .$$

Under Assumptions 2.2 and 3.1, the energy is $\mu$-strongly convex.

**Lemma D.2** (Theorem 9; Domke, 2020). *Suppose Assumptions 3.1 and 2.2 hold. Then $\lambda \mapsto \mathcal{E}(q_\lambda)$ is $\mu$-strongly convex.*

An alternative proof is also presented by Kim et al. (2023, Theorem 2). Regardless of the way we prove Lemma D.2, coercivity immediately follows from the basic properties of strongly convex functions.

*Proof of Lemma 3.9.*

$$\langle \nabla_{\lambda_t} \mathcal{E}(q_{\lambda_t}) - \mathcal{E}(q_{\lambda_*}), \lambda_t - \lambda_* \rangle$$
$$= \langle \nabla_{\lambda_t} \mathcal{E}(q_{\lambda_t}), \lambda_t - \lambda_* \rangle - \langle \nabla_{\lambda_*} \mathcal{E}(q_{\lambda_*}), \lambda_t - \lambda_* \rangle .$$

Applying Lemma D.2,

$$\geq \frac{\mu}{2} \|\lambda_t - \lambda_*\|_2^2$$
$$+ (\mathcal{E}(q_\lambda) - \mathcal{E}(q_{\lambda_*}) - \langle \nabla_{\lambda_*} \mathcal{E}(q_{\lambda_*}), \lambda_t - \lambda_* \rangle)$$
$$= \frac{\mu}{2} \|\lambda_t - \lambda_*\|_2^2 + \mathrm{D}_{\lambda \mapsto \mathcal{E}(q_\lambda)}(\lambda_t, \lambda_*) .$$

$\square$

