# OpenReview forum: "Stochastic Gradient Variational Inference with Price's Gradient Estimator from Bures-Wasserstein to Parameter Space"
_ICML.cc/2026/Conference — ICML 2026 regular_

### Official Review · Reviewer_SLgF · 2026-03-06

**Soundness:** 4
**Presentation:** 4
**Significance:** 2
**Originality:** 3
**Overall Recommendation:** 4
**Confidence:** 3

**Summary:**

In this paper, the authors study stochastic variational inference in the full-rank Gaussian family and compares two optimisation perspectives: stochastic proximal Bures–Wasserstein gradient descent (SPBWGD), which operates in the Bures–Wasserstein geometry of Gaussian measures, and stochastic proximal gradient descent (SPGD), which operates in Euclidean parameter space. The key idea is that previously reported advantages of Wasserstein variational inference are not mainly due to the underlying geometry, but rather due to the use of a stronger gradient estimator based on Price’s theorem, which incorporates Hessian information.

The paper makes two main contributions. First, it shows that the same Price-type second-order estimator used in SPBWGD can also be used in SPGD, which gives matching state-of-the-art non-asymptotic complexity under strong convexity and smoothness assumptions on the target potential. Second, it improves the analysis of SPBWGD by improving the dependence on step sizes through a variance analysis based on the Bregman divergence. The experiments support the claim that, once both methods use Price’s gradient, the performance gap between SPGD and SPBWGD becomes small, while both degrade when using first-order reparameterization gradients.

**Compliance With Llm Reviewing Policy:**

Affirmed.

**Final Justification:**

I appreciate the authors' responses to my questions. I believe that the paper provides a nice theoretical contribution to the literature and offers new insights. But I still feel that the paper is not a strongly significant contribution for ICML and so on balance and therefore I would maintain my score as "Weak Accept".

**Key Questions For Authors:**

1. The main theoretical gains rely on Price’s gradient, which requires Hessian information and raises the per-iteration cost substantially. Can the authors provide runtime-normalised or cost-normalised comparisons, rather than only iteration-normalised ones? A convincing wall-clock analysis would substantially strengthen the practical significance of the paper. Also, are there approximations to the Hessian that could be used which would be computationally faster but may preserve the theoretical benefits.

2. The paper strongly suggests that variance control in the stochastic gradients is the main reason these methods work better with Price’s estimator. Can the authors provide direct empirical evidence for this, for example by measuring estimator variance, signal-to-noise ratio, or related diagnostics? On this point, I wonder what the impact would be if you used stochastic gradients with variance reduction techniques?

3. To what extent do the authors expect the “estimator matters more than geometry” conclusion to persist beyond the strongly convex full-rank Gaussian setting considered here? Some discussion or perhaps empirical evidence for nonconvex targets or richer variational families would help clarify the scope of the contribution.

**Limitations:**

Yes

**Strengths And Weaknesses:**

Strengths

The paper is technically strong and very well-written. The authors do a great job of clearly addressing a well-posed question. The comparisons they make do seem to be fairer than much of the prior literature because they align the gradient estimators across the two optimisation viewpoints. I think this is quite important if one wants to determine whether geometry or estimator quality is responsible for an observed improvement, and so comparing methods under mismatched estimators alone is not sufficiently informative. This paper corrects that issue directly.

The theoretical contributions appear to be accurate as far as I can tell. The proofs are clearly strutured and mainly utilise standard results and inequalies which are commonly used in this literature with a few interesting new ideas. For example, the use of a Bregman-divergence-based variance control argument appears to be a new technical refinement rather than a superficial proof adjustment. The resulting improvement in the SPBWGD analysis, together with the matching complexity result for SPGD with Price’s gradient, provides a strong theoretical story.

The paper is also reasonably clear and well-structured. The main message is easy to follow, and the narrative is well organised around a concrete conceptual question: whether the advantage of Wasserstein VI comes from geometry or from the stochastic gradient estimator. This framing makes the paper accessible even to readers who are not specialists in Bures–Wasserstein methods.
Finally, the paper does offer a useful conceptual insight. Even if the end result is somewhat deflationary, showing that the apparent advantage of WVI is largely an estimator effect rather than a geometric one is a worthwhile clarification for the literature.

Weaknesses

I think the main weakness is significance. While the paper provides some interesting theoretical improvements and a clearer comparison than prior work, the overall contribution is not especially exciting. The core message is largely that standard parameter-space variational inference can match Wasserstein VI once it is given the same second-order estimator. That is a useful correction, but it feels more incremental than transformative.

A second weakness is the limited practical relevance of the Hessian-based estimator. The paper’s theoretical improvements rely on Price’s gradient, which requires Hessian information. This substantially increases computational cost, and in dense full-rank Gaussian settings the per-iteration complexity would be at least cubic. It's therefore not clear that the improved iteration complexity translates into improved practical performance when wall-clock time is taken into account. I think this is the central issue for the paper if it is to appear at ICML. The results suggest that controlling gradient variance is important, but it remains unclear whether exact Hessians are a practically viable way to achieve that benefit.

Relatedly, the experimental evaluation is somewhat incomplete. The experiments support the qualitative claim that estimator choice matters more than geometry, but they do not convincingly address the computational tradeoff. I would have liked to see runtime-normalised comparisons, scaling studies, or stronger evidence that the second-order estimator is worthwhile relative to its added cost. Without this, the practical message remains a bit incomplete.

There is also some mismatch between the strength of the theoretical results and the scope of the conclusions. The analysis is limited to strongly convex, smooth targets and the full-rank Gaussian family. Those assumptions are standard for this kind of theory, but they also make the setting fairly narrow. The paper does not fully establish how much of the main lesson carries over to more realistic nonconvex posteriors or richer variational families.

---

> ### Author Rebuttal · Authors · 2026-03-30
>
> We thank the Reviewer for their review.
>
> > While the paper provides some interesting theoretical improvements and a clearer comparison than prior work, the overall contribution is not especially exciting. The core message is largely that standard parameter-space variational inference can match Wasserstein VI once it is given the same second-order estimator. That is a useful correction, but it feels more incremental than transformative.
>
> We would like to clarify that our main contribution is a bit more high-level than simply showing that parameter-space VI can match Wasserstein VI. Prior work often attributes the empirical advantages of Wasserstein VI to its geometry; our results show that a significant portion of this advantage instead arises from the use of higher-quality (second-order) estimators. This distinction is important because it challenges a prevailing interpretation in the literature and suggests that improvements previously attributed to Wasserstein geometry may, in fact, be achievable within standard VI frameworks. In this sense, our work makes the point that we hope will guide the design of stochastic gradient variational inference algorithms: the effect of the choice of gradient estimator should be carefully considered.
>
> > The paper’s theoretical improvements rely on Price’s gradient, which requires Hessian information. This substantially increases computational cost, and in dense full-rank Gaussian settings the per-iteration complexity would be at least cubic. It's therefore not clear that the improved iteration complexity translates into improved practical performance when wall-clock time is taken into account.
> >
> > The experiments support the qualitative claim that estimator choice matters more than geometry, but they do not convincingly address the computational tradeoff. I would have liked to see runtime-normalised comparisons, scaling studies, or stronger evidence that the second-order estimator is worthwhile relative to its added cost. Without this, the practical message remains a bit incomplete.
>
> Again, our work is of theoretical interest, and the experiments have been designed to evaluate the theory rather than the real-world performance of these algorithms. Also, in the Discussions (Section 5), we state: *“On a practical note, it isn’t clear if Price’s gradient is always better. For instance, at each iteration, SPGD with the reparametrization gradient requires $\Omega(d^2)$ operations (matrix-vector product). Moving to second order increases the cost to $\Omega(d^3)$ operations (matrix-matrix product). Meanwhile, SPBWGD requires $\Omega(d^3)$ in both cases. Therefore, ... BBVI with SPGD and the reparametrization gradient could be more efficient ... depending on ... $\kappa$.”*
>
> > The analysis is limited to strongly convex, smooth targets and the full-rank Gaussian family. Those assumptions are standard for this kind of theory, but they also make the setting fairly narrow. The paper does not fully establish how much of the main lesson carries over to more realistic nonconvex posteriors or richer variational families.
>
> We very much agree with this comment. However, it is currently very difficult to analyze gradient-based VI in settings more general than strongly convex-smooth targets when involving stochastic gradients. While tools for analyzing the non-convex setting exist [1], they require strong assumptions or unrealistic algorithmic design choices (*e.g.*, fixing the number of iterations in advance). Possibly due to this reason, [2] only present results for the deterministic non-convex setting and skip the stochastic non-convex setting. The same limitation also applies to richer variational families such as Gaussian mixtures. However, in practice, stochastic gradient variational inference is typically performed using location-scale variational families. Therefore, we believe our setting covers the widest range of use cases.
>
> > The paper strongly suggests that variance control in the stochastic gradients is the main reason these methods work better with Price’s estimator. Can the authors provide direct empirical evidence for this, for example by measuring estimator variance, signal-to-noise ratio, or related diagnostics?
>
> We chose not to present results comparing the gradient variance at a specific point in the optimization space as this can be misleading: unless one gradient estimator completely dominates another across the whole optimization space, the gradient variance may be large or small depending on where it is evaluated. Therefore, we find that comparing the convergence speed is the most formal and correct way to evaluate gradient estimators, provided that the effect of the step size can be properly controlled.
>
> 1. Kim *et al.* (2023). On the convergence of black-box variational inference. *NeurIPS*.
> 2. Diao *et al.* (2023). Forward-backward Gaussian variational inference via JKO in the Bures-Wasserstein space. *ICML*.

---

> > ### Author Rebuttal · Reviewer_SLgF · 2026-04-01
> >
> > Thank you to the authors for their response to my comments. I do not have any further questions.
> >
> > I have decided to maintain my score of "Weak Accept" based on the view that the paper provides a nice theoretical contribution to the literature and offers new insights. But I still feel that the paper is not a strongly significant contribution for ICML and so on balance I think "Weak Accept" is appropriate.

---

### Official Review · Reviewer_dZ7A · 2026-03-10

**Soundness:** 4
**Presentation:** 3
**Significance:** 3
**Originality:** 3
**Overall Recommendation:** 4
**Confidence:** 4

**Summary:**

In this paper, the authors study the connection between Wasserstein Variational Inference (WVI) and Black-box Variational Inference (BBVI) for the famility of Gaussian distributions with non-degenerate covariance matrices: trying to match a Gaussian $\mathcal{N}(m,\Sigma)$ to a target $\pi \propto e^{-U}$. The paper focuses on the backward-forward scheme for both WVI and BBVI setting where the gradients and proximal operators are carried out in the Wasserstein and Euclidean geometries, respectively.  Existing works have shown that WVI has better convergence complexity bounds than BBVI interms of the interplay between the condition number $\kappa$ of the log-target and the approximation error $\epsilon$. The paper argues that this improvement is more about the choice of stochastic estimators for the gradients of each setting, rather than the underlying geometry of these schemes. To this end, the contributions are as follows:
- They first improved the complexity of the WVI from [Diao et al. 2023], setting ground for a fair comparison.
- Via the Price's theorem and the Stein's identity, they show that there are in fact two ways to obtain estimator for the gradient, one relates to the second order $\nabla^2 U$ and the other one relates to $\nabla U$. The first one is called Price's estimator while the second one is called Reparameterization estimator. WVI uses Price's estimator while BBVI uses Reparameterization estimator.

- The authors show that BBVI with Price's estimator can achieve comparable complexity to the above (improved) complexity of WVI. Hence bridging the theoretical gap between two directions.

**Compliance With Llm Reviewing Policy:**

Affirmed.

**Final Justification:**

The paper provides new insights into parameter-space Gaussian VI and Wasserstein Gaussian VI, effectively bridging the gap between the two approaches. I find the work theoretically interesting, and the presentation is a strong aspect of the paper. I assigned a score of 4 (weak accept) because the empirical evaluation could be more comprehensive and thorough.

**Key Questions For Authors:**

- Can we compare the variance of Price's estimator and Reparametrization estimator?

**Limitations:**

Limitations not discussed thoroughly, although there are no obvious limitations.

**Strengths And Weaknesses:**

# Strength
- The paper is well-written and easy to follow. The derivations are insightful.
- From the connection, BBVI now can use Price's estimator to improve its performance (although the other way around is not helpful: WVI with reparametrization performs worse).
- The complexity improvement upon [Diao et al. 2023] for WVI, and the complexity improvement for BBVI with Price's estimator that matches that of WVI.

# Weakness
- The experiments do not shed light to the theoretical bounds. Since these bounds depend on the error $\epsilon$ and the condition number of the log-target, it would be informative if we can verify this depedence in practice, e.g., how the convergence is affected by $\kappa$?

---

> ### Author Rebuttal · Authors · 2026-03-30
>
> We thank the Reviewer for their review.
>
> > The experiments do not shed light to the theoretical bounds. Since these bounds depend on the error $\epsilon$ and the condition number kappa of the log-target, it would be informative if we can verify this depedence in practice, e.g., how the convergence is affected by $\kappa$?
>
> Our experimental results should be understood qualitatively rather than quantitatively. For instance, if the theory suggests a different dependence on $\kappa$ or $\epsilon$, the empirical results would exhibit a corresponding difference in convergence speed and robustness to the step size. After all, none of the problems we are interested in practice are strongly log-concave and log-smooth. In our case, the bounds suggest that Hessian-based methods should perform similarly, while the reparametrization gradient should behave significantly worse (due to its poorer dependence on conditioning and dimension), and the experiments demonstrate this behavior.
>
> > Can we compare the variance of Price's estimator and Reparametrization estimator?
>
> We chose not to present results comparing the gradient variance at a specific point in the optimization space. While this has been a common way to evaluate estimators in past works, it can be misleading: unless one gradient estimator completely dominates another across the whole optimization space, the gradient variance may be large or small depending on where it is evaluated. Therefore, find that comparing the convergence speed is the most formal and correct way to evaluate gradient estimators, provided that the effect of the step size can be properly controlled.

---

> > ### Author Rebuttal · Reviewer_dZ7A · 2026-04-02
> >
> > Thank you for your response. I keep the positive rating on the work.

---

### Official Review · Reviewer_j2pT · 2026-03-12

**Soundness:** 3
**Presentation:** 3
**Significance:** 3
**Originality:** 3
**Overall Recommendation:** 5
**Confidence:** 2

**Summary:**

The authors study the behaviour of Wasserstein Variational Inference (WVI) and black-box variational inference (BBVI). In particular, the authors consider the Gaussian variational family, for which WVI has better convergence guarantees than BBVI. The authors argue that this difference can be traced back to the gradient estimator of WVI. With a small adjustment the authors use the same gradient estimator for BBVI and derive the same convergence complexity for both (which is also claimed to be state-of-the-art).

**Compliance With Llm Reviewing Policy:**

Affirmed.

**Final Justification:**

I recommend this paper for acceptance. I had only minor concerns and the short rebuttal addressed most of them. Other reviewer's comments also go in a similar direction, highlighting that the paper is well written, the math is sound, and the problem addessed is relevant. As I do not consider myself as an expert in the area, my overall confidence is still low.

**Key Questions For Authors:**

As I am not an expert in this specific subfield, I expressed a low confidence in my score. I am looking forward to reading the thoughts of other reviewers and I will take those into account for my final review. I have only one question concerning the evaluation and some typos I noticed.

### Questions
1. Concerning empirical validation, the experiments in the baseline agree with the statement that the Hessian based update is better than reparameterized gradients but maybe do not explore different geometries well enough. I believe that in order to argue that this claim was shown experimentally, more evaluation is needed (potentially also only on toy examples)

### Typos
Line 131 (right): rerplacing → replacing

Line 150 (right): equation outside of margins

Line 139 (right): “and” is out of place and should be removed

Line 286 (left): equation outside of margins

Line 301 (right): equation outside of margins

Line 407 (right): reference should be outside of parenthesis

**Limitations:**

yes

**Strengths And Weaknesses:**

### Strengths:
1. The paper is well written and easy to follow
2. The proposed approach is well-introduced and the math seems solid
3. The interpretation of the superiority of WVI in terms of gradient estimator is interesting, especially because it translated to BBVI as well


### Weaknesses:
1. Limited empirical analysis. The theoretical claim is supported empirically by comparing reparameterized gradients with hessian-based gradients. While the superiority of the latter is not a surprise, I wonder whether one can rule out this easily for other reasons like geometry. For a more actionable discussion see the questions section

---

> ### Author Rebuttal · Authors · 2026-03-30
>
> We thank the Reviewer for their review.
>
> > Limited empirical analysis. The theoretical claim is supported empirically by comparing reparameterized gradients with hessian-based gradients. While the superiority of the latter is not a surprise, I wonder whether one can rule out this easily for other reasons like geometry. For a more actionable discussion see the questions section
>
> > Concerning empirical validation, the experiments in the baseline agree with the statement that the Hessian based update is better than reparameterized gradients but maybe do not explore different geometries well enough. I believe that in order to argue that this claim was shown experimentally, more evaluation is needed (potentially also only on toy examples)
> ​
>
> We would like to clarify that the conclusion that the role of the geometry is minimal holds only for the comparison between the Euclidean parameter space and the Wasserstein space. It’s very much possible that using other geometries, such as that induced via the KL pseudo-metric as in natural gradient methods, the results may come out differently. We mention this fact in the Discussions (Section 5): *“However, this doesn’t completely rule out the possibility that measure-space algorithms can be more effective. NGVI, which uses the Fisher metric, … empirical evidence suggests that NGVI methods can converge significantly faster than BBVI (Lin et al., 2019). However, our theoretical understanding of NGVI is still limited …”* A more complete analysis would have involved results on NGVI, but our current theoretical tools are unfortunately insufficient for this. Establishing the convergence of NGVI under comparable assumptions itself is a major open problem.

---

> > ### Author Rebuttal · Reviewer_j2pT · 2026-04-03
> >
> > I would like to thank the authors for their rebuttal. I have now read other reviewer's comments and will keep my scorse as is, recommending for acceptance.

---

### Official Review · Reviewer_1WZG · 2026-03-12

**Soundness:** 4
**Presentation:** 4
**Significance:** 4
**Originality:** 3
**Overall Recommendation:** 5
**Confidence:** 3

**Summary:**

The paper compares existing methods for Bures-Wasserstein and black-box variational inference (specifically, stochastic proximal gradient descent-based BBVI) by controlling for the difference in gradient estimators typically implemented alongside these two algorithms. In both theoretical analyses of iteration complexity and empirical comparisons of variational free-energy achieved at convergence, they find that the primary difference in reported performance between BWVI and BBVI comes down to the former's use of a second-order gradient estimator for updating the covariance matrices of Gaussian variational proposals.  Experimental problems are taken from a standard benchmark for Bayesian inference and the two competing gradient estimators are tested across step-sizes.

**Compliance With Llm Reviewing Policy:**

Affirmed.

**Key Questions For Authors:**

1) Was the mention of NGVI a typo?
2) Is the proximal operator applied in performing Euclidean-space variational inference just for the sake of strict one-to-one comparability to the use of the proximal operator in BWGD to ensure parameter validity after gradient updates?

**Limitations:**

yes

**Strengths And Weaknesses:**

Strengths:
* Combined theoretical proofs with empirical results
* Software implementation connects to a probabilistic programming ecosystem
* Experiments come from an established benchmark of Bayesian inference problems, PosteriorDB
* Demonstrating a scenario where typical reparameterization-based VI fails to converge, across learning rates, provides a strong motivation for investigating gradient estimators with better convergence properties.

Weaknesses:
* The paper mentions natural-gradient variational inference (NGVI) in Section 4, but then never features any experiments on NGVI.
* Paper could use measurements of gradient-estimator variance, although these would not be strictly necessary.

---

> ### Author Rebuttal · Authors · 2026-03-30
>
> We thank the Reviewer for their review.
>
> > The paper mentions natural-gradient variational inference (NGVI) in Section 4, but then never features any experiments on NGVI.
>
> That is indeed a typo. Thanks for spotting this. This has now been fixed.​
>
> > Paper could use measurements of gradient-estimator variance, although these would not be strictly necessary.
>
> Indeed, while directly comparing the gradient variance has been a common way to evaluate estimators, this can be misleading: unless one gradient estimator completely dominates another across the whole optimization space, the gradient variance may be large or small depending on where it is evaluated. Therefore, we find that comparing the convergence speed is the most formal and correct way to evaluate gradient estimators, provided that the effect of the step size can be properly controlled.
>
> > Is the proximal operator applied in performing Euclidean-space variational inference just for the sake of strict one-to-one comparability to the use of the proximal operator in BWGD to ensure parameter validity after gradient updates?
>
> This is a great question. The Reviewer's suggested reason is one, but there is also a practical reason. As explained in the Background (Section 2), for both parameter space and Wasserstein space optimization, the free energy/evidence lower bound is a non-smooth function(al). Therefore, to ensure robust/stable performance, either projections onto a compact set (a closed set of covariance matrices with lower-bounded eigenvalues) or proximal operators should be used. Now, for strongly log-smooth targets, the theory suggests how to set the eigenvalue bound [1]. However, in practice, this results in another hyperparameter that needs to be tuned manually. (A common way to side-step the need for a compact set has been to use elaborate covariance matrix parametrizations, but this also has significant downsides both theoretically and practically. See [2].) Lastly, the theory suggests that, under strong log-concavity and log-smoothness, proximal gradient methods can use step sizes up to twice as large [3]. Such robustness has also been observed empirically in [1,2,4].
>
> 1. Domke, J. (2020, November). Provable smoothness guarantees for black-box variational inference. In International Conference on Machine Learning (pp. 2587-2596). PMLR.
> 2.  Kim, K., Oh, J., Wu, K., Ma, Y., & Gardner, J. (2023). On the convergence of black-box variational inference. Advances in Neural Information Processing Systems, 36, 44615-44657.
> 3. Domke, J., Gower, R., & Garrigos, G. (2023). Provable convergence guarantees for black-box variational inference. Advances in neural information processing systems, 36, 66289-66327.
> 4. Diao, M. Z., Balasubramanian, K., Chewi, S., & Salim, A. (2023, July). Forward-backward Gaussian variational inference via JKO in the Bures-Wasserstein space. In International Conference on Machine Learning (pp. 7960-7991). PMLR.

---

> > ### Author Rebuttal · Reviewer_1WZG · 2026-04-03
> >
> > I thank the authors for explaining their use of proximal operators.

---

### Decision · Program_Chairs · 2026-04-30

**Decision:**

Accept (regular)

**Comment:**

The paper shows that the apparent advantage of Wasserstein VI over standard Gaussian VI comes mainly from the Hessian-based Price gradient estimator, and that BBVI can match WVI iteration complexity when given the same estimator.

While the reviewers praise the clean framing, strong theory, fair estimator-vs-geometry comparison, clear writing, and useful correction to the prior narrative around WVI, they also raise several concerns, including modest significance, limited runtime-normalized practical analysis, and the heavy Hessian cost of Price’s estimator.

The authors' rebuttals clarified the role of proximal operators, corrected minor issues like the NGVI typo, explicitly discussed the practical cost tradeoff of Hessian-based estimators, and emphasized that the experiments were meant to test the theory rather than optimize wall-clock performance.
The reviewers indicated that these responses have addressed most of their criticisms.

Therefore, we have decided to accept the paper for presentation at ICML. We would still recommend that the authors take the reviewers' feedback into account when preparing the camera-ready version.